# Programmed cell revival from imminent cell death enhances tissue repair and regeneration

Kollori Dhar[1,2,3,13], Kautilya Kumar Jena [2,13], Subhash Mehto[4], Rinku Sahu [2,3], Krushna C Murmu [2,3], Atharva Anand Mahajan [5,6], Sibaram Behera[7], Ravi Kiran Putchala[8], Reuben Jacob Mathew [1], Ramyasingh Bal[2], Soumya Kundu[1], Santosh Kumar Das[1,9], Swati Chauhan [2], Sameekshya Satapathy[1,9], Rina Yadav[2,3], Swatismita Priyadarsini[2], Khyathi Ratna Padala [1], Prashanth Namdigalla[1], Sanchita Mishra[10], Prerana Muralidhara[11], Kushagra Bansal[11], Kesavardhana Sannula [10], Punit Prasad[2], Kiran Kumar Bokara[1,9], Divya Tej Sowpati [1,9], Anindya Ghosh-Roy [7], Pravati Kumari Mahapatra[12], Rohan Jayant Khadilkar [5,6], Ramesh Yelagandula[8] & Santosh Chauhan [1,2,9] ✉

## Abstract

**Cell recovery from near-death states is a critical yet poorly understood aspect of cell biology. Here, we describe a tightly-regulated programmed cell revival process after exposure of cells to cell death-inducing lysosomotropic agents, such as L-leucyl-L-leucine methyl ester (LLOMe). In the initial stage of cell recovery, we observe increased chromatin accessibility and upregulation of genes and pathways associated with embryonic development, regeneration, stemness, and inflammation. Subsequently, vital pathways governing metabolism, organelle biogenesis, membrane trafficking, transport, and cytoskeleton remodeling are activated, resulting in the complete renewal of cells. Consistent with the links of this transcriptional profile to tissue repair and regeneration, we found LLOMe to enhance the healing of skin wounds and corneal alkali burns in mice, promote hematopoietic progenitor/stem cell production in *Drosophila melanogaster*, induce tadpole tail regeneration in frogs, and mediate axon regeneration in *Caenorhabditis elegans*. Using both genetic and pharmacological approaches, we show NF-κB signaling to be critical for both cell revival and regeneration. This study characterizes cell revival from near-death conditions as a programmed cell-intrinsic mechanism, which could be harnessed for therapeutic applications in regenerative medicine.**

**Keywords** Lysosomal Cell Death and Revival; Programmed Cell Revival; LLOMe; Anastasis; NF-κB Signaling in Cell Revival and Regeneration
**Subject Categories** Autophagy & Cell Death; Signal Transduction; Stem Cells & Regenerative Medicine

See also: B Sundaram and T-D Kanneganti

## Introduction

The classical perspective on cell death is that once the cell is committed to it, the process is irreversible (Alberts et al, 2014; Fuchs and Steller, 2011). Recent studies challenged the dogma that dying cells can recover, even after the initiation of the cell death process (Ding et al, 2016; Gong et al, 2017; Nano et al, 2023; Sun et al, 2017; Tang et al, 2012). Nevertheless, the lack of reagents and methods to investigate the process of cell resuscitation has perpetually caused uncertainty regarding the existence, mechanism, and significance of this phenomenon.

Lysosomal membrane permeabilization (LMP), leading to leakage of lysosomal content into the cytosol, results in lysosomal cell death (Boya and Kroemer, 2008; Kavcic et al, 2017; Villamil Giraldo et al, 2014; Wang et al, 2018). If the leakage is massive, the cell dies by necrosis; however, controlled release of cathepsins initiates a cascade of cell signaling events leading to apoptosis (Boya and Kroemer, 2008; Kavcic et al, 2017; Villamil Giraldo et al, 2014; Wang et al, 2018). Various external and internal stimuli can trigger LMP, including lysosomotropic agents. Among these agents, L-Leucyl-l-leucine methyl ester (LLOMe) is the most studied (Brojatsch et al, 2015; Droga-Mazovec et al, 2008; Kavcic et al,

¹CSIR-Centre for Cellular and Molecular Biology, Hyderabad, Telangana 500007, India. ²BRIC-Institute of Life Sciences, Bhubaneswar 751023, India. ³Regional Centre for Biotechnology, NCR Biotech Science Cluster, Faridabad 121001, India. ⁴Biosciences and Bioengineering, Indian Institute of Technology Dharwad, Karnataka 580011, India. ⁵Stem Cell and Tissue Homeostasis Laboratory, Advanced Centre for Treatment, Research and Education in Cancer (ACTREC), Tata Memorial Centre, Kharghar, Navi Mumbai, Maharashtra 410210, India. ⁶Homi Bhabha National Institute, Training School Complex, Anushakti Nagar, Mumbai 400085, India. ⁷Department of Cellular & Molecular Neuroscience, National Brain Research Centre, Manesar, Haryana, India. ⁸Laboratory of Epigenetics, Cell Fate and Disease, Centre for DNA Fingerprinting and Diagnostics, Hyderabad, India. ⁹Academy of Scientific and Innovative Research (AcSIR), Ghaziabad 201002, India. ¹⁰Department of Biochemistry, Division of Biological Sciences, Indian Institute of Science, Bengaluru, Karnataka 560012, India. ¹¹Molecular Biology and Genetics Unit, Jawaharlal Nehru Centre for Advanced Scientific Research, Jakkur, Bangalore, Karnataka 560064, India. ¹²Department of Zoology, Utkal University, Bhubaneswar 751004, India. ¹³These authors contributed equally: Kollori Dhar, Kautilya Kumar Jena. ✉E-mail: schauhan@ccmb.res.in

2020; Lima et al, 2013; Thiele and Lipsky, 1992; Uchimoto et al, 1999). LLOMe polymerizes inside lysosomes, causing rapid but reversible damage to the lysosomal membrane. The mechanism involves the accumulation of LLOMe in the lysosomes, followed by cathepsin-mediated polymerization of LLOMe into its membrano-lytic detergent form. This triggers LMP and the release of cathepsins into the cytosol, initiating the intrinsic pathway of apoptosis (Kavcic et al, 2020; Thiele and Lipsky, 1990; Uchimoto et al, 1999). At low concentrations, LLOMe perturbs multiple signaling pathways, including TFEB, autophagy, and mTOR (Jia et al, 2018; Nakamura et al, 2020; Yabuki et al, 2021). Similar to LLOMe, glycyl-L-phenylalanine 2-naphthylamide (GPN) and sphingosine accumulate in lysosomes and cause membrane damage, resulting in $Ca^{2+}$ release (Hoglinger et al, 2015; Morgan and Galione, 2021; Morgan et al, 2020) and apoptosis (Castelli et al, 2022; Cuvillier, 2002).

This study found that cells could be resuscitated from early cell death conditions induced by sublethal concentrations of lysosomotropic agents, including LLOMe, GPN, and sphingosine. The LLOMe-induced cell death phase is characterized by extensive membrane blebbing, an annexin-positive plasma membrane, mitochondrial and other organelle fragmentation, reactive oxygen species (ROS) production, aberrant calcium flux, and the activation of molecular markers of cell death. The cell revival phase involves the biogenesis of all the cell organelles and also a reversal of all other cellular changes observed during the death phase. We observed the formation of a novel cellular compartment during revival, where highly polarized mitochondria tightly encircle large functional multivesicular bodies. The gene expression analysis suggests that the revival process is a meticulously programmed event, where multiple signaling and metabolic pathways required for survival and growth are induced in a well-synchronized manner. Interestingly, the program initiates the activation of pathways essential for embryonic development, growth, and differentiation. Additionally, chromatin remodeling enzymes and circadian rhythm genes were activated when cells initiated a new phase of life. This is followed by a stepwise upregulation of genes related to inflammatory and immune responses, metabolism, membrane trafficking, ion transport, extracellular matrix organization, and other pathways necessary for the full resuscitation of cells. We found that these massive transcriptome changes were not brought about by genome-wide alterations in methylation patterns, but by enhanced chromatin accessibility. Inhibitor screens and gene knockdown experiments suggest that multiple signaling mechanisms work in a concerted manner for cell resuscitation. Nevertheless, the NF-κB pathway was mainly found to be indispensable for cell revival.

Given that the pathways triggered by LLOMe in a cell culture model recapitulate those necessary for effective embryonic development, tissue repair, regeneration, and stemness, we performed experiments to assess the effectiveness of LLOMe in inducing these processes in various animal models. Our findings revealed that LLOMe significantly improved skin and corneal wound healing in a mouse model, facilitated tadpole tail regeneration in a frog model, promoted axon regeneration in the *Caenorhabditis elegans* model, and enhanced hematopoietic stem and progenitor cells in the lymph gland of *Drosophila melanogaster*. Taken together, this study opens up an entirely new avenue for exploration in cell biology and regenerative medicine.

# Results

## Cells revive from an early cell death state

LLOMe at concentrations of ~2–8 mM causes lysosome membrane rupture in various cell types, resulting in floating cells that are considered dead in the literature (Droga-Mazovec et al, 2008; Eriksson et al, 2020; Kavcic et al, 2020). We observed that when mouse embryonic fibroblasts (MEFs) were treated with LLOMe, they underwent a series of morphological changes (Figs. 1A,B and EV1A; Movies EV1–3). The cells displayed a rounded morphology within 5 min of treatment (C2 stage). After ~30 min of treatment, almost all the cells detached from the surface and exhibited severe apoptotic blebbing (C3 stage). To our surprise, a large portion of these cells (80–90%) reattached to the surface within 2–3 h (C4 stage). By 6 h, they started to regain their typical morphology (C5 stage). After 16 h of treatment, the cells appeared normal (C6 stage) and resumed their normal division process (Fig. 1B,C; Movies EV1–3). Throughout this entire experiment, the cell media was not changed. Initially, the revived cells exhibited a high degree of vacuolation (6–16 h), but these vacuoles disappeared by 24 h, resulting in morphologically normal cells (Movies EV1–3).

To ensure that the trajectory of cell death is reversed, we collected LLOMe (4 mM) treated floating cells at the C3 stage, washed them with PBS, resuspended them in fresh media (without LLOMe), and replated. Within 6 h, all the cells were attached to the surface and started gaining morphology, and by 24 h, almost all cells regained typical fibroblast morphology and were dividing normally (Fig. EV1B,C; Movie EV4).

Next, we investigated whether this phenomenon is specific to LLOMe treatment or is shared by other lysosome-disrupting agents. MEFs were exposed to various LMP-inducing agents such as alum (Hornung et al, 2008), crystalline silica (Hornung et al, 2008), siramesine (Ostenfeld et al, 2008), GPN (glycyl-L-phenyla-lanine 2-naphthylamide) (Berg et al, 1994) and sphingosine (Hoglinger et al, 2015; Villamil Giraldo et al, 2014) at different concentrations. Even at high concentrations of alum and silica, cells did not detach from the surface for an extended period (Fig. EV1D). Cells treated with GPN, siramesine, LLOMe, and sphingosine exhibited rounding, detachment from the surface, and membrane blebbing (Fig. EV1D). The cells treated with LLOMe, GPN, and sphingosine could revive (Figs. 1D,E and EV1D; Movies EV5, 6), while those treated with siramesine did not (Fig. EV1D). In a separate experiment, floating cells (C3 stage cells) were collected from all these treatments, washed, and re-plated. While cells treated with LLOMe, GPN, and sphingosine recovered normally, those treated with siramesine did not (Fig. 1F). The common feature among these three agents (LLOMe, GPN, and sphingosine), is that they all accumulate in lysosomes and cause membrane damage, resulting in $Ca^{2+}$ release (Hoglinger et al, 2015; Morgan and Galione, 2021; Morgan et al, 2020). For all further experiments, we proceeded with LLOMe.

Revival from LLOMe-induced early apoptotic conditions was observed in multiple cell types including BHK-21 (golden hamster

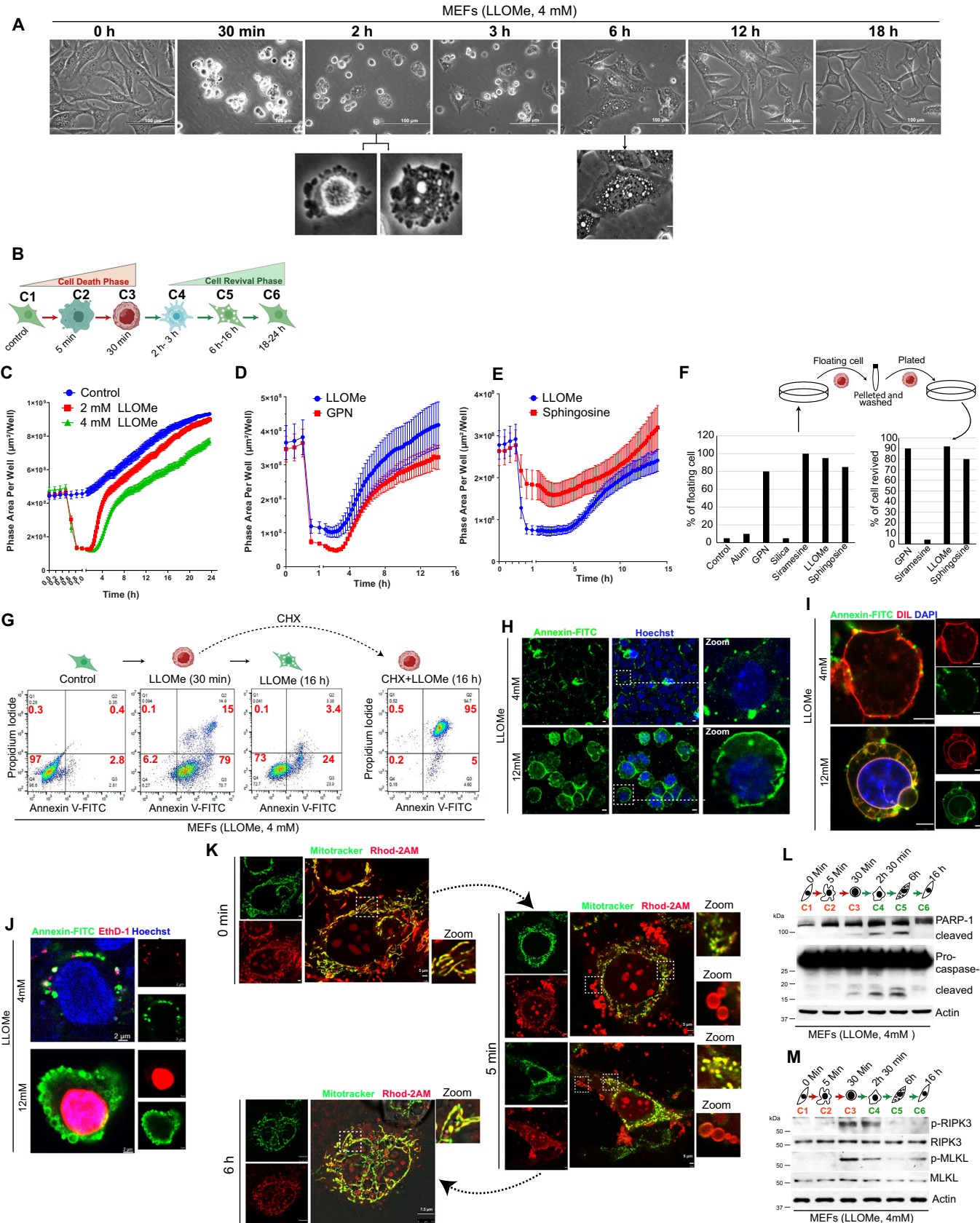

**Figure 1. Cells revive from apoptosis-like conditions.**

(**A**) Snapshots of time-lapse live microscopy images of MEF cells treated with 4 mM LLOMe. Magnification 40X. Scale Bar, 100 μm. Lower images are digital magnification. Refer to Movie EV1. (**B**) Upon treatment of cells with sublethal LLOMe concentrations, the cells revive back from apoptosis-like conditions. The process is divided into 6 phases (C1–C6 stages) based on observed morphological changes. The schematic depicts cell morphology at each stage. (**C**) The graphs depict quantification of phase area per well (area of well covered by cell) from time-lapse live microscopy of MEF cells treated with LLOMe (2 and 4 mM) (refer to Movie EV2). (**D**) LLOMe (4 mM) or GPN (200 μM) (refer to Movie EV5). (**E**) LLOMe (4 mM) or sphingosine (15 μM) (refer to Movie EV6), performed using IncuCyte S3. Mean ± SD, nine fields. The drop in the line represents an increase in floating cells and a reduction in confluency (phase area) in wells. (**F**) MEF cells were treated with different lysosomotrophic agents, and floating cells were collected and washed before replating in another dish. Graphs represent the percentage of floating cells (left panel) and the percentage of cells that revived (right panel) upon replating from a representative experiment. (**G**) Representative flow cytometry analysis of MEF cells treated with 4 mM LLOMe for 30 min or 16 h (with or without cycloheximide) and stained with Annexin V-FITC and Propidium Iodide (PI). (**H–J**) Snapshots of representative live cell confocal microscopy of MEF cells treated with LLOMe (4 or 12 mM) for 30 min and stained with (**H**) Annexin V-FITC (green) and hoechst (blue), Scale Bar, 5 μm, zoom images are digital magnification (**I**) Annexin V-FITC (green), DIL (red) and DAPI (Blue), Scale Bar, 5 μm (**J**) Annexin V-FITC (green), ethidium homodimer-1 (EthD-1) (red), and hoechst (blue), Scale Bar, 2 μm. Zoom panels are digitally magnified images. (**K**) Snapshots of representative live cell confocal microscopy of MEF cells that were stained with Rhod-2 AM (calcium indicator, red) and Mitotracker (mitochondrial marker, green) and treated with 4 mM LLOMe at the indicated time points (0 min, 5 min, and 6 h). Scale Bar, 5 or 7.5 μm as indicated. Zoom panels are digitally magnified images. The 0 min image is part of the EV2K control image. (**L, M**) Western blot analysis with the lysate of MEF cells treated with 4 mM LLOMe at different time points, with indicated (**L**) apoptosis or (**M**) necroptosis marker antibodies. Source data are available online for this figure.

kidney fibroblast cells), NIH/3T3 (mouse embryonic fibroblasts), MDA-MB-231 (Breast cancer epithelial cell line), HEK293T (kidney epithelial cells), and HeLa (cervical cancer epithelial cells) but not in, UM-UC3 (bladder cancer), HT-29 (colon cancer), and A549 cells (lung cancer) (Fig. EV1E,F). The primary bone marrow-derived macrophages (BMDM) failed to revive at higher concentrations (2 mM and above). At lower concentrations, the morphological features of revival were not similar to those of dividing cells (Fig. EV1G), although they shrank in size and also showed vacuolation. On the other hand, the primary MEFs and adult primary cardiac fibroblasts isolated from mice exhibited typical morphological changes as observed in other cell lines when treated with LLOMe (Fig. EV2A,B). However, the timings of floating and revival were different (Fig. EV2A,B). Taken together, the data suggest that many types of cell lines and non-immune primary cells could recover from the edge of cell death induced by LLOMe.

## Resurrection from plasma membrane damage

Due to lysosome membrane permeabilization, LLOMe-treated cells lose the lysotracker staining within 5–10 min (C2 stage) of treatment (Fig. EV2C; Movie EV7). However, once they started reviving, the same cells regained normal lysosomes and lysotracker staining (Movie EV7). This data suggests that LLOMe at 4 mM concentrations severely damages lysosomes. In FACS analysis, most (~80%) of the C3 stage floating cells were Annexin V positive (early apoptotic), whereas most (~73%) of the revived cells at the C5 stage showed no Annexin V staining (Fig. 1G), suggesting that cells could revive from the early apoptotic phase and lysosome damage. In these experiments, if protein translation was inhibited with cycloheximide at the C3 stage, the cells never recovered and exhibited Annexin V/propidium iodide (PI) double positivity, indicating they were in a late apoptotic stage and dead (Fourth panel, Fig. 1G).

In confocal imaging of the live cell, the 4 mM LLOMe-treated C3 stage floating cells displayed distinct and scattered Annexin V-FITC-stained plasma membrane, indicating phosphatidylserine (PS) flipping to the outer leaflet of the membrane (Upper panels, Figs. 1H and EV2D). These cells had a permeable nucleus to the live cell dye, Hoechst (Upper panels, Fig. 1H), but were impermeable to dead cell dyes such as DAPI (Upper panels, Figs. 1I and EV2D), ethidium homodimer-1 (EthD-1) (Upper panels, Figs. 1J

and EV2E,F; Movie EV8), PI (Upper panels, Fig. EV2G), and SYTOX (Fig. EV2H; Movie EV9). A mild staining of EthD-1, PI, or SYTOX was observed in the plasma membrane and cytoplasm but not in the nucleus (Upper panels, Figs. 1J and EV2E,G; Movies EV7–9), indicating that 4 mM LLOMe disrupts the plasma membrane but not the nuclear membrane.

MEF cells treated with 8 mM of LLOMe could also recover from severe apoptosis-like conditions, but no recovery was seen when cells were treated with 12 mM of LLOMe (Fig. EV2I; Movie EV10). The 12 mM LLOMe-treated floating cells (30 min) were in the early and late apoptotic stages (Annexin/PI double positive), whereas 16 h post treatment, most cells were either in the late apoptotic or necrotic/necroptotic stages (only PI positive) (Fig. EV2J). The 12 mM LLOMe concentration led to extensive PS flipping, resulting in strong Annexin V-FITC staining of plasma membrane and nuclear staining with PI, DAPI, and ethidium homodimer-1 in live cell imaging (Lower panels, Figs. 1H–J and EV2D–G; Movie EV8), indicating significant nuclear membrane damage. As anticipated, trypsinized floating cells exhibited no Annexin V-FITC and nuclear PI staining (Fig. EV2G).

In conclusion, cells can recover from early apoptosis even with a compromised plasma membrane; however, once the nuclear membrane is compromised, there is no turning back from cell death.

## Revival from hallmarks of cell death

Hallmarks of apoptosis initiation and lysosomal-dependent cell death include mitochondria fragmentation (or fission) (Arnoult, 2007) and vesicular $Ca^{2+}$ release into the cytosol, leading to PS externalization (Boitier et al, 1999; Milani et al, 2023; Mirnikjoo et al, 2009). Upon treatment with LLOMe (4 mM), mitochondrial fragmentation and a significant number of $Ca^{2+}$ positive vesicles (Rhod-2 AM stained) were observed in the live cells within 5 min (Figs. 1K and EV2K; Movie EV11). However, both the mitochondrial fragmentation and the number of $Ca^{2+}$ vesicles returned to normal levels in revived cells (6 h) (Fig. 1K). While cytosolic $Ca^{2+}$ and mitochondrial fragmentation are recognized as hallmarks of apoptosis, they are also believed to play a crucial role in promoting cell survival and are necessary for the repair of damaged plasma membrane (Andrews and Corrotte, 2018; Barisch et al, 2023; Cheng et al, 2015; Horn et al, 2020; Idone et al, 2008).

In Western blotting analysis with 4 mM LLOMe-treated cells, a weak cleavage of apoptosis markers (PARP-1 and caspase-3), a very weak cleavage of pyroptosis marker GSDMD (but not of Caspase-1), and a mild induction of necroptosis markers (p-RIPK3 and p-MLKL) was detected in C3, C4, and C5 stages (Figs. 1L,M and EV2L,M). In the fully revived cells (C6 stage), these cell death markers were reverted to normal (Figs. 1L,M and EV2L,M).

Next, we employed a ZipGFP-based caspase reporter, which constitutively expresses mCherry, and GFP is only produced upon caspase activation (To et al, 2016). This reporter allowed us to observe weak caspase activation (GFP) in many transfected HEK293T cells, which revived upon treatment with sublethal concentrations of LLOMe (Movie EV12). Whereas there was no caspase activity in control (ZipGFP-transfected) cells (Movie EV12). This data is consistent with our Western blot data. Taken together, the data indicate that cells can revive from a low level of caspase activation.

Collectively, the data indicate that cells have the potential to resuscitate despite experiencing significant lysosomal disruption, mitochondrial fragmentation, phosphatidylserine exposure, and mild activation of proteins involved in cell death execution.

We observed a weak activation of the caspase during the revival phase. We tested whether caspase activation directly contributes to the revival process by using pan-caspase inhibitors (zVAD). We used a range of zVAD concentrations (20, 40, and 80 μM), which were suggested in the literature to be effective in MEFs. We found that lower concentrations of zVAD (20 and 40 μM) did not affect the recovery of MEF cells from the death phase (Fig. EV2N). However, a higher concentration of zVAD (80 μM) impaired revival, resulting in delayed recovery and a final revival of ~50–60% (Fig. EV2N). A similar result was obtained when the LLOMe-treated floating cells were collected and replated in a fresh culture plate along with zVAD (Appendix Fig. S1). These findings suggest that caspase activation may play a role, though not a crucial one, in regulating the revival process.

## Organelle dynamics during revival from cell death-like conditions

Next, we were interested in understanding how different cell organelles respond to this extensive lysosomal damage, leading to apoptosis-like conditions and the status of various organelles in the reviving cells (Fig. 2A). We investigated the status of microtubules (α-Tubulin), mitochondria (TOM20), Golgi complex (RCAS1), endoplasmic reticulum (ERp72), lysosome (lysotracker), early endosome (EEA1), late endosome (Rab7), and autophagosome (LC3B) at 6-time points spanning the phenomenon using confocal microscopy in fixed cells (Fig. 2A). The C3 stage floating cells were attached to the microscopic slides using the cytospin method. We found that within 5 min of LLOMe (4 mM) treatment, microtubules, mitochondria, the Golgi apparatus, and the endoplasmic reticulum became highly fragmented (Figs. 2B and EV3A). To our surprise, within just 2–3 h of LLOMe treatment (C4 stage), the reattached cells started showing recovery of organelle morphology. By 16 h (C5-C6 stage), all these organelles regained their normal structure (Figs. 2B and EV3A,B).

## Reviving cells exhibit large acidic vacuoles enclosed by a network of mitochondria

The early endosomes labeled with EEA1 significantly increased in size (~2–2.2 μm compared to the usual 100–400 nm) during the C3 floating cell stage but returned to normal after 16 h of treatment (Fig. 2C). Similarly, during the initial recovery phase, we observed large autophagosomes (~4 μm) labeled with LC3 (Fig. EV3C,D). These autophagosomes may play a crucial role in meeting the increased catabolic demands of the recovered cells and could also contribute to the clearance of cellular waste.

The lysotracker-labeled lysosomal compartments and Rab7-labeled late endosomal compartments vanished shortly after LLOMe treatment (Fig. 2D). Intriguingly, after 2–3 h of recovery, cells regenerated abnormally large lysosome/late endosomes (~3–7 μm compared to the normal size of 100–500 nm) (Figs. 2D and EV3E,F), which were sustained up to 16 h post-recovery. The membranes of these large vesicles were positive for Rab7 (Figs. 2E and EV3E), while the lumen was marked by lysotracker and magic red (Cathepsin-B marker dye) (Figs. 2E,F and EV3G H), suggesting that these are enzymatically active acidic compartments. The live and fixed cell microscopy of these cells showed small acidic vesicles (positive for cathepsin-B) within these large acidic vacuoles (Figs. 2E,F and EV3G,H; Movie EV13), appearing like multi-vesicular bodies (MVBs). Interestingly, a network of mitochondria was observed surrounding these enlarged acidic compartments, and in live microscopy, a close association between these vesicles and mitochondria was evident (Figs. 2G and EV3I; Movie EV14). After ~24 h of treatment, these acidic compartments were found to be of normal size and number (Fig. EV3J). We envisage that these mitochondria-encircled acidic compartments serve the metabolic and energy demands of reviving cells.

Next, we determined the mitochondrial membrane potential using JC-1 dye to evaluate the health of the mitochondria across all stages. JC-1 dye displays membrane potential-dependent accumulation in mitochondria, emitting green fluorescence at low membrane potential, and at higher membrane potential, it forms aggregates and emits red. Upon LLOMe treatment, the mitochondria depolarized rapidly, which is also a hallmark of cell death (Fig. 2H). The floating cells (C3 stage) population has most cells that have depolarized mitochondria (~60% depolarized), but as soon as the cells touched back to the surface, the polarization of mitochondria increased (~96% polarized) steeply within the next 2 h. By 24 h, the potential was normalized to that of untreated cells (Fig. 2H,I). As polarized mitochondria are highly energy-efficient, this rapid and steep increase in polarization may help cells meet their energy demands during revival. We also measured reactive oxygen species (ROS) production using CellROX dye (Fig. EV3K,L). A rapid spike of ROS levels was observed in the floating cells concomitant with mitochondrial depolarization. Then ROS levels slowly returned to normal as the cells adhered to the surface and grew (Fig. EV3K,L).

## Programmed cell revival

To comprehend the intricacies of the revival process, we performed RNA-sequencing analysis from cells taken at 6 stages ($n = 3$) of the revival process (Fig. 3A). The principal component analysis (PCA) revealed that the transcriptome of stages C1 to C3 (cell death phase) were closely grouped, indicating minimal variations in gene regulation during this phase (Fig. 3B). On the other hand, the transcriptome of reviving cells (stages C4 to C6) was distinct from the transcriptome of the death phase. Interestingly, even within the reviving stages, the gene expression pattern of C4 stage cells was

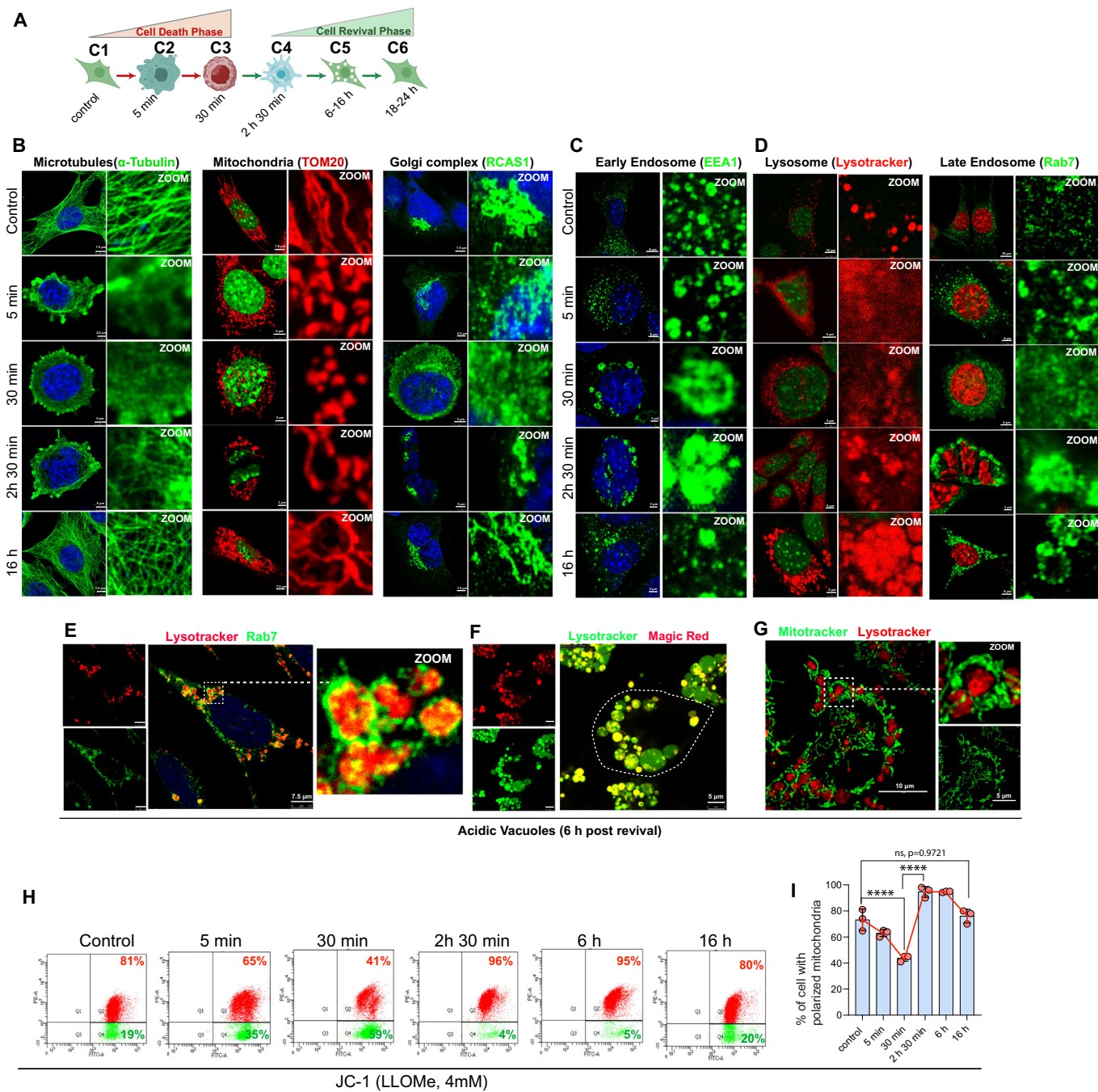

**Figure 2. Organelle dynamics during cell death and the resuscitation phase.**

(A) Schematic shows the stages at which confocal microscopy is performed to understand changes in cell organelle states and morphology. (B–D) Representative confocal microscopy images of MEF cells treated with 4 mM LLOMe for the indicated time points and immunostained with (B) α-tubulin (microtubules) or TOM20 (mitochondria) or RCAS1 (Golgi complex), (C) EEA1 (early endosomes), (D) Rab7 (late endosomes) and lysosomes are stained with lysotracker red. The nucleus is stained with DAPI (pseudo-colored to red or green in some panels for better contrast). Zoom panels show digital magnifications. Scale Bar, 2.5 to 7.5 μm as described in the figures. (E–G) Representative confocal microscopy images of MEF cells treated with 4 mM LLOMe for 6 h and cells were stained with (E) lysotracker red, followed by fixation and immunostaining with anti-Rab7 (green), (F) Magic Red (cathepsin-B activity) and lysotracker green, (G) Mitotracker green and lysotracker red. Zoom panels are digital magnifications. (H, I) Left panels, flow cytometry analysis of mitochondrial membrane potential using JC-1 dye in MEF cells treated with 4 mM LLOMe at the indicated time points. Right panel, the bar graph shows the quantification of JC-1 data at the indicated time points ($n = 3$, mean ± SD, ****$p < 0.0001$, #not significant (ns) $p = 0.9721$, ordinary one-way ANOVA with Tukey's multiple comparison test). Source data are available online for this figure.

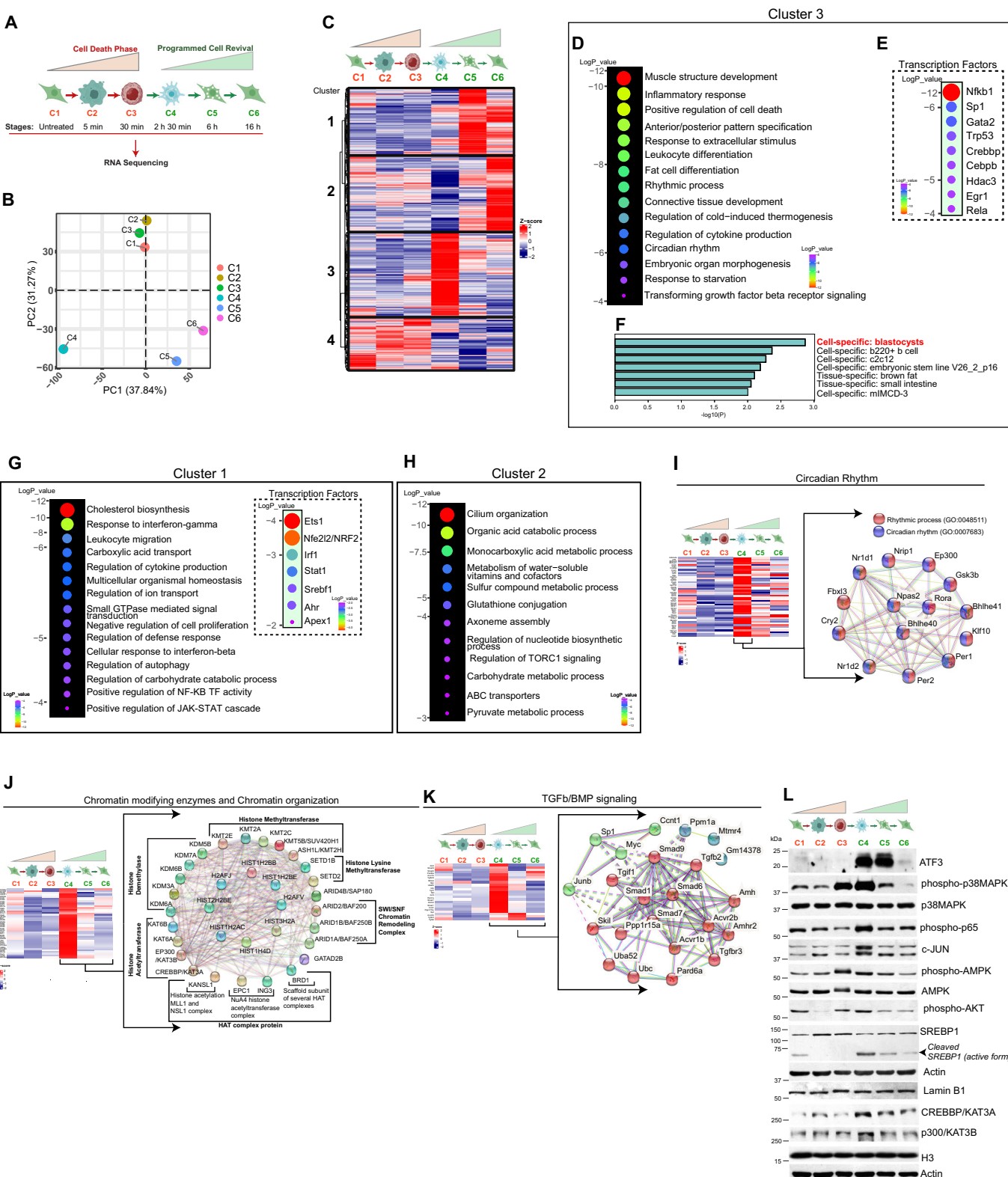

**Figure 3.  Transcriptome analysis of programmed cell revival.**

(A) RNA sequencing experiment ($n = 3$ biological replicates) is performed with MEF cells treated with 4 mM LLOMe at the indicated six-time points (C1–C6) with three biological replicates. (B) Principal component analysis (PCA) of RNA sequencing data ($p < 0.05$, $n = 3$) showing the transcriptional changes in MEF cells at different stages of LLOMe treatment. (C) Hierarchical clustering heatmap of differentially expressed genes ($p < 0.05$, >1.5 folds, ≤0.6 folds, Wald chi-squared test, $n = 3$) across all six time points (C1–C6). The z-scores represent the relative expression levels. (D, E) The cluster 3 genes from panel C were subjected to (D) Metascape pathway analysis or (E) Metascape TRRUST analysis for transcription factors. The top pathways (metascape) or top transcription factors (TRRUST) were depicted by a bubble plot. The size of each bubble represents the significance levels (log $p$ value). (F) The cluster 3 genes were subjected to enrichment analysis in PaGenBase using the Metascape analysis tool. The graph represents the top gene signature of cell or tissue type obtained from cluster 3 genes. (G, H) The cluster 1 or cluster 2 genes were subjected to Metascape pathway analysis (left panel) or TRRUST analysis (right panel) and the top pathways or top transcription factors were depicted by a bubble plot. The size of each bubble represents the significance (log $p$ value). (I–K) Left panel, heatmap generated for genes representing GO terms (I) circadian rhythm, (J) Chromatin-modifying enzymes and chromatin organization, (K) TGFb/BMP signaling, induced ($p < 0.05$, >1.5 folds, ≤0.6 folds, base mean >10, Wald chi-squared test, $n = 3$) in C4 to C6 stages compared to C3 stage. Right panel, STRING analysis of core genes of the pathway. (L) Western blot analysis with the lysate of MEF cells treated with 4 mM LLOMe for indicated time points with indicated antibodies. Source data are available online for this figure.

notably different from that of C5 and C6 stage cells, suggesting diverse changes in the transcriptome during the revival process. Indeed, the hierarchical clustering of transcriptome data shows that the gene expression perturbations were dramatic when the cells revived (C4 to C6 stages) (Fig. 3C).

Cluster 3, the largest cluster consisting of approximately 2300 genes, primarily shows upregulation during the C4 stage (Fig. 3C; Dataset EV1), which is the onset of the revival process. Cluster 1 genes are predominantly induced during the C5 stage (Fig. 3C; Dataset EV1). Cluster 2 genes have low expression in the C4 stage, induced in the C5 stage, and highly upregulated in the C6 stage (Fig. 3C; Dataset EV1). Notably, a distinct cluster of genes is induced temporally during the C4, C5, and C6 stages, suggesting that cell revival from early apoptosis is a precisely regulated event, and thus, we named this phase programmed cell revival. Cluster 4 encompasses genes expressed from the C1 to C3 stages but downregulated during the revival stages (C4 to C6) (Fig. 3C; Dataset EV1).

## Activation of embryonic developmental pathways at the onset of programmed cell revival

We performed gene ontology (GO) enrichment analysis (GEA) using Metascape (Zhou et al, 2019) tools to identify signaling pathways and molecular processes overrepresented in the three gene clusters (1 to 3) (Fig. 3D–F).

The activation of the cluster 3 gene represents the first step of programmed cell revival. The Metascape analysis of cluster 3 showed multiple pathways related to embryonic development (muscle structure, connective tissue, blood vessels, and reproductive structure), differentiation (muscles, lymphocyte, myeloid, and fat cell), organ morphogenesis, anterior-posterior pattern specification, inflammatory/immune response (cytokine production and adaptive immunity) and circadian rhythm were induced (Fig. 3D; Dataset EV2). The top transcription factors identified by TRRUST software (Han et al, 2018) as potential regulators of the cluster 3 genes program include NF-κB, Sp1, GATA2, and CREBBP (Fig. 3E). NF-κB, Sp1, GATA2, and CREBBP transcription factors play essential roles in controlling genes crucial for embryonic development, proliferation, self-renewal, maintenance, and stem cell functionality (Espin-Palazon and Traver, 2016; Liu et al, 2022a; Marin et al, 1997; Tanaka et al, 2000; Tsai et al, 1994) (Fig. 3E). Notably, gene signature analysis using the PaGenBase (Pan et al, 2013) predicted top cell type related to cluster 3 is the "blastocyst",

a structure formed earliest in the embryonic development of mammals (Fig. 3F). Taken together, the data indicate that the reviving cells launch a molecular program mimicking embryonic cells for reinitiating a new life.

Cluster 1 genes (C5 stage) and cluster 2 genes (C6 stage) analysis showed the overrepresentation of pathways related to cholesterol, steroid, lipids, fatty acids, carbohydrates, nucleotides, vitamin metabolism, and ATP generation (Fig. 3G,H; Dataset EV2). Further, in alignment with heightened cellular trafficking and organelle dynamics during this phase of rejuvenation (Fig. 2), multiple genes associated with ion transportation, amino acid transportation, membrane trafficking, and autophagy were stimulated, likely to facilitate diverse biochemical, enzymatic, and metabolic requirements of biosynthetic pathways (Fig. 3G,H; Dataset EV2). This data suggests that the cells' intense metabolic activity at this stage supports the biogenesis of organelles and other metabolic processes in the reviving cells. The inflammatory responses, specifically the interferon response, were maintained at a high level in these stage cells. The pathways related to cell cycle, cell morphology, extracellular matrix organization, chemotaxis, etc., were also induced at this stage to prepare cells for cell division and migration.

Taken together, the transcriptomic data indicate that resuscitation from cell death is a highly regulated, well-programmed process.

## Molecular pathways and gene networks of programmed cell revival

Next, we performed pathway analysis to understand stepwise transcriptome changes with genes induced ($p_{adj} < 0.05$, >1.5 fold) in C4, C5, and C6 revival stages compared to the C3 death phase. The C4 stage vs C3 stage-induced genes ($p_{adj} < 0.05$, >1.5 fold, Dataset EV3) showed similar pathways to cluster 3, where primarily embryonic developmental and differentiation pathways were upregulated (Fig. EV4A; Dataset EV3). In addition, inflammatory pathways (interferon response and TNFα), cholesterol/sterol biosynthesis, integrated stress response (ATF3/4, EIF2AK1, PERK, and HSF-1), and circadian rhythm pathways were significantly overrepresented. The Reactome molecular pathway analysis of the same gene set (C4 vs C3) showed that genes related to signaling pathways such as FOXO, RUNX1, WNT, NOTCH1, p53, NRF2, TGF-β, ATF4, NPAS4, mTORC, and NF-kB were upregulated during the onset of revival (Fig. EV4B; Dataset EV3). Several of

these signaling modules play critical roles in embryonic development, stem cell proliferation and maintenance, cell growth, cell differentiation, tissue repair, and regeneration (Choi et al, 2022; Friedman, 2009; Karra et al, 2015; Massague and Sheppard, 2023; Zhou et al, 2022). The C5 stage and C6 stage pathways were similar to cluster 1 and cluster 2, respectively, where multiple pathways related to metabolism (nucleotide, lipid, carbohydrate, protein), cellular transport, cellular trafficking, cell migration, inflammation, and cell proliferation were induced (Fig. EV4C,D; Dataset EV4).

Next, we attempted to understand whether the revival pathways genes were upregulated, concerted, or stepwise. To achieve this, we surveyed the transcription levels of genes in activated key pathways across all stages from C1 to C6 using heatmap analysis. We observed that a few pathway genes show concerted activation at the C4 stage. Noteworthy examples in this category included circadian rhythm genes (Fig. 3I), chromatin-modifying enzymes/genes (Fig. 3J), and TGFβ/BMP signaling (Fig. 3K). The chromatin-modifying enzymes include histone demethylases (*KDM3A, 5B, 6A/B, 7A*), histone lysine methyltransferases (*SET1B* and *SETD2*), SWI/SNF chromatin remodeling complex proteins (*BAF200, BAF250A,* and *BAF250B*), histone acetyltransferases (*KAT3A, KAT6A/B,* and *EP300*) and histone methyltransferases (*KMT2A/2C/2E/2H, KMT5B*) (Fig. 3J). The upregulation of circadian rhythm genes (*Per1, Per2, Cry2, Npas2, Bhlehe40*, etc.) and epigenetic genes/pathways at the beginning of revival may serve to reset the cellular clock and modify the epigenome to facilitate global transcriptome expression. Additionally, several genes related to cell cycle/proliferation pathways (*GSK-3β, Wee1, Myc, Fosb, E2f3*, etc.) were specifically induced during this stage, potentially preparing the cells for division (Fig. EV4E).

Many pathway genes were activated progressively during cell revival. Some examples of stepwise induction of pathways are depicted in heatmaps of genes related to WNT signaling (Fig. EV4F), NOTCH signaling (Fig. EV4G), cholesterol/lipid biosynthesis (Fig. EV4H), organelle biogenesis/maintenance (Fig. EV4I), axon guidance (Fig. EV4J), nucleotide/fatty acid metabolism (Fig. EV5A), ion/amino acid transport (Fig. EV5B), and vesicle transport and membrane trafficking (Fig. EV5C). The data indicate that a highly coordinated, complex, but programmed process is responsible for reviving the cells from a near-death phase.

We also examined the changes in the transcriptome as the cells entered the death phase (C1 to C2 and C3). In comparison to the C1 stage, ~136 genes were upregulated, and 157 genes were downregulated in the C2 stage ($p < 0.05$, >1.5 fold, <0.5 fold) (Datasets EV5 and EV6). About ~300 genes were upregulated and downregulated in stage C3 ($p < 0.05$, >1.5 fold, <0.5 fold) in comparison to the C1 stage (Datasets EV5 and EV6). Not many pathways with significant $p$ values were overrepresented during the death phase transition (Dataset EV5).

We noted that the gene expression patterns in the RNA-Seq at the C6 stage (16 h after LLOMe treatment) differed from those at the C1 stage (untreated cells), despite their morphological similarities. This discrepancy may indicate that the cells have not reached a basal state at the molecular level in the C6 stage. Consequently, we conducted another RNA-Seq experiment with untreated cells (C1 stage), cells treated with LLOMe for approximately 3 h (C4 stage), and fully revived cells treated with LLOMe for 30 h (extended-C6 stage) (see Fig. EV5D). We found

that most genes significantly ($p < 0.05$, >1.5 fold) activated at the C4 stage (Dataset EV7) returned to basal levels in the extended-C6 stage (Fig. EV5E).

We found that primary BMDMs, when treated with LLOMe, do not exhibit the same morphological changes as non-immune dividing cells do. Next, we performed qRT-PCR using genes that represent major pathways upregulated during revival. We found that, except for one, none of these genes were induced during the revival process in BMDM; rather, their expression was reduced (Appendix Fig. S2). This data further demonstrates that immune cells do not undergo the programmed cell revival phenomenon.

Cells undergoing apoptosis induced by ethanol have been shown to recover when the ethanol is removed (Sun et al, 2017). While our study employs a distinct inducer of cell death, comparison of our RNA-seq data to that of Sun et al (Sun et al, 2017) revealed notable similarities. Re-analysis of the Sun et al dataset using Metascape revealed a striking concordance in the activation of developmental pathways and transcription factors regulating early revival genes (Appendix Fig. S3). Furthermore, both datasets demonstrated early responses involving genes related to chromatin modification, as well as pathways including membrane trafficking, MAPK signaling, WNT signaling, TGF-β signaling, p53 signaling, and organelle biogenesis. However, a significant difference was the presence of a cytokine response (specifically, IFN response) in our early response genes, which were not observed by Sun et al. Shared patterns were also observed in late-response genes and pathways, including metabolic processes and cell morphogenesis. Collectively, these findings suggest that the revival process following cell death may employ a conserved pathway but with certain differences.

## Activation of the master regulators of vital pathways during programmed cell revival

Next, we examined the protein levels of some of the master regulators of vital pathways (Fig. 3L). We found robust activation of ATF3 (global regulator of immune and metabolic pathways), phospho-p38 MAPK (global regulator of stress response, proliferation, differentiation, development, and inflammation. etc.), phospho-AKT (global regulator of cell survival and growth), c-JUN (a component of AP-1, regulator of the cell cycle, growth, and inflammation), phospho-AMPK (regulator of energy metabolism and homeostasis), NF-κB (p65 subunit, regulator of inflammation, immunity, differentiation, cell growth, and apoptosis), SREBP1 (regulator of lipid metabolism), p300/KAT3B, and CREBBP/KAT3A (chromatin modifiers, regulator of embryonic development, growth control, and homeostasis) at C4 and C5 stages of programmed cell revival (Fig. 3L). The data suggest that multiple vital signaling pathways are induced at the onset of revival.

## Genome-wide DNA methylation changes during programmed cell revival

We questioned what regulates massive transcriptional changes observed during PCR. DNA methylation is the epigenetic mechanism that drives transcriptional changes during embryonic development and multiple other physiological conditions. We profiled the distribution of 5-methylcytosine (5mC) and 5-hydroxymethylcytosine (5hmC) using Oxford Nanopore sequencing (Sigurpalsdottir et al, 2024) with DNA isolated from four different stages, C1, C3, C4, and extended-C6

(Appendix Fig. S4A). We first profiled the landscape of 5mC in three different sequence contexts: CpG, CHG, and CHH, where H indicates any base other than a G. There was an abundant 5mC in these contexts across all the stages (Appendix Fig. S4B). In contrast, 5hmC was generally absent, with the lowest amounts detected in the CpG context in line with previous reports (Jones, 2012). We did not observe any differences in 5mC or 5hmC among the samples (Appendix Fig. S4B,C). We also analyzed specific changes in the promoter region (2 kb upstream of each transcription start site) (Appendix Fig. S4D). There was no overall difference in promoter methylation across the time points. Taken together, these results suggest that DNA methylation may not be the primary factor governing the gene expression changes observed during programmed cell revival.

## Genome-wide changes in chromatin accessibility during programmed cell revival

Chromatin accessibility allows the selective interaction of regulatory proteins/complexes to modulate the transcription of genes (Li et al, 2023; Thurman et al, 2012). The opening of chromatin is correlated with enhanced gene transcription (Li et al, 2023; Thurman et al, 2012). To explore potential mechanisms controlling the transcriptional dynamics across the cell death phase and programmed cell revival, we mapped the chromatin accessibility at the C1, C3 (30 min LLOMe treatment), C4 (3 h LLOMe treatment), and extended-C6 (30 h LLOMe treatment) stages by ATAC-seq (Fig. 4A). PCA analysis shows that the genome-wide chromatin accessibility profile (or differentially accessible peaks) is similar for control (C1) and fully revived (extended-C6) cells but very distinct for the C4 stage (onset of revival) cells (Fig. 4B). Next, we compared differential accessible peaks (DAPs) between the stages of PCR. Compared to the control (C1) or the floating stage cells (C3), C4 stage DNA showed >2500 significant DAPs (Fig. 4C). Among these, at ~2000 loci, there is increased chromatin accessibility at the onset of cell revival (C4 stage) (Fig. 4C). This justifies the massive transcriptional changes observed during the C4 stage in transcriptome analysis. Very minimal changes in DAPs (~250) were observed at the floating cells stage (C3) when compared to the control (C1) (Fig. 4C) Aligned with PCA analysis, no DAPs were observed in fully revived (C6) cells compared to the controls, suggesting that the cell revived fully back to its normal state (Fig. 4C). Our data suggest that chromatin accessibility was significantly increased as the cells started the revival program. The ATAC-seq peaks at loci with increased accessibility at C4 vs C1 ($p_{adj} < 0.01$, log2 foldchange >2) were compared across different stages. Comparisons indicate that a unique set of loci became accessible during the onset of revival (C4 stage) (Fig. EV5F).

Using Homer software (http://homer.ucsd.edu/homer/), we annotated the genes present on either side of the DAPs region and performed pathways analysis (Metascape and Reactome) with genes that showed significantly higher accessibility ($p_{adj} < 0.01$, log2 foldchange >2) in the C4 stage (revived cells) compared to the C3 stage (death phase cells). Interestingly, the pathways that were overrepresented in this analysis (Fig. 4D,E; Dataset EV8) were strikingly similar to transcriptome analysis. This included developmental (muscles, blood, neurons, etc) and morphogenesis pathways and pathways related to metabolism, transport, trafficking, chromatin modification, and inflammation (Fig. 4D,E; Dataset EV8). Also, in striking similarity to transcriptome analysis,

TRRUST analysis (Han et al, 2018) indicated NF-κB as the top transcription factor regulating genes with increased chromatin accessibility at the C4 stage (Fig. 4D). We analyzed ATAC-peaks across all the stages (C1 to C6) for some of the representative developmental, metabolic, and pro-inflammatory genes (Atf3, Hoxb9, Rsad2, Insig1, and Ccl2) promoter and gene body. For all these genes, the chromatin accessibility peaks were induced at the C4 stage (onset of revival) and were reduced to normal at the C6 stage (fully revived) (Fig. EV5G).

Using Homer, we performed transcription factor-specific motif analysis on the significant peaks with increased accessibility at stage C4 compared to C3 ($p_{adj} < 0.01$, log2 fold-change >2). We identified multiple DNA sequence motifs known to bound by transcription factors (TFs) that potentially regulate the revival phase (C4 stage) (Fig. 4F; Appendix Table S1). This includes stress-induced master TFs such as ATFs (Atf1, Atf2, Atf3, Atf4, and Atf7) that play a significant role in the metabolic and immune homeostasis of cells. Other includes immediate early response TFs (Fos, Fra1, JunB, and c-Jun), pro-inflammatory TFs (Batf, NFkb-p65/p50, Irf1, Irf3, Irf8, STATs, PU.1, and Ets1), oxidative stress and metabolism responsive TFs (Mitf, Nrf2, Tfe3, Bach1, Chop, Foxo1, and Srebp1), circadian rhythm TFs (Clock, BMAL, and Npas2), and embryonic development and stemness TFs (Gata3, cMyc, Klf1, Klf4, RunX2, HoxA3, etc) (Fig. 4F; Appendix Table S1).

Among transcription factors, we observed that NF-κB-p65-regulated genes were specifically induced at the C4 stage. We compared and plotted the ATAC-seq signals at p65 (RelA) ChIP-seq peaks (Ngo et al, 2020) at different stages (C1 to C6) of LLOMe treatment (Fig. 4G). Enhanced chromatin accessibility was observed at p65 ChIP loci at the C4 stage compared to the C1 or C3 stage (Fig. 4G), indicating that, indeed, NF-κB-p65 directly interacts with chromatin to enhance gene expression at the onset of the resuscitation phase.

Taken together, our results indicate that chromatin accessibility (epigenetic changes) leading to transcription factor binding and dynamic regulation of genes governs the programmed cell revival.

## Calcium signaling and mitochondrial functions are critical for programmed cell revival

During the revival process, we observed dynamic changes in cell organelles, particularly in acidic organelles and mitochondria. Additionally, we noticed an increase in membrane and vesicular $Ca^{2+}$ trafficking. To determine which of these cellular processes are crucial for cell revival, we treated cells with specific inhibitors and evaluated their revival using high-content screening assays (Fig. 5A,B). The inhibitors used in screening include dynasore hydrate (an endocytosis inhibitor), 3-methyladenine (a PI3K/autophagy inhibitor), BAPTA-AM (an intracellular calcium chelator), cytochalasin D (an actin polymerization inhibitor), nocodazole (a microtubule polymerization inhibitor), chloroquine/bafilomycin A1 (a lysosome acidification inhibitor), N-acetyl cysteine (NAC, an antioxidant/ROS), MitoTEMPO (a mitochondrial superoxide antioxidant), Mdivi (a mitochondrial division inhibitor), and carbonyl cyanide chlorophenylhydrazone (CCCP, a mitochondrial depolarization agent). Except for MitoTempo and NAC, other inhibitors considerably reduced the revival of cells (Fig. 5A,B). BAPTA-AM and CCCP treatment completely prevented revival, suggesting a pivotal role of $Ca^{2+}$

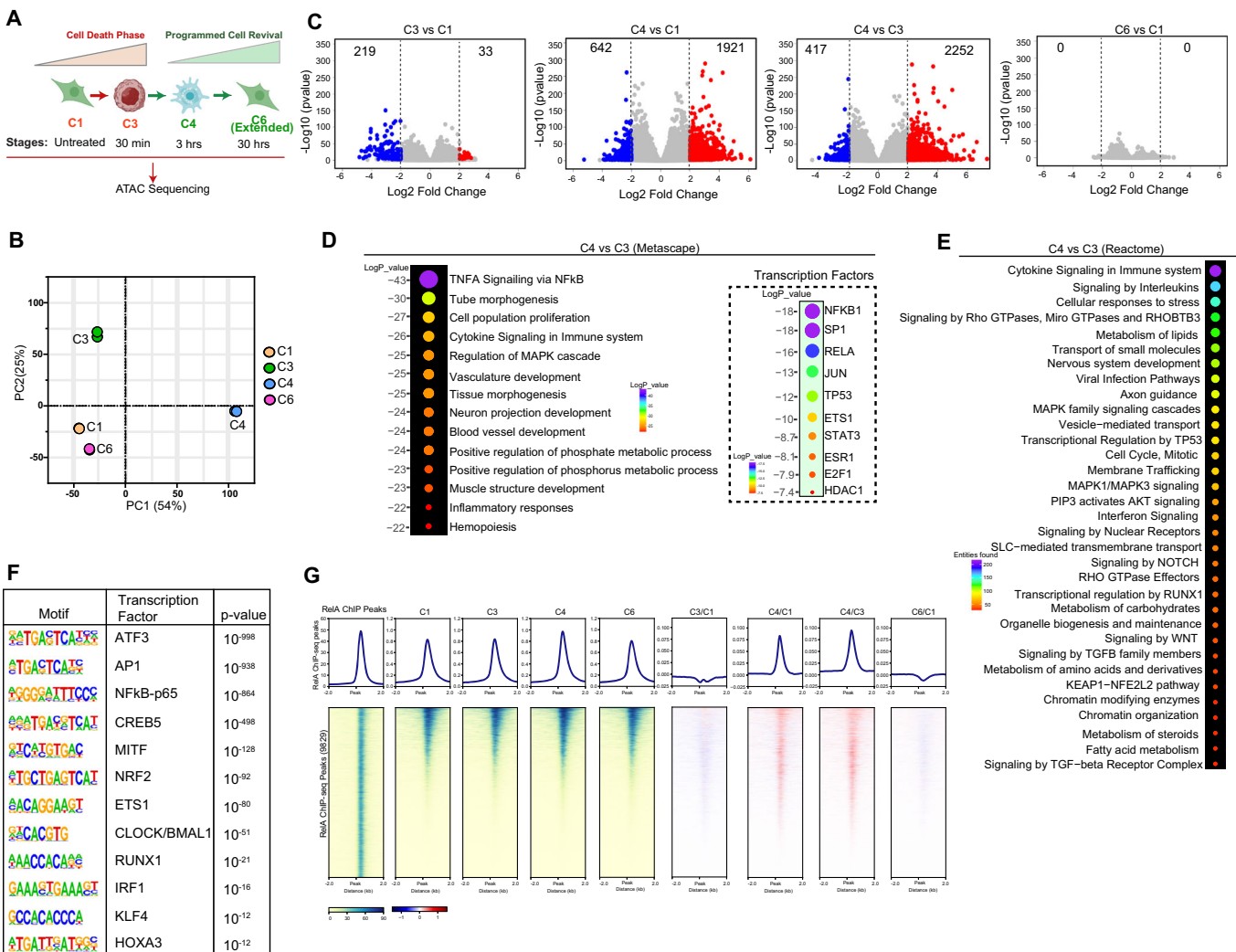

**Figure 4. The ATAC-seq analysis during cell death and resuscitation phase.**

(A) ATAC sequencing experiment is performed with MEF cells treated with 4 mM LLOMe at the indicated four time points (C1, C3, C4, C6) with two biological replicates. (B) Correlation plot showing the principal component analysis (PCA) of ATAC sequencing data of all differential accessible peaks in MEF cells at different stages of LLOMe treatment (C1, C3, C4, and C6). (C) Volcano plot depicting the change in DNA accessibility between untreated and 30 min post treatment (C3 vs C1); untreated and 3 h post treatment (C4 vs C1); 30 min post treatment and 3 h post treatment (C4 vs C3); untreated and 30 h post treatment (C6 vs C1). (Points indicated are the peaks with differential accessibility, red points indicate the loci with an increase in accessibility ($p_{adj} < 0.01$, $\log_2$foldchange >2, $n = 2$) and blue points indicate the loci with an increase in accessibility ($p_{adj} < 0.01$, $\log_2$foldchange <−2, $n = 2$). (Numbers indicated are the differential peaks with a cut-off) ($p$ value: Wald test; adj $p$ value: Benjamini–Hochberg test). (D, E) The peaks with an increase in accessibility at C4 compared to C3 ($p_{adj} < 0.01$, $\log_2$foldchange >2) ($p$ value: Wald test; adj $p$ value: Benjamini–Hochberg test) were subjected to (D) Metascape pathway analysis and TRRUST analysis, and the top pathways or top transcription factors were depicted by a bubble plot. The size of each bubble represents the significance (log $p$ value) and (E) Reactome pathway analysis; the top pathways or top transcription factors were depicted by a bubble plot. (F) Homer motif analysis was performed on the significant peaks with an increase in accessibility at stage C4 compared to C3. The top identified known motifs of the transcription factors were represented along with the significance ($p$ value).($p_{adj} < 0.01$, $\log_2$foldchange <−2, $n = 2$). ($p$ value: Wald test; adj $p$ value: Benjamini–Hochberg test). Full list of TFs is shown in Appendix Table S1. (G) The accessibility of RelA (p65) ChIP sequencing peaks (GSE132792_RelA_peaks.bed.gz) were compared and plotted at different stages of LLOMe treatment to MEF cells (C1, C3, C4, and C6). The ATAC-seq signal at p65 ChIP-seq peaks in C1, C3, C3/C1, C4, C4/C1, C4/C3, C6, and C6/C1, are plotted as heatmaps with RelA ChIP-seq peaks (GSM3892239) in the first row. (The heatmaps represent a 4 kb window, centered on the peak midpoint and sorted based on the C4 compared to RelA ChIP-seq peaks). On top are the metaplots representing average profiles of ATAC-seq signal at RelA ChIP-seq peaks in the respective condition.

signaling and mitochondrial function in revival (Fig. 5A–C; Movie EV15). Unexpectedly, autophagy inhibitors and lysosome acidification inhibitors have a moderate effect, where about 20-50% of cells were able to revive back normally (Fig. 5A,B). Our results suggest that multiple pathways contribute to the revival of cells, where $Ca^{2+}$ signaling and mitochondrial function appear to be critical for recovery.

We found that autophagosome numbers were dramatically increased (Fig. EV3C) and multiple autophagy genes were upregulated (Appendix Fig. S5A) during revival. Also, at the protein level, LC3B-II accumulation is increased during cell revival (Appendix Fig. S5B), suggesting autophagy is induced during the revival process. We tested whether autophagy is important for

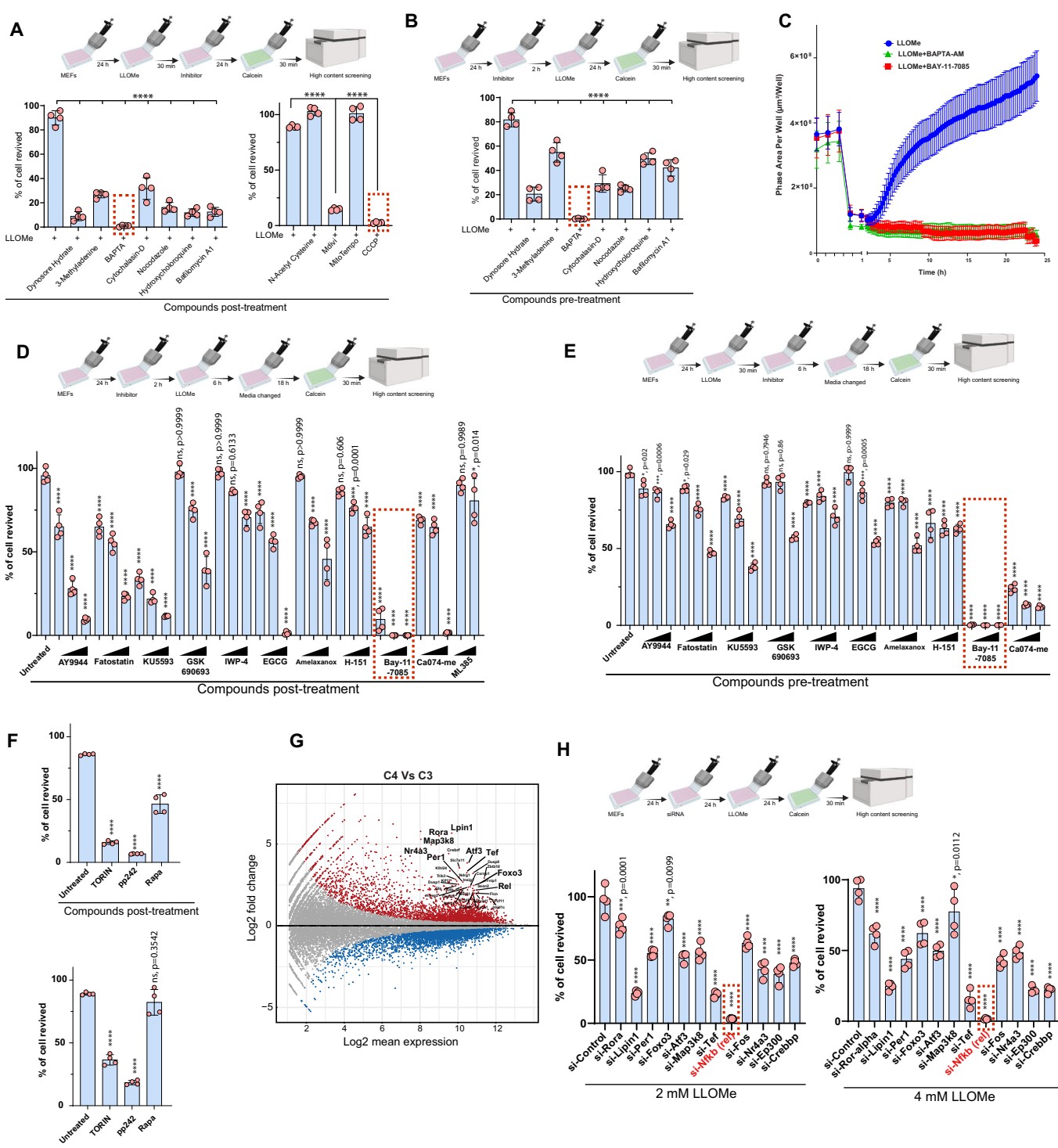

revival by using MEF knockout for ULK1 and ATG5, the proteins essential for autophagosome biogenesis. While ULK1^KO showed a slight decrease in live cell count (Appendix Fig. S5C,D), there was no significant defect in cell resuscitation in ATG5^KO MEF cells (Appendix Fig. S5E,F). These findings, combined with inhibitor experiments, suggest that canonical autophagy may not play a major role in the cell revival process.

## The NF-κB is indispensable for programmed cell revival

To test the role of key signaling pathways in the revival process, we inhibited them using increasing concentrations of specific inhibitors added either before LLOMe treatment (Fig. 5D,F) or after LLOMe treatment (Fig. 5E,F). This includes inhibition of cholesterol biosynthesis (AY 9944), lipid biosynthesis (Fatostatin),

**Figure 5. The pathways/genes critical for programmed cell revival.**

(A, B) Top panels, schematic representation of the experimental design. The MEFs were either treated with inhibitors (A) after 4 mM LLOMe exposure or (B) before 4 mM LLOMe exposure, followed by staining with live cell dye Calcein-AM and quantification was performed using high content screening. Bottom panel, Graph depicts the percentage of MEF cells revived (****$p < 0.0001$, mean ± SD, $n = 4$, 25 fields/well, one-way ANOVA, Dunnett's multiple comparison test). (C) Time-lapse live microscopy performed using IncuCyte S3 (refer to Movies EV1 13, 14) of MEF cells treated with BAPTA-AM (Calcium chelator) or BAY-11-7085 (NF-κB inhibitor) after 4 mM LLOMe exposure. The graph depicts quantification of phase (cell covered) area per well ($\mu m^2$/well), Mean ± SD, nine fields. The drop in line represents increased floating cells and reduced confluency (phase area) in wells. (D–F) Top panel, schematic representation of the experimental design where MEFs were either treated with inhibitors (D, F) after 4 mM LLOMe exposure or (E, F) before 4 mM LLOMe exposure, followed by staining with live cell dye Calcein-AM and quantification was performed using high-content screening microscopy. Bottom panel, the graph depicts the percentage of MEF cells revived. mean ± SD, *$p < 0.05$, ***$p < 0.001$, ****$p < 0.0001$, ns non-significant, $n = 4$, 25 fields/well, one-way ANOVA, Tukey's multiple comparison test. Exact $p$ values are depicted in the Figure and Appendix Table S2. (G) MA-plot analysis represents log2 fold-change versus log2 mean expression between the C3 stage and the C4 stage transcriptome. Each dot is a gene, and the genes that are differentially expressed ($p < 0.05$, Wald chi-squared test) are highlighted in red (upregulated in C4 stage) or blue (downregulated in C4 stage). The topmost induced genes at the C4 stage compared to the C3 stage are labeled, and some important genes among them are bolded. (H) The MEF cells were subjected to siRNA knockdown for a few of the topmost induced genes (from panel G) and treated with 2 or 4 mM LLOMe, followed by staining with live cell dye Calcein-AM, and quantification was performed using high-content screening. Bottom panels, Graph depicts the percentage of MEF cells revived (*$p = 0.0112$, **$p = 0.0099$, ***$p = 0.0001$, ****$p < 0.0001$, mean ± SD, $n = 4$, 25 fields/well, one-way ANOVA, Tukey's multiple comparison test. Exact $p$ values are depicted in the Figure and Appendix Table S2. Source data are available online for this figure.

mTOR (Torin, pp242, Rapamycin), WNT (IWP-4), AKT (GSK690693), ATM (KU5593), TBK1 (Amelaxanox), interferon response (STING inhibitor, H-151), HSP90 (Epigallocatechin gallate, EGCG), NF-κB (Bay-11-7085), NRF2 (ML385), and lysosome cathepsins (Ca-074-me). A marginal but significant revival defect was observed with many vital pathway inhibitors, suggesting that multiple pathways contribute to the programmed cell revival (Fig. 5C–E). However, the most striking and consistent defect was observed when NF-κB was inhibited using Bay-11-7085 (Fig. 5C–E; Movie EV16). This data supports RNA-seq and ATAC-seq analysis, where we observed that NF-κB-dependent gene expression is dominant during revival. The treatment of Torin and pp242, which inhibit both mTORC1 and mTORC2, could severely reduce the revival of cells, but rapamycin, which only inhibits mTORC1 (Fig. 5F), didn't have a profound effect on the revival of cells, suggesting a role of both mTORC1/2 pathways in revival.

The robustly induced genes at the C4 stage compared to the C3 stage (Fig. 5G) were found to be associated with pathways linked to embryonic development, cellular differentiation, circadian rhythm, metabolism, longevity, regeneration, and innate immunity (Dataset EV3). The example of these genes includes *Lpin1*, a nuclear coactivator, which plays critical role in lipid metabolism; *Rora*, a nuclear receptor, which plays important role in embryonic development, cellular differentiation, immunity, circadian rhythm as well as lipid, steroid, and glucose metabolism; *Per1/2*, a core component of the circadian clock; *Foxo3*, a transcription factor that plays critical function in metabolism, stem cell homeostasis, and longevity; *ATF3*, a master transcription factor for stress response and is pro-regenerative factor important for immune/metabolic homeostasis; *Tef*, a transcription factor play critical role in embryonic development and differentiation; *Rel*, a core component of NF-κB, which is a master regulator of cell proliferation, differentiation, and immunity; *Jun/Fos*, AP-1 transcription factor subunits; *Nr4a3*, a nuclear receptor with function in cell proliferation, differentiation, metabolism and inflammation; *Crebrf/Crebbp* are histone remodelers with important function in embryonic development, growth, circadian rhythm, and homeostasis by coupling chromatin remodeling to transcription factor recognition (Fig. 5G; Dataset EV3). The siRNA knockdown of *Lpin1, Rora, Per1/2, Foxo3, Atf3, Map3k8, Tef, Rel, Fos, Nr4a3,*

*Ep300*, and *Crebbp* significantly compromised the revival of the cells to a different extent (Fig. 5H). Again, the most striking results were obtained with p65-*Rel* knockdown (Fig. 5H), where none of the cells revived back, suggesting that, indeed, NF-κB plays a critical role in the resuscitation of the cells.

## No oncogenic transformations or senescence in revived cells

Since the LLOMe-treated cells undergo multiple stresses, and also multiple pathways related to proliferation were induced during revival, we tested whether revived MEF cells gained any cancerous properties, like faster proliferation, migration, or DNA damage. In MTT assays, the LLOMe-treated revived cells were not proliferating faster; rather, they were slightly slower in growth than untreated control cells (Fig. EV6A). Similarly, in scratch assays, no significant difference was observed in cell migration (Fig. EV6B). Next, we treated the MEF cells multiple times with LLOMe and plated the revived cells from each cycle for clonogenic assays (Fig. EV6C). Again, the number of colonies was either the same or less in the case of LLOMe-treated revived cells (Fig. EV6D). We tested whether LLOMe treatment results in DNA damage. We found that there was no genomic DNA fragmentation upon LLOMe treatment (Fig. EV6E). Next, the MEF cells were subjected to seven cycles of LLOMe treatment and assessed for DNA damage using comet assays (Fig. EV6F) and γH2A.X foci staining (Fig. EV6G). Our results showed no discernible increase in comet tail length or γH2A.X foci formation, indicating that LLOMe does not induce significant DNA damage even after multiple cycles of treatment. Finally, to rigorously evaluate the potential for LLOMe-treated cells to induce tumor formation, we performed in vivo tumorigenicity assays in mice ($n = 6$) (Fig. EV6H). Cells treated with multiple cycles of LLOMe were injected into mice, and the injection sites were monitored for tumor growth over time. We observed no tumor formation in any of the animals tested (Fig. EV6H). Collectively, the data indicate that the revived cells do not seem to possess any advantages in terms of growth or the ability to form tumors.

We investigated whether repeated exposure to LLOMe might accelerate senescence, a key consideration for potential in vivo applications. Even after seven cycles of LLOMe treatment, we

observed no significant increase in cellular senescence, as determined by β-gal staining (Fig. EV6I). Furthermore, Western blot analysis, using established senescence markers such as p16ink4, p21, and γH2A.X, revealed no appreciable differences between cells treated with one cycle of LLOMe and those subjected to multiple cycles (Fig. EV6I). This indicates that LLOMe treatment, at least under these conditions, does not appear to promote premature senescence.

## LLOMe enhances skin wound healing in a mouse model

An effective wound-healing process consists of four key stages: homeostasis, inflammation, proliferation, and resolution/remodeling. The healing of wounds entails a sequence of clearly defined morphogenetic alterations that mirror embryonic development (Bielefeld et al, 2013; Khan et al, 2021). Various molecular pathways such as Wnt/β-catenin (Choi et al, 2022), Notch (Chigurupati et al, 2007), Tgf-b (Penn et al, 2012), FoxO (Ponugoti et al, 2013), and Hox play crucial roles in promoting efficient wound healing, tissue regeneration, and embryonic growth. In addition to these pathways, numerous other signaling pathways, including insulin signaling, Rho GTPases, MAPK, NF-κB, cytokine/interleukin/interferon, calcium signaling, and various metabolic pathways, are activated during the programmed revival process, showing surprising similarities to those induced during wound healing (Krizanova et al, 2022). Consequently, we conducted experiments to investigate the potential of LLOMe treatment in enhancing wound healing in a full excision wound model in mice. LLOMe is dissolved in water at a concentration of 4 and 8 mM, and 20 µl of it is topically applied to the wound surface using sterile surgical cotton every day twice a day (Fig. 6A). Wounds were imaged and measured daily. The group treated with LLOMe showed significantly faster wound healing compared to the control group (Fig. 6B). Within one day of treatment, the average reduction in wound size was 27% in the 4 mM LLOMe group, ~50% in the 8 mM group, and only 3% in the control group (Fig. 6B). By the 3rd day, the average wound size reduction in the 4 mM and 8 mM LLOMe-treated groups was 64% and 78%, respectively, while the control group only showed a 23% reduction. These results suggest that LLOMe could promote skin wound healing.

Multiple cytokines/chemokines, both pro-inflammatory and anti-inflammatory, are essential for wound healing (Mahmoud et al, 2024; Ridiandries et al, 2018). For example, many pro-inflammatory cytokines, such as TNF-α, IL-6, IL-1β, IL-17A, and IFN-γ, play a crucial role in the chemotaxis and migration of various immune cells to the wound-healing site. Many chemokines, such as CXCL1, 2, 4, 7, 19, 12, and CCL2, 3, 5, play a role in all four phases of wound healing. Our RNA-seq analysis suggests the upregulation of multiple cytokine signaling pathways. We performed expression analysis of many of these cytokines, IL17-A, CCL3, CCL5, CXCL1, CXCL10, and IL-6 by qRT-PCR during various stages of cell revival (Fig. EV7A). We also tested secretion of IL-4, IL-10, IL-6, IL-1β, and TNFα using ELISA (Fig. EV7B). We found that all these regenerative anti- and pro-inflammatory cytokines were expressed and secreted in significantly higher amounts from reviving cells. It appears that a sublethal concentration of LLOMe induces a balanced response, where both pro-

inflammatory and anti-inflammatory cytokines/chemokines work together for better tissue repair and regeneration.

## LLOMe accelerates the healing of corneal injury in a mouse model

Chemical injuries to the cornea can severely impair vision. Despite medical and surgical interventions, a considerable number of these injuries continue to lead to vision loss (Clare et al, 2022). Corneal wound healing involves a complex interplay of various cell types, growth factors, and cytokines. Quick re-epithelialization is essential for effective healing of the cornea. The alkali-burned cornea mouse model is the most commonly used model to study chemical injuries to the cornea. In this model, the 0.1 N NaOH is used for complete debridement of corneal epithelium and is confirmed by fluorescent dye staining (Fig. 6C,D). LLOMe is administered topically three times a day. The results showed a significant reduction of the epithelial defect from day 2 in LLOMe-treated groups (the green-colored area indicates epithelial defect) compared to the injured control group (Fig. 6D,E). The experiment was continued until day 7, and the extent of re-epithelialization was dramatically better in LLOMe-treated groups compared to controls (Fig. 6D,E). The results indicate that LLOMe could be a potent therapeutic option for treating corneal burn injuries.

## LLOMe accelerates frog tadpole tail regeneration

The primary focus of regenerative medicine is to discover methods for stimulating the regeneration of organs and appendages. The frog tadpole tail regeneration model is a robust model for investigating the fundamental principles of tissue regeneration (Beck et al, 2009; Phipps et al, 2020; Tseng and Levin, 2008).

We tested the efficacy of LLOMe in accelerating a tadpole tail regeneration in *Polypedates maculatus* (Asian tree frog) model (Hota et al, 2018; Mahapatra et al, 2023; Mahapatra and Mohanty-Hejmadi, 1994). The pathways (Notch, MAPK, mTOR, Ras, TGFβ, Hippo, Wnt, and inflammation) induced during limb regeneration in this model (Mahapatra et al, 2023) were strikingly similar to the pathways upregulated upon LLOMe treatment. The tails of *Polypedates maculatus* tadpoles at Gosner stage 26 were amputated, and then they were subjected to various concentrations of LLOMe treatments (administered in water) for four different doses (Figs. 6F and EV7C). Blastema is a cluster of undifferentiated cells that forms at the site of injury and has the ability to regenerate into an organ or appendage (Beck et al, 2009; Seifert and Muneoka, 2018). A rapid blastema formation was observed in the treated group within 24 h of amputation (Fig. EV7C). At this stage, we observed increased production of reactive oxygen species (ROS), heightened lysosomal activity, and intracellular calcium flux in the regenerating tail tissues treated with LLOMe (Fig. EV7D–F), which mirrored the conditions observed in vitro. All these changes are crucial for tissue regeneration in different animal models (Coward et al, 1974; Love et al, 2013; Marchant, 2019; Shi et al, 2020). Over several days, the length of the regenerated tails was measured (Fig. 6F). The treated group displayed significantly accelerated tail regeneration across all concentrations of LLOMe compared to the control group (Fig. 6F,G). These findings suggest that LLOMe has the potential to enhance tissue regeneration in the tadpole tail regeneration model.

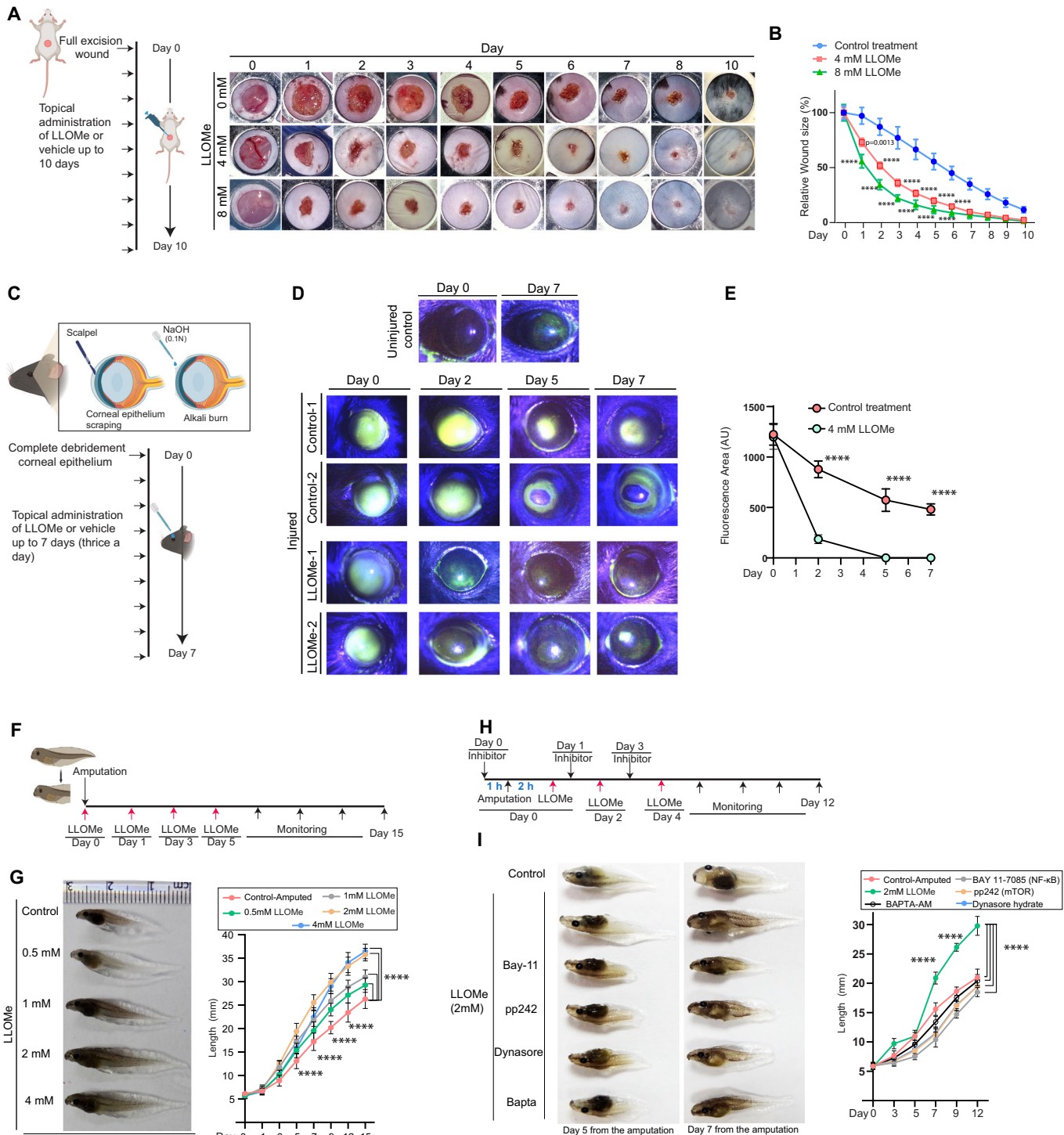

## NF-κB, mTOR, and calcium signaling are critical for frog tadpole tail regeneration

Next, we tested whether the pathways that we found to be critical for the revival of LLOMe-treated cells, were also important for the regeneration of the tadpole tails. Indeed, treatment with Bay-11-7085 (NF-κB inhibitor), pp242 (mTOR inhibitor), Dynasore hydrate (endocytosis inhibitor), and BAPTA-AM (Ca²⁺ chelator) dramatically suppressed LLOMe-induced tail regeneration (Fig. 6H,I), indicating that LLOMe enhances regeneration by inducing similar pathways as observed during programmed cell revival in vitro.

Figure 6.   LLOMe accelerates wound healing and tadpole tail regeneration.

(A) Left panel schematic depicting the experimental design and the timeline of topical administration of LLOMe or vehicle control. Right panel, representative images showing wound healing in mice skin after full excision wound and topical treatment with 4 or 8 mM LLOMe or vehicle control. (B) Graph depicting the relative wound size for vehicle control and LLOMe (4 or 8 mM) treated wounds. (**$p = 0.0013$, ****$p < 0.0001$, mean ± SD, $n = 3$, two-way ANOVA, Dunnett multiple comparison test). (C) Schematic depicting the experimental design and the timeline of topical administration of LLOMe or vehicle control. (D) Representative fluorescent images of uninjured controls, injured controls, and 4 mM LLOMe-treated mouse eyes at days 0, 2, 5, and 7. (E) The graph depicts the relative fluorescence area of injured control and LLOMe-treated mouse eyes. (****$p < 0.0001$, mean ± SD, $n = 3$, two-way ANOVA, Sidak multiple comparison test). (F) Schematic depicting the experimental timeline for LLOMe treatment and monitoring the tail regeneration in tadpole larvae. (G) Representative images of tail regeneration in tadpoles after treatment with the indicated concentration of LLOMe on day 9 from the day of amputation. The graph depicts the tail length upon treatment with the indicated concentrations of LLOMe and time. (****$p < 0.0001$, mean ± SD, $n = 15$, two-way ANOVA, Dunnett multiple comparison test). (H) Schematic depicting the experimental timeline for LLOMe and inhibitor treatments and monitoring the tail regeneration in tadpole larvae. (I) Representative images of tail regeneration in tadpoles after treatment with LLOMe (2 mM) and inhibitors at days 5 and 7 from the day of amputation. The graph depicts the tail length upon treatment with LLOMe alone or with the indicated inhibitors over the indicated period of time. (****$p < 0.0001$, mean ± SD, $n = 10$, two-way ANOVA, Dunnett multiple comparison test). Source data are available online for this figure.

## LLOMe induces axon regeneration in the *Caenorhabditis elegans* model

To further test the efficacy of LLOMe in inducing regeneration, we utilized the *Caenorhabditis elegans* axon regeneration model, an excellent model for studying neuronal regeneration (Byrne and Hammarlund, 2017; Ghosh-Roy and Chisholm, 2010). An axotomy with one of the posterior lateral microtubule (PLM) neurons was performed on day 3 (A3) animals using a pulsed UV laser, and the response was measured (Basu et al, 2017; Kumar et al, 2021) (Fig. 7A). The axotomy of PLM reduces posterior touch response in worms, which is measured as the posterior touch response index (PTRI) as described previously, and methods (Basu et al, 2017; Kumar et al, 2021) (Fig. 7A). The regrowth length after axotomy is significantly enhanced with LLOMe treatment (Fig. 7B,C). Additionally, functional restoration, as measured by posterior touch response, was significantly improved after LLOMe treatment, as reflected in the increased recovery index (Fig. 7D). These results suggest that LLOMe treatment enhances axon regeneration and accelerates functional recovery.

## LLOMe increases the hematopoietic progenitor/stem cells in the *Drosophila* lymph glands

The *Drosophila melanogaster* larval lymph gland is a well-studied model for understanding hematopoietic stem and progenitor cells (HSPCs) and recapitulates several aspects of vertebrate hematopoiesis (Evans et al, 2003; Morin-Poulard et al, 2021). They are the primary site of hematopoiesis, containing myeloid-like progenitor/stem cells that differentiate into functional hemocytes in the circulation of pupae and adults. Multiple signaling pathways that were perturbed by LLOMe, such as NF-κB, WNT, NOTCH, JAK-STAT, mTOR, FOXO, and insulin-like growth factor, were shown to play critical roles in HSPCs maintenance and function (Evans et al, 2003; Koranteng et al, 2022; Morin-Poulard et al, 2021; Ramesh et al, 2021; Tokusumi et al, 2012). We investigated JAK/STAT pathway activity in vivo using reporter line 10XSTAT92E-eGFP, which drives GFP expression under the control of ten STAT92E binding sites(Bach et al, 2007) (Fig. 7E). Our analysis shows that LLOMe treatment significantly increased the 10xStat-GFP reporter activity in the *Drosophila* larval lymph gland as compared to the control (Fig. 7F,G).

The *Drosophila* larval lymph gland consists of core hematopoietic progenitors that are positive for the progenitor-specific driver thioester-containing protein-4 "tep4" (Blanco-Obregon et al, 2020). We used *tep4-Gal4*, UAS-GFP X wild-type larvae, where the

tep4 positive population can be identified using GFP expression (Fig. 7H). Third instar *Drosophila* larvae were treated with LLOMe (8 mM) for 14 h (Fig. 7H). There was a pronounced increase in the *tep4* positive core progenitor index per lymph gland lobe, indicating an increase in the stem cell pool upon LLOMe treatment (Fig. 7I,J). We also observed a consequent increase in overall mitotic activity as indicated by an increase in H3P-positive cells per lymph gland lobe (Fig. 7I,J). Further, a significant increase in the expression of Relish (NF-κB) was evident in the lymph glands of the treated group (Fig. 7K–M), which is consistent with its role in maintaining the pool of hematopoietic stem and progenitor cells (Ramesh et al, 2024). Taken together, the data show that LLOMe robustly induces HSPCs in the lymph glands of *Drosophila*.

## Discussion

This study reveals that the recovery from initial cell death conditions is a meticulously orchestrated and programmed event, where coordinated action of multiple signaling pathways governs the revival process. Consequently, we coined the term programmed cell revival to describe it. The most significant outcome of this study is that the knowledge gained about the programmed cell revival process has been successfully leveraged to enhance wound healing, tissue regeneration, and stemness in animal models. We believe that there is significant potential to harness this process for therapeutic advances in regenerative medicine.

### Cell death versus cell death-like conditions

We serendipitously found that many cell types can recover from cell death-like conditions when exposed to sublethal concentrations of LLOMe. Similarly, sphingosine and GPN, two other lysosomotropic agents, exhibited a comparable effect. Our findings indicate that cells can recover even when the integrity of the plasma membrane is compromised, but not after disruption of the nuclear membrane. Our live cell imaging data demonstrates that cells can survive and proliferate even after the cytoplasmic uptake of dead cell dyes like DAPI, ethidium homodimer-1, SYTOX, and PI. Thus, strong staining of the nucleus with nuclear impermeant dyes visualized by microscopy (not FACs) could be considered a definitive hallmark of cell death. Morphological changes, annexin/PI staining (by FACS), activation of molecular markers of cell death, or slight LDH release could be deceptive in marking cells to be dead.

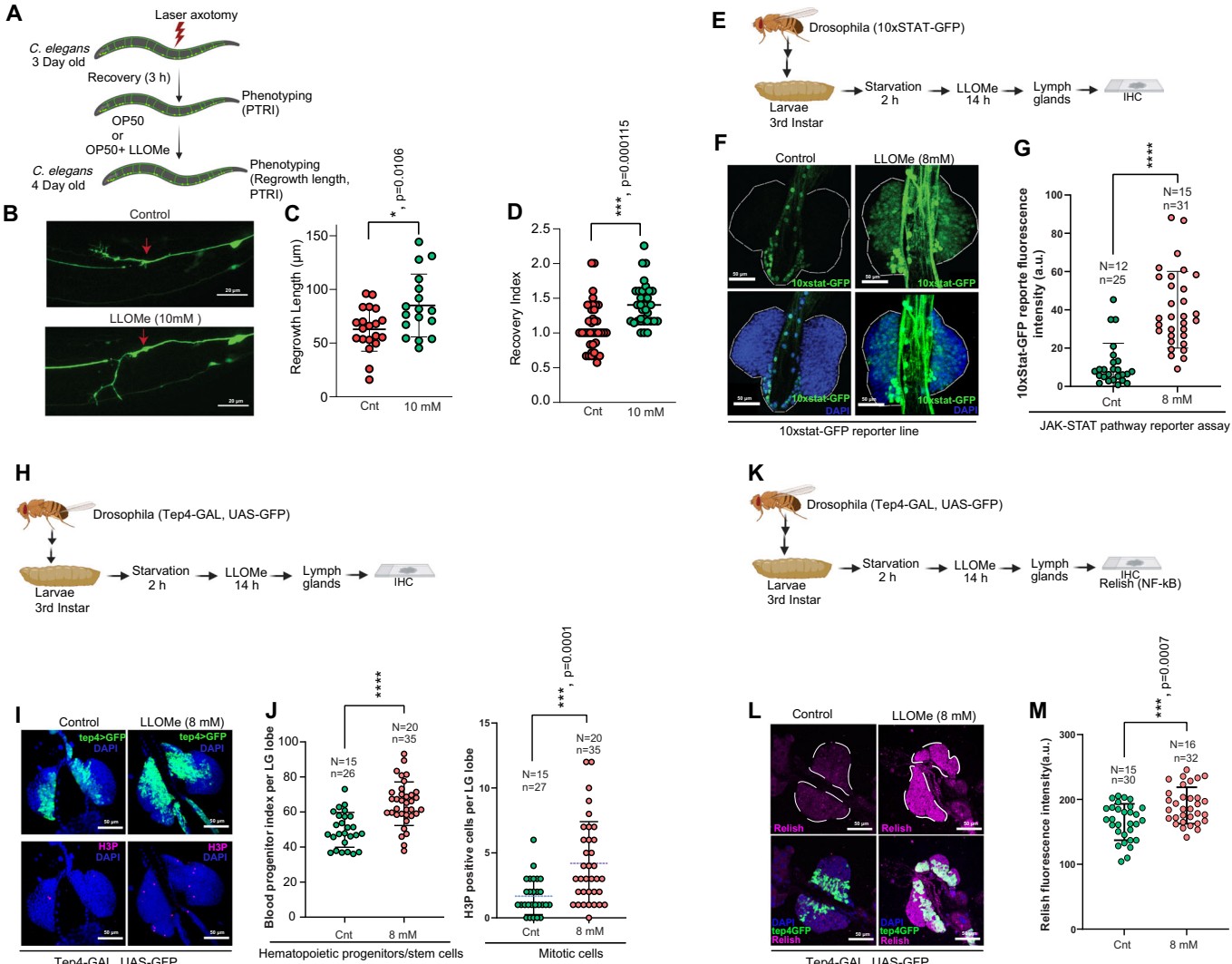

**Figure 7. LLOMe accelerates axon regeneration and stemness.**

(A) Schematics representing axotomy and treatment plan of *C. elegans* for evaluating phenotypes of axon regeneration and posterior touch response index (PTRI). (B) Representative images showing axon regeneration after LLOMe (10 mM) treatment *in C. elegans*. The red arrow indicates the axotomy position. (C) The bar graph depicts regrowth length 24 h after axotomy. (*$p = 0.0106$, mean ± SD, $N = 4$, $n = 17$–20, unpaired *T*-test). (D) The bar graph shows an improved recovery index after LLOMe treatment. (***$p = 0.000115$, mean ± SD, $N = 3$, $n = 35$–38, unpaired *T*-test). Scale Bar, 20 μm. Exact *p* values are depicted in the Figure and Appendix Table S2. (E) Schematic depicting the experimental timeline of LLOMe treatment of *Drosophila* (*10xSTAT-GFP*) third instar larva. (F) Representative immunohistochemistry confocal images of *Drosophila* third instar larval lymph gland showing (top panels) JAK/STAT pathway activation. (G) The graphs depict *10xSTAT-GFP* reporter fluorescence intensity. (****$p < 0.0001$, mean ± SD, two-tailed unpaired Student's *t*-test, Welch's correction, *N* (number of larvae) for control is 12 and *n* (number of lobes) is 25, whereas *N* for LLOMe treated is 15 and n is 31. Scale Bar, 50 μm. Exact *p* values are depicted in the Figure and Appendix Table S2. (H) Schematic depicting the experimental timeline of LLOMe treatment of *Drosophila* (*Tep4-Gal4, UAS-GFP*) 3rd instar larva. (I) Representative immunohistochemistry (IHC) confocal images of larval lymph gland showing (tep4 (*Tep4-Gal4, UAS-GFP*, indicated as Tep4>GFP) positive core hematopoietic progenitors and phospho-histone H3 (H3P, Serine 10) positive mitotically active cells. (J) The graphs depict blood progenitor index per lymph gland lobe or H3P-positive cells per lobe. ***$p = 0.0001$, ****$p < 0.0001$, mean ± SD, two-tailed unpaired Student's *t*-test, Welch's correction, *N* (number of larvae) and *n* (number of lobes). Scale Bar, 50 μm. Exact *p* values are depicted in the Figure and Appendix Table S2. (K) Schematic depicting the experimental timeline of LLOMe treatment of *Drosophila* (*Tep4-Gal4, UAS-GFP*) 3rd instar larva, followed by IHC with Relish antibody. (L) Representative IHC confocal images with Relish antibody. (M) The graph depicts Relish fluorescence intensity. ***$p = 0.0007$, mean ± SD, two-tailed unpaired Student's *t*-test, Welch's correction, *N* (number of larvae) and *n* (number of lobes). Scale Bar, 50 μm. Exact *p* values are depicted in the Figure and Appendix Table S2. Source data are available online for this figure.

## Cell biology of programmed cell revival

The floating cells that were poised to re-initiate the program of life were found to be devoid of intact cell organelles (Fig. 2). Intriguingly, most of the organelle's morphology was restored within a few hours except for acidic compartments. The cells that survived the LLOMe treatment displayed large acidic and enzymatically active vacuoles. Highly networked and polarized mitochondria surrounded these transient structures. Such organelles were not reported before in the literature, so we envisage that

these complex units may act as a center for providing energy and metabolites for cell revival. However, further studies may be needed to understand their precise function in PCR. Other dramatic changes that were evident in the reviving cells were increased mitochondrial polarization (95–96%), ROS production, and increased membrane trafficking. We believe that all these changes trigger multiple signaling pathways to meet the energy, biosynthetic, and metabolic demands of resuscitating cells.

## Pathways and genes of programmed cell revival

The activation of pathways associated with tissue development, morphogenesis, differentiation, pattern specification, chromatin modification/organization, circadian rhythms, and various signaling pathways such as TGFb, WNT, NOTCH, HOX, RUNX1, and AKT at the beginning of the revival process suggests that an embryonic developmental signaling program is reawakening cells. Bioinformatics analysis of gene signatures during the initial phase of cell revival indicates a resemblance to early-stage pluripotent stem cells. This leads to an intriguing hypothesis that dying cells may lose their original identity and undergo reprogramming, allowing them to restart their differentiation processes upon revival.

Multiple inflammatory pathways were activated during PCR, with the IFN response being particularly notable. This response is crucial for antiviral defense, wound healing, and tissue regeneration (Carey et al, 2024; Carvajal Ibanez et al, 2023; Di Domizio et al, 2020; Leibowitz et al, 2021; Wu et al, 2021). This suggests that the revival process is not random but rather a meticulously orchestrated phenomenon, where different pathways are activated at specific stages of revival. Notably, the activation of circadian rhythms and chromatin-modifying/organizing enzymes occurred exclusively at the onset of PCR. The precise mechanisms by which a cell remembers and orchestrates the revival remain unclear. Inhibitor studies aimed at identifying the primary pathways governing this process demonstrated that multiple signaling events collaborate to regulate revival, with some being essential for successful resuscitation. For instance, inhibiting the NF-κB pathway through chemical or genetic means resulted in the failure of PCR, aligning with observations that blocking this pathway hindered regeneration in frog tadpole tails.

Many genes, such as *Lipin1, Foxo3, Per1, Cxcl2, RelA*, and *Atf3*, that were strongly upregulated at the onset of PCR were shown to be important in wound healing, and tissue regeneration (Jiang et al, 2015; Katoku-Kikyo et al, 2021; Miao et al, 2019; Niethamer et al, 2023). Our data strongly suggests that the in vitro knowledge of the PCR process could be extrapolated and exploited to identify critical modulators of the wound healing and tissue regeneration process.

## Epigenetic regulation of programmed cell revival

At the onset of the revival phase, significant alterations were observed in the transcriptional landscape of the cells. There was an upregulation of numerous genes involved in various pathways. We hypothesized that the epigenetic modifications occurring during revival lead to these transcriptional changes. To investigate this, we conducted nanopore genomic sequencing to assess DNA methylation. However, our analysis revealed no significant changes in DNA

methylation patterns across the different stages. Subsequently, we performed ATAC sequencing, which indicated that the initiation of PCR was associated with enhanced chromatin accessibility, suggesting a possible mechanism underlying the transcriptional changes. This study raises critical questions about the nature of epigenetic modifications that facilitate these transcriptional shifts and whether cells possess a form of genetic memory that enables them to transition back from a state of death to revival. Further exploration of these mechanisms could provide insights into cellular resilience and adaptability.

## Healing and regenerative capacity of LLOMe

Cell death pathways and proteins are crucial components in tissue repair and regeneration across various model organisms such as *Drosophila*, planaria, hydra, and mouse (Ankawa et al, 2021; Bergmann and Steller, 2010; Chera et al, 2009; Fogarty and Bergmann, 2017; Li et al, 2010; Vriz et al, 2014). Additionally, the activation of apoptosis proteins, such as caspase-3/8, and necroptosis kinase RIP3, is shown to play a crucial role in the reprogramming of induced pluripotent stem cells (iPSCs) from fibroblast cells. It is believed that apoptotic cells release signals that stimulate the growth and division of neighboring healthy cells in a paracrine mode (Bergmann and Steller, 2010; Chera et al, 2009; Fogarty and Bergmann, 2017; Li et al, 2010; Vriz et al, 2014). The sublethal concentrations of LLOMe weakly induce apoptotic features.

Cell death, encompassing both non-lytic apoptosis and various forms of lytic cell death (pyroptosis, necroptosis, and necrosis), plays a complex role in wound healing and regeneration (Artlett, 2013; Liu et al, 2022b; Mehrotra et al, 2024; Mu et al, 2022; Zhou et al, 2020). While apoptosis is generally recognized as a key process facilitating tissue remodeling and resolution of inflammation without causing further damage, the impact of lytic cell deaths is more complex. Traditionally, lytic cell death has been viewed as potentially detrimental, contributing to fibrosis and impaired healing (Liu et al, 2022b). This negative association stems from the characteristic release of damage-associated molecular patterns (DAMPs) during lytic cell death (Liu et al, 2022b). Nevertheless, DAMPs production is essential for tissue repair and regeneration (Venereau et al, 2015).

It's important to acknowledge that both pro-inflammatory and anti-inflammatory cytokines are indispensable for successful wound healing (Mahmoud et al, 2024). Their temporal sequence of production and the delicate equilibrium between them are vital for orchestrating cellular recruitment, matrix deposition, and eventual tissue remodeling (Wu et al, 2025). Emerging evidence from multiple studies demonstrates that necroptosis and pyroptosis, under specific conditions, can exert beneficial effects on wound healing and regeneration (Artlett, 2013; Klemm et al, 2024; Lloyd et al, 2019; Zhou et al, 2020). A recent study shows that the secretome of pyroptotic cells possesses wound-healing properties (Mehrotra et al, 2024). Therefore, understanding the complex roles of each cell death pathway and their interplay within the wound microenvironment is essential for developing targeted therapeutic strategies to optimize tissue repair and regeneration.

It is reasonable to anticipate a close similarity in development and regeneration, considering that regenerative processes have to

reconstruct structures that were previously formed during development. Indeed, multiple studies have shown that gene regulatory networks implicated in embryonic development are redeployed during regeneration (Johnston et al, 2021; Loubet-Senear and Srivastava, 2024; Nacu and Tanaka, 2011; Sinigaglia et al, 2022; Soubigou et al, 2020). The molecular signaling events that govern the revival of cells from LLOMe-induced cell death-like conditions are strikingly similar to the pathways required for wound healing and tissue regeneration, including the reactivation of the embryonic development program. Wound healing has three stages after hemostasis, i.e., inflammation, proliferation, and remodeling. LLOMe induces all. We also found that both pro-inflammatory and anti-inflammatory cytokines are induced during the revival of the cells. It appears that somehow a perfect balance of these cytokines is maintained during revival following treatment with sublethal concentrations of LLOMe, resulting in a significant increase in wound healing and regeneration without fibrosis or scar formation. A deeper understanding of this process may provide us not only with intriguing insights into cell biology but also with novel approaches for tissue repair and regeneration.

# Methods

## Cell culture

All the cell lines, including HEK293T, MDA-MB-231, HT-29, HeLa, UM-UC-3, NIH/3T3, BHK-21, WT SV40 MEF, and A549, were obtained from the American Type Culture Collection (ATCC), US. Cells were cultured in DMEM (Gibco) supplemented with 10% heat-inactivated fetal bovine serum, FBS (Gibco) and 10,000 units/ml penicillin-streptomycin (Gibco). All cell lines were maintained at 37 °C and 5% $CO_2$ and were routinely checked for mycoplasma contamination. Experiments were carried out using cells that were below the 20th passage.

## Inhibitors and reagents

### Lysosomotropic agents used in this study

Leu-Leu methyl ester hydrobromide, LLOMe (Millipore-Sigma; 4, 8 and 12 mM), Imject™ Alum Adjuvant (Thermo Fisher Scientific; 1 mg/ml); Gly-Phe-β-naphthylamide, GPN (Cayman; 200 µM);

**Reagents and tools table**

| Reagent/resource | Reference or source | Identifier or catalog number |
| --- | --- | --- |
| **Experimental models** | | |
| HEK293T | ATCC | CRL-3216 |
| MDA-MB-231 | ATCC | HTB-26 |
| HT-29 | ATCC | HTB-38 |
| HeLa | ATCC | CCL2 |
| UM-UC-3 | ATCC | CRL-1749 |
| NIH/3T3 | ATCC | CRL-1658 |
| BHK-21 | ATCC | CCL-10 |
| WT SV40 MEF | ATCC | CRL-2907 |
| A549 | ATCC | CCL-185 |
| C57BL/6J mice | The Jackson Laboratory | Cat#000664;RRID:IMSR_JAX:000664 |
| *Polypedates maculatus* | Pravati Kumari Mahapatra's Lab | N/A |
| *Caenorhabditis elegans* | Anindya Ghosh-Roy's Lab | N/A |
| *Drosophila melanogaster* | Rohan Jayant Khadilkar's Lab | N/A |
| **Recombinant DNA** | | |
| ZipGFP-Casp3 | Addgene | Cat#81241 |
| **Antibodies** | | |
| Anti-pro Caspase-1 + p10 + p12 antibody [EPR16883] (Rabbit-mAb) | Abcam | Cat#179515; RRID:AB_2884954 |
| Gasdermin D (Rabbit-pAb) | Cell Signaling Technology | Cat#96458; RRID:AB_2894914 |
| Phospho-RIP3 (Thr231/Ser232) (mouse specific) (Rabbit-pAb) | Cell Signaling Technology | Cat#57220; RRID:AB_2799526 |
| Phospho-MLKL (Ser345) (D6E3G) (Rabbit-mAb) | Cell Signaling Technology | Cat#37333; RRID:AB_2799112 |
| RIP3 (Rabbit-pAb) | Abcam | Cat# ab152130 |
| MLKL (D6W1K) (mouse specific) (Rabbit-mAb) | Cell Signaling Technology | Cat#37705; RRID:AB_2799118 |
| Anti-beta Actin (Mouse-mAb) | Abcam | Cat# ab6276; RRID:AB_2223210 |
| EP300 (Rabbit-pAb) | GeneTex | Cat# GTX134238; RRID:AB_2887243 |

| Reagent/resource | Reference or source | Identifier or catalog number |
| --- | --- | --- |
| ATF3 (D2Y5W) (Rabbit-mAb) | Cell Signaling Technology | Cat#33593; RRID:AB_2799039 |
| Histone H3 (Rabbit-pAb) | Invitrogen | Cat# PA5-17697; RRID:AB_10980629 |
| Anti-LC3B (Rabbit-pAb) | Millipore-Sigma | Cat# L7543; RRID:AB_796155 |
| Phospho-NF-κB-p65 (Ser536) (93H1) (Rabbit-mAb) | Cell Signaling Technology | Cat#3033; RRID:AB_331284 |
| CBP antibody [C3], C-term (Rabbit-pAb) | GeneTex | Cat# GTX101249; RRID:AB_1240560 |
| Lamin B1 (D4Q4Z) (Rabbit-pAb) | Cell Signaling Technology | Cat#12586; RRID:AB_2650517 |
| Phospho-AMPKα (Thr172) (clone: 40H9) (Rabbit-mAb) | Cell Signaling Technology | Cat#2535; RRID:AB_331250 |
| AMPKα (D5A2) (Rabbit-mAb) | Cell Signaling Technology | Cat#5831; RRID:AB_10622186 |
| Anti-c-Jun (phospho S73) antibody [E107] (Rabbit-mAb) | Abcam | Cat# ab32447; RRID:AB_2129699 |
| Phospho-Akt (Ser473) (Rabbit-pAb) | Cell Signaling Technology | Cat#9271; RRID:AB_329825 |
| Phospho-p38 MAPK (Thr180/Tyr182) (Rabbit-pAb) | Cell Signaling Technology | Cat#9211; RRID:AB_331641 |
| p38 MAPK (Rabbit-pAb) | Cell Signaling Technology | Cat#9212; RRID:AB_330713 |
| Anti-SREBP1 Antibody, clone 2121 (Mouse-mAb) | Millipore-Sigma | Cat#04-469; RRID:AB_612072 |
| Caspase-3 (Rabbit-pAb) | Cell Signaling Technology | Cat#9662; RRID:AB_331439 |
| PARP (Rabbit-pAb) | Cell Signaling Technology | Cat#9542; RRID:AB_2160739 |
| ERp72 (D70D12) (Rabbit-mAb) | Cell Signaling Technology | Cat#5033; RRID:AB_10622112 |
| EEA1 (C45B10) (Rabbit-mAb) | Cell Signaling Technology | Cat#3288; RRID:AB_2096811 |
| Rab7 (D95F2) (Rabbit-mAb) | Cell Signaling Technology | Cat#9367; RRID:AB_1904103 |
| RCAS1 (D2B6N) (Rabbit-mAb) | Cell Signaling Technology | Cat#12290; RRID:AB_2736985 |
| Anti-beta Tubulin (Rabbit-pAb) | Abcam | Cat#ab6046; RRID:AB_2210370 |
| LC3b (Rabbit-pAb) | MBL | Cat#PM036; RRID:AB_2274121 |
| Anti-phospho-Histone H3 (Ser10) (Rabbit-pAb) | Millipore-Sigma | Cat#06-570; RRID:AB_310177 |
| Alexa Fluor 488 goat anti-mouse | Invitrogen | Cat#A11029 |
| Alexa Fluor 488 goat anti-Rabbit | Invitrogen | Cat#A11034 |
| Alexa Fluor 568 goat anti-mouse | Invitrogen | Cat#A11031 |
| Alexa Fluor 568 goat anti-Rabbit | Invitrogen | Cat#A11036 |
| Alexa Fluor 647 goat anti-mouse | Invitrogen | Cat#A21236 |
| Alexa Fluor 647 goat anti-Rabbit | Invitrogen | Cat#A21245 |
| Goat anti-Mouse IgG (H + L) Secondary Antibody [HRP] | Novus Biologicals | Cat#NB7539 |
| Goat anti-Rabbit IgG (H + L) Secondary Antibody [HRP] | Novus Biologicals | Cat#NB7160 |
| p-Histone H2A.X (Ser 139) (Mouse-mAb) | Santa Cruz | Cat#sc-517348; RRID:AB_2783871 |
| p21 Waf1/Cip1 (12D1) (Rabbit-mAb) | Cell Signaling Technology | Cat#2947; RRID:AB_823586 |
| Anti-p16/INK4a, clone EP435Y (Rabbit-mAb) | Millipore-Sigma | Cat#04-239; RRID:AB_1587343 |
| **Oligonucleotides and other sequence-based reagents** | | |
| RORA siRNA | Millipore-Sigma | Cat#SASI_Mm01_00034667 |
| LPIN1 siRNA | Millipore-Sigma | Cat#SASI_Mm02_00311168 |
| ATF3 siRNA | Millipore-Sigma | Cat#SASI_Mm02_00311770 |
| MAP3K8 siRNA | Millipore-Sigma | Cat#SASI_Mm01_00157847 |
| NR4A3 siRNA | Millipore-Sigma | Cat#SASI_Mm01_00025082 |
| RELA siRNA | Millipore-Sigma | Cat#SASI_Mm01_00060588 |
| FOXO3 siRNA | Millipore-Sigma | Cat#SASI_Mm02_00324883 |
| PER1 siRNA | Millipore-Sigma | Cat#SASI_Mm01_00160668 |
| FOS siRNA | Millipore-Sigma | Cat#SASI_Mm01_00192758 |
| TEF siRNA | Millipore-Sigma | Cat#SASI_Mm01_00163226 |
| EP300 siRNA | Millipore-Sigma | Cat#SASI_Mm01_00159721 |

| Reagent/resource | Reference or source | Identifier or catalog number |
|---|---|---|
| CREBBP siRNA | Millipore-Sigma | Cat#SASI_Mm02_00287878 |
| Mouse CCL3(MIP-1a) Forward primer 5′-GATTCCACGCCAATTCATCG-3′ | This study | N/A |
| Mouse CCL3(MIP-1a) Reverse primer 5′-CTGCCGGTTTCTCTTAGTCAG-3′ | This study | N/A |
| Mouse CCL5(RANTES) Forward primer 5′-GGGTACCATGAAGATCTCTGC-3′ | This study | N/A |
| Mouse CCL5(RANTES) Reverse primer 5′-GGAGAGGTAGGCAAAGCAG-3′ | This study | N/A |
| Mouse CXCL1 Forward primer 5′-AACCGAAGTCATAGCCACAC-3′ | This study | N/A |
| Mouse CXCL1 Reverse primer 5′- CAGACGGTGCCATCAGAG-3′ | This study | N/A |
| Mouse CXCL10 Forward primer 5′-AGCACCATGAACCCAAGTG-3′ | This study | N/A |
| Mouse CXCL10 Reverse primer 5′- CTGGCCCGTCATCGATATG-3′ | This study | N/A |
| Mouse IL-4 Forward primer 5′-TGGAAGCCCTACAGACGAG-3′ | This study | N/A |
| Mouse IL-4 Reverse primer 5′- GCATTTTGAACGAGGTCACAG-3′ | This study | N/A |
| Mouse IL-6 Forward primer 5′-CCTCTGGTCTTCTGGAGTACC-3′ | This study | N/A |
| Mouse IL-6 Reverse primer 5′- ACTCCTTCTGTGACTCCAGC-3′ | This study | N/A |
| Mouse IL-10 Forward primer 5′-GTCATCGATTTCTCCCCTGTG-3′ | This study | N/A |
| Mouse IL-10 Reverse primer 5′- ATGGCCTTGTAGACACCTTG-3′ | This study | N/A |
| Mouse IL-17A Forward primer 5′-CAAACATGAGTCCAGGGAGAG-3′ | This study | N/A |
| Mouse IL-17A Reverse primer 5′- ACACGCTGAGCTTTGAGG-3′ | This study | N/A |
| Mouse ATF3 Forward primer 5′-AAGATGAGAGGAAAAGGAGGC-3′ | This study | N/A |
| Mouse ATF3 Reverse primer 5′-CTCAGCATTCACACTCTCCAG-3′ | This study | N/A |
| Mouse FOS Forward primer 5′-CAGCCTTTCCTACTACCATTCC-3′ | This study | N/A |
| Mouse FOS Reverse primer 5′-GGATAAAGTTGGCACTAGAGACG-3′ | This study | N/A |
| Mouse MAP3K8 Forward primer 5′-ACTCTGCCCTCTTTGAACG-3′ | This study | N/A |
| Mouse MAP3K8 Reverse primer 5′-GAGGTCAATGTAGAGGGAACG-3′ | This study | N/A |
| Mouse NR4A3 Forward primer 5′-CAGAGCCTGAACCTTGATATCC-3′ | This study | N/A |
| Mouse NR4A3 Reverse primer 5′-TGTGATCTTGGTGCATAGCTC-3′ | This study | N/A |
| Mouse RSAD2 Forward primer 5′- | This study | N/A |
| **Chemicals, Enzymes and other reagents** | | |
| Dulbecco's Modified Eagle Medium (DMEM) | Gibco | Cat#10569010 |
| Fetal Bovine Serum (FBS) | Gibco | Cat#26140079 |
| Penicillin-Streptomycin | Gibco | Cat#15070063 |
| Leu-Leu methyl ester hydrobromide (LLOMe) | Millipore-Sigma | Cat#L7393 |
| Imject™ Alum Adjuvant | Thermo Fisher Scientific | Cat#77161 |
| Gly-Phe-β-naphthylamide (GPN) | Cayman Chemical | Cat#14634 |
| Silica | Millipore-Sigma | Cat#381276 |
| Siramesine fumarate salt | Millipore-Sigma | Cat#SML0976 |
| D-erythro-Sphingosine | Millipore-Sigma | Cat#567726 |
| Amlexanox | InvivoGen | Cat#inh-amx |
| H-151 | Cayman Chemical | Cat#25857 |
| Bafilomycin A1 | Cell Signaling Technology | Cat#54645S |
| BAPTA/AM | Millipore-Sigma | Cat#196419 |
| Bay-11-7085 | Cayman Chemical | Cat#14795 |
| Torin 1 | InvivoGen | Cat#inh-tor1 |

| Reagent/resource | Reference or source | Identifier or catalog number |
|---|---|---|
| Dynasore Hydrate | Millipore-Sigma | Cat# D7693 |
| Cytochalasin D | Millipore-Sigma | Cat#250255 |
| Nocodazole | Millipore-Sigma | Cat#M1404 |
| pp242 | InvivoGen | Cat#inh-pp242 |
| AY 9944 | Cayman Chemical | Cat#14611 |
| Fatostatin (hydrobromide) | Cayman Chemical | Cat#13562 |
| IWP-4 | Cayman Chemical | Cat#13954 |
| ML385 | Millipore-Sigma | Cat#SML1833 |
| Epigallocatechin gallate (EGCG) | Millipore-Sigma | Cat#E4143 |
| KU-55933 | Millipore-Sigma | Cat#SML1109 |
| GSK690693 | Cayman Chemical | Cat#16891 |
| 3-Methyladenine | Millipore-Sigma | Cat#M9281 |
| Rapamycin | Thermo Fisher Scientific | Cat#ALF-J62473-MF |
| Hydroxychloroquine sulfate, HCQ | Millipore-Sigma | Cat#H0915 |
| Mdivi-1 | Millipore-Sigma | Cat#M0199 |
| N-Acetyl-L-Cysteine, NAC | Millipore-Sigma | Cat#A7250 |
| MitoTEMPO | Millipore-Sigma | Cat#SML0737 |
| Carbonyl cyanide m-chlorophenylhydrazone, CCCP | Millipore-Sigma | Cat#C2759 |
| CA-074 methyl ester | Cayman Chemical | Cat#18469 |
| Collagenase, type-IV | GIBCO | Cat#17104-019 |
| Trypsin (2.5%) | GIBCO | Cat#15090046 |
| DNAse I | Roche | Cat#10104159001 |
| NP-40 lysis buffer | Invitrogen | Cat#FNN0021 |
| PMSF | Millipore-Sigma | Cat#P7626 |
| phosSTOP | Roche | Cat#49068455001 |
| Protease inhibitors | Roche | Cat#1183617000 |
| Tween-20 | SIGMA | Cat#P1379 |
| 2x Laemmli Sample Buffer | Bio-Rad | Cat#1610737 |
| Prolong Gold antifade mountant with DAPI | Invitrogen | Cat#P36931 |
| FluoroBrite DMEM | GIBCO | Cat#A18967-01 |
| Hoechst 33342 | Invitrogen | Cat#H3570 |
| Dil Stain | Invitrogen | Cat#D282 |
| Ethidium homodimer-1 | Invitrogen | Cat#L3224 |
| DAPI | Invitrogen | Cat#D1306 |
| Calcein-AM | Invitrogen | Cat#L3224 |
| Trizol | Invitrogen | Cat#15596026 |
| PowerUp™ SYBR™ Green Master Mix | Applied Biosystems | Cat#A25742 |
| Lipofectamine RNAiMAX | Invitrogen | Cat#13778075 |
| Pierce BCA Protein Assay Kit | Thermo Fisher Scientific | Cat#23225 |
| Annexin V-FITC/PI | ebiosciences | Cat##88800574 |
| LysoTracker™ Red DND-99 | Invitrogen | Cat#L7528 |
| MitoTracker™ Red CMXRos | Invitrogen | Cat#M7512 |
| MitoTracker™ Green FM | Invitrogen | Cat#M7514 |
| Rhod-2 AM | Invitrogen | Cat#R1245MP |

| Reagent/resource | Reference or source | Identifier or catalog number |
|---|---|---|
| Magic Red | Immunochemistry Technologies | Cat#937 |
| LysoTracker™ Green DND-26 | Invitrogen | Cat#L7526 |
| MitoProbe™ JC-1 assay kit | Invitrogen | Cat#M34152 |
| CellROX™ reagent | Invitrogen | Cat#C10448 |
| High-Capacity cDNA Reverse Transcription kit | Applied Biosystems | Cat#4368813 |
| NEBNext Ultra™ directional RNA library prep kit | NEB | Cat#E7420 |
| NEBNext Poly(A) mRNA magnetic isolation module | NEB | Cat#E7490 |
| ELISA MAX Deluxe Set Mouse IL-10 | BioLegend | Cat#431414 |
| ELISA MAX Deluxe Set Mouse IL-4 | BioLegend | Cat#431104 |
| ELISA MAX Deluxe Set Mouse IL-1β | BioLegend | Cat#432604 |
| ELISA MAX Deluxe Set Mouse TNF-α | BioLegend | Cat#430904 |
| ELISA MAX Deluxe Set Mouse IL-6 | BioLegend | Cat#431304 |
| Qubit dsDNA HS assay kit | Invitrogen | Cat#Q32851 |
| High-sensitive tape station kit | Agilent | Cat#5067-5585 (reagents) Cat#5067-5584 (screentape) |
| Tn5 Transposase | Zymo | Cat#D5458 |
| **Software** | | |
| ImageJ/Fiji | NIH | https://imagej.nih.gov/ |
| Adobe Illustrator | Adobe | https://www.adobe.com/products/illustrator.html |
| Prism 9.0 | Graphpad | https://www.graphpad.com/ |
| Leica LAS AF | Leica Microsystem | N/A |
| ComplexHeatmap" library | Bioconductor Forum | https://www.bioconductor.org/ |
| DESeq2 version 1.14.1 | Bioconductor Forum | https://www.bioconductor.org/ |

Silica (Millipore-Sigma; 1200 µg/ml); Siramesine fumarate salt (Millipore-Sigma; 200 µM); and D-erythro-sphingosine, free base, High Purity (Millipore-Sigma; 15 µM).

*Inhibitors used in high-content microscopy imaging for this study*
Amlexanox (InvivoGen; 5–200 µM); H-151 (Cayman; 5–20 µM); Bafilomycin A1 (CST; 500 nM); BAPTA/AM (Millipore-Sigma; 20 µM); Bay-11-7085 (Cayman; 5–20 µM); Torin 1 (Invivogen; 1 µM); Dynasore Hydrate (Millipore-Sigma; 50 µM); Cytochalasin D (Millipore-Sigma; 500 nM); Nocodazole (Millipore-Sigma; 2 µM); pp242 (Invivogen; 1 µM); AY 9944 (Cayman; 5–20 µM); Fatostatin (hydrobromide) (Cayman; 10–50 µM); IWP-4 (Cayman; 5–50 µM); ML385 (Millipore-Sigma; 1–5 µM); Epigallocatechin gallate (EGCG) (Millipore-Sigma; 5–200 µM); KU-55933 (Millipore-Sigma; 10–50 µM); GSK690693 (Cayman; 2.5–10 µM); 3-Methyladenine (Millipore-Sigma; 5 mM); Rapamycin (Thermo; 500 nM); Hydroxychloroquine sulfate, HCQ (Millipore-Sigma; 50 µM); Mdivi-1 (Millipore-Sigma; 25 µM); N-acetyl-L-cysteine, NAC (Millipore-Sigma; 1 mM); MitoTEMPO (Millipore-Sigma; 10 µM); carbonyl cyanide m-chlorophenylhydrazone, CCCP (Millipore-Sigma; 10 µM); and, CA-074 methyl ester (Cayman; 5–20 µM).

## Isolation of mouse cardiac fibroblast

The mouse experiments were conducted following approval by the institutional animal ethics committee at the Institute of Life Sciences (ILS), Bhubaneswar, India. The mice were housed in the animal facility at ILS. The chest was opened to expose the heart, which was then perfused using an ice-cold perfusion buffer (1X PBS + 2% FBS) by inserting a 25G needle into the left ventricle through the apex. The heart was excised and rinsed twice in ice-cold PBS. The whole heart was minced into small pieces using sterile scissors in ice-cold 1X PBS. The minced tissue was then transferred into a 15-ml centrifuge tube containing 2 ml of digestion buffer (2 mg/ml collagenase type-IV, 1% FBS + 1% penicillin-streptomycin) and incubated at 37 °C for 10 min, followed by manual trituration 10–12 times with a 5-ml serological pipette. The supernatant was passed through a 40-µm cell strainer into a 50-ml Falcon tube stored on ice. The cells were centrifuged, washed with ice-cold 1X PBS, resuspended in DMEM containing 10% FBS and penicillin-streptomycin, and cultured at 37 °C in a $CO_2$ incubator.

## Primary mouse embryo fibroblast isolation

The primary mouse embryo fibroblasts (MEFs) were obtained from mouse fetuses aged 12.5 to 14.5 days (with the appearance of the copulation plug designated as 0.5 days) from time-mated C57BL/6J female mice. The pregnant female mouse was euthanized by cervical dislocation, and the uterus was harvested. The fetal tissues were finely minced with a scalpel in DPBS and then incubated in 200 µl of 0.25% trypsin/EDTA solution containing 0.1 µg/ml

DNAse I per embryo for 10–15 min in a water bath with gentle agitation. The trypsin was neutralized by adding DMEM containing 10% FBS, and the cell suspension was transferred into 75 cm² culture flasks. The cells were then incubated at 37 °C and 5% CO₂ overnight. The following day, the media were replaced to remove cell debris and unattached tissues.

### siRNA knockdown

The MEF cells were transfected with 30 nM control and gene-specific siRNA using Lipofectamine RNAiMAX transfection reagents (Invitrogen #13778075) as per the manufacturer's instructions.

### Western blotting

The cells were washed with ice-cold PBS and lysed in NP-40 lysis buffer (Invitrogen #FNN0021) containing 1 mM PMSF (Sigma # P7626), phosSTOP (Roche #49068455001), and protease inhibitors cocktail (Roche #1183617000), incubated on ice for 30 min and centrifuged at 12,000 rpm for 30 min at 4 °C. The supernatant was collected, and the protein concentration was evaluated using the BCA Protein Kit (Pierce™ BCA Protein Assay Kit #23225). The lysates were resolved using SDS-PAGE and transferred onto a nitrocellulose membrane (Bio-Rad). The membranes were blocked with 5% skimmed milk for 1 h, followed by incubation with primary antibodies overnight at 4 °C with gentle shaking. Then, the membranes were washed three times with 1X PBS/PBST, followed by incubation with HRP-conjugated secondary antibodies for 1 h at room temperature with gentle shaking. After three 1X PBS/PBST washes, the blots were developed using the enhanced chemiluminescence reagents to visualize the target protein.

Primary antibodies used in Western blotting with dilutions: Anti-pro Caspase-1 + p10 + p12 Antibody (Abcam; 1:1000); Gasdermin D (CST; 1:1000); Phospho-RIP3 (Thr231/Ser232) Antibody (Mouse Specific) (CST; 1:1000); Phospho-MLKL (Ser345) (D6E3G) (CST; 1:1000); RIP3 (Abcam; 1:1000); MLKL (D6W1K) (CST; 1:1000); Anti-beta Actin (Abcam; 1:5000); EP300 (GeneTex; 1:1000); ATF3 (D2Y5W) (CST; 1:1000); Histone H3 (Invitrogen; 1:1000); Anti-LC3B (Millipore-Sigma; 1:1000); Phospho-NF-κB-p65 (Ser536) (93H1) (CST; 1:1000); CBP antibody [C3], C-term (GeneTex; 1:1000); Lamin B1 (D4Q4Z) (CST; 1:1000); Phospho-AMPKα (Thr172) (40H9) (CST; 1:1000); AMPKα (D5A2) (CST; 1:1000); Anti-c-Jun (phospho S73) antibody [E107] (Abcam; 1:1000); Phospho-Akt (Ser473) (CST; 1:1000); Phospho-p38 MAPK (Thr180/Tyr182) (CST; 1:1000); p38 MAPK (CST; 1:1000); Anti-SREBP1 Antibody, clone 2121 (Millipore-Sigma; 1:1000); Caspase-3 (CST; 1:1000); p-Histone H2A.X (Ser 139) (Santa Cruz; 1:1000); p21 Waf1/Cip1 (12D1) (CST; 1:1000); Anti-p16/INK4a, clone EP435Y (Millipore Sigma; 1:1000) and PARP (CST; 1:1000). HRP-conjugated secondary antibodies were purchased from Novus (1:2000).

### Invitro scratch assay

MEF cells were seeded in a six-well plate and exposed to LLOMe for specific time points. The control (untreated) and LLOMe-treated cells were then trypsinized and plated onto six-well plates to form a confluent monolayer. A micropipette tip was used to create a scratch across the width of the well. An EVOS 2000 inverted fluorescence microscope (cell imaging system) at 4X magnification was used to photograph the rate of cell migration in the same scratched region every 2 h until the entire scratch was closed entirely.

### Immunofluorescence assay

The cells were cultured on glass coverslips and treated with LLOMe as desired. The cells were then fixed in 4% paraformaldehyde for 15 min at room temperature. After fixation, the cells were permeabilized with 0.1% Triton X-100 for 15 min and blocked with 1% bovine serum albumin (BSA) for 30 min at room temperature. Subsequently, the cells were incubated with the specified primary antibody for 1 h at room temperature. After three 5-min washes with PBS, the cells were stained with Alexa Fluor-conjugated secondary antibodies for 1 h at room temperature. The coverslips were washed three times with PBS, once with dH₂O, and mounted with Prolong Gold antifade mountant with DNA stain DAPI (Invitrogen #P36931). The cells were visualized using a super-resolution Leica STED SP8 confocal microscope.

For immunofluorescence of floating cells, cells were collected 30 min post LLOMe treatment, centrifuged, resuspended in PBS, and immobilized on glass slides by Cytospin 4 (Thermo Scientific). The cells were then fixed in 4% paraformaldehyde for 15 min at room temperature and processed for staining as described above.

For staining of lysosomes and mitochondria, the cells were stained with 50 nM LysoTracker™ Red DND-99 (Invitrogen #L7528) and 100 nM of MitoTracker™ Red CMXRos (Invitrogen #M7512), respectively, for 10 min at 37 °C prior to treatment with 4 mM LLOMe, followed by fixation in 4% paraformaldehyde at the indicated time points. Coverslips were washed once with dH₂O and mounted with ProLong™ Gold antifade mountant with DAPI. The cells were then imaged by a Leica SP8 confocal microscope.

Primary Antibodies used in immunofluorescence assays with dilutions: ERp72 (D70D12) (CST; 1:100); EEA1 (C45B10) (CST; 1:200); Rab7 (D95F2) (CST; 1:200); RCAS1 (D2B6N) (CST; 1:200); Anti-beta Tubulin (Abcam; 1:500); p-Histone H2A.X (Ser 139) (Santa Cruz; 1:50); Tom20 (F-10) (Santa Cruz; 1:50) and LC3B (MBL; 1:500). Alexa Fluor-conjugated secondary antibodies were purchased from Invitrogen.

### Live cell imaging

The cells were cultured on 6-well plates or 35 mm glass-bottomed dishes (ibidi) and treated with LLOMe as indicated in the figures. The cells were visualized using a ZEISS Celldiscoverer 7 live imaging microscope.

To assess the revival of cells from cell death-like conditions induced by different lysosomotropic agents, the cells were seeded on a six-well plate and treated with the indicated concentration of the reagent. Images were acquired using an EVOS 2000 microscope at 10X and 40X magnifications at specified time points.

For live cell microscopy using a confocal microscope, MEF cells were seeded in 35 mm glass-bottomed dishes (ibidi). The cells were treated with dyes such as 25 nM MitoTracker™ Green FM (Invitrogen #M7514) and 1 μM Rhod-2 AM (Invitrogen) under growth conditions for 30 min prior to treatment with the indicated

concentration of LLOMe. The staining solution was replaced by Fluorobrite DMEM (Gibco) supplemented with 10% FBS. Images were acquired using a Leica SP8 lighting confocal microscope.

For a 6-h time point, MEF cells seeded in 35 mm glass-bottomed dishes (ibidi) were treated with a specified concentration of LLOMe. The cells were stained with the indicated dyes, such as 25 nM MitoTracker™ Green FM (Invitrogen), 1 µM Rhod-2 AM (Invitrogen), 15X Magic Red (Immunochemistry Technologies), 50 nM Lysotracker Red DND-99 (Invitrogen), and 50 nM Lyso-Tracker™ Green DND-26 (Invitrogen) under growth conditions for 30 min. The staining solution was replaced by FluoroBrite DMEM supplemented with 10% FBS, followed by image acquisition by Leica SP8 lighting confocal microscope.

To perform live cell microscopy for Annexin-FITC using a confocal microscope, the cells were treated with the specified concentration of LLOMe. After 30 min, the floating cells collected by centrifugation were resuspended in 100 µL of binding buffer along with 5 µl Annexin V-fluorescein isothiocyanate (ebiosciences #88800574) and the indicated dyes such as 2 mg/mL Hoechst 33342 (Invitrogen #H3570), 4 ng/µL Dil (Invitrogen #D282), 4 µM ethidium homodimer-1 (Invitrogen #L3224), 20 µg/ml PI (ebiosciences #88800574), 75 nM MitoTracker™ Red CMXRos (Invitrogen #M7512) and 0.1 µg/ml DAPI (Invitrogen # D1306) for 5 min at room temperature in the dark. After removal of the binding buffer by centrifugation, the cells were resuspended in fresh DMEM, mounted on slides and imaged using a Leica SP8 confocal microscope. Trypsinized cells processed through the same method served as a control.

## MTT assay

The cells were seeded in a 6-well plate and treated with 4 mM LLOMe for 16 h. The control and LLOMe-treated cells were then trypsinized, and ~2000 cells were seeded in a 25 cm² flask. After 2 days, at each time point, the media was removed, and the cells were washed with PBS and treated with MTT (3-[4,5-dimethylthia-zolyl-2-2,5-diphenyltetrazolium bromide) (5 mg/ml, Sigma) for 2 h at 37 °C. The formazan crystals produced were dissolved in DMSO, and the absorbance was measured at 570 nm.

## High content microscopy imaging

To quantify cell revival after inhibitor and/or LLOMe treatment, the cells were plated in a 96-well plate. The cells were then treated with 4 mM LLOMe and the specified inhibitors for the designated time points. Similarly, to quantify cell revival after siRNA transfection and/or LLOMe treatment, control and siRNA-transfected cells were plated at a density of 6000 cells per well in a 96-well plate. After 24 h of growth, the cells were stained with 2-µm Calcein-AM (Invitrogen #L3224) for 30 min and then imaged using the Cell Insight CX7 LZR high-content screening platform (Thermo Scientific). Automated image scanning and analysis were performed using HCS Studio and iDEV software, respectively. Images were acquired at 10X magnification, capturing 25 fields per well. The number of live cells per well was counted based on intensity and area. All data processing and analysis were computer-based.

## Annexin V/propidium iodide staining

To assess the cell death, the Annexin V/propidium iodide (PI) double staining method was used (eBiosciences). The cells were seeded on a six-well plate and treated with the indicated concentrations of LLOMe. At specified time points, the cells were washed with PBS and trypsinized. Floating cells were collected by centrifugation of the culture supernatant 30 min after LLOMe treatment. The cells were then resuspended at a density of $1 \times 10^5$ cells/ml in 100 µL of binding buffer along with 5 µl Annexin V-fluorescein isothiocyanate (FITC) or Allophycocyanin (APC) and 5 µL Propidium Iodide (PI) for 30 min at room temperature. The cells were analyzed by FACS Fortessa (Beckton and Dickinson), and the results were analyzed using Cell Quest Pro software/FlowJo software.

## JC-1 and CellRox staining

MitoProbe™ JC-1 assay kit (Invitrogen) was used to assess mitochondrial membrane potential/polarization. Oxidative stress in live cells was measured using the CellROX™ reagent (Invitrogen #C10448). The cells were incubated with 10 µL of 200 µM JC-1 dye or 5 µM of CellRox in 1X PBS at 37 °C, 5% $CO_2$ for 30 min, and then washed and resuspended in 1X PBS. The cells were analyzed by FACS Fortessa (Beckton and Dickinson), and the results were analyzed using Cell Quest Pro software/FlowJo software.

## RNA isolation and quantitative real-time PCR

Total RNA was extracted from cells using TRIzol™ reagent (Invitrogen) according to the manufacturer's protocol. 1 µg of RNA was reverse transcribed using the High-Capacity cDNA Reverse Transcription kit (Applied Biosystems). The resulting cDNA was used for qRT-PCR using PowerUp™ SYBR™ Green Master Mix for qPCR (Applied Biosystems) according to the manufacturer's instructions. The fold-change in gene expression was determined using the $2^{-\Delta\Delta Ct}$ method, with normalization to β-Actin as a loading control. The graphs were generated using the Graph Pad software.

## Sample preparation for RNA sequencing

The total RNA was isolated from cells for the indicated time points (three biological replicates) using the RNeasy mini kit. The quality and quantity of total RNA were assessed using agarose gel and Qubit 3.0, respectively. Following RNA quality assessment, 800–900 ng of total RNA was used for library preparation with the NEBNext Ultra™ directional RNA library prep kit, along with the NEBNext Poly(A) mRNA magnetic isolation module. The library was quantified using the Qubit dsDNA HS assay kit (Invitrogen), and its quality and fragment length distribution were evaluated using the high-sensitive tape station kit (Agilent 2200). Subsequently, the library was sequenced on the HiSeq 400 Illumina platform.

## RNA-seq data processing and gene expression analysis

The paired-end (PE) reads quality checks for each sample were conducted using FastQC v.0.11.5 (http://www.bioinformatics.babraham.ac.uk/projects/fastqc/). Adapter sequences were trimmed using the "bbduk" script. STAR (v.2.5.3a) was used for sequence alignment to the mouse genome build GRCm38.p6genome.fa and Gencode.vM18.annotation.gtf for

annotation(Dobin et al, 2013). A count matrix was generated for differential gene analysis using feature Counts from the subread-1.5.3 package with Q = 10 for mapping quality, and these count files were used as input for downstream differential gene expression analysis using the DESeq2 version 1.14.1(Love et al, 2014). Genes with read counts of ≤ 10 in any comparison were removed, followed by differential analysis of the given comparison using the DESeq2 function library. "*p*" values were adjusted using the Benjamini and Hochberg multiple testing correction. Significantly differentially expressed genes were identified based on a fold-change of 1.5-fold or greater (up- or downregulated) and a *p* value <0.05. Unsupervised hierarchical clustering was performed, and the heatmap was plotted using the "ComplexHeatmap" library (https://www.bioconductor.org/), where the gene expression matrix was transformed into z-score.

## ATAC-seq chromatin assay

To generate ATAC-seq libraries, 50,000 cells from both control and LLOMe-treated groups were used, and the libraries were constructed as previously described (Buenrostro et al, 2013). Briefly, cells were washed twice in PBS, counted, and nuclei were isolated using 50 μl of hypotonic buffer (10 mM Tris, pH 7.4; 10 mM NaCl; 3 mM $MgCl_2$; 0.1% NP-40). The nuclei were collected by centrifugation, resuspended in tagmentation buffer, and treated with 2.5 μl of Tn5 Transposase (Zymo, Catalog No: D5458) for 30 min at 37 °C. DNA from the transposed nuclei was then isolated and PCR-amplified using barcoded Nextera primers. Library quality control was performed using a high-sensitivity TapeStation and Qubit measurements, followed by paired-end sequencing on the Illumina NovaSeq X platform.

## ATAC-seq data analysis

ATAC-seq reads were processed using nfcore/atacseq v2.1.2 with standard settings. Reads were trimmed using trim-galore v0.6.7, aligned to the mm10 genome using BWA MEM v0.7.17, and duplicated reads were marked with Picard MarkDuplicates v3.0.0. Alignment was filtered using samtools v1.17, thereby removing multimapping and duplicated reads. Peaks were called from the filtered and sorted BAM using MACS2 v2.2.7.1 with parameters –Narrow–BAMPE–keep-dup all–nomodel. Consensus peaks were obtained by merging peak calls using bedtools v2.8. Heatmaps and metaplots were generated using deepTools version 3.5.6. DNA sequence motif analysis in ATAC-seq peaks was performed using HOMER version homer/v5.1, 7-16-2024.

## Nanopore library preparation and methylation sequencing

Genomic DNA was isolated from the MEF cells. End repair and dA tailing were performed on gDNA, and libraries compatible with Oxford Nanopore sequencing were prepared using the ligation sequencing kit LSK114 following the manufacturer's instructions. About 40 fmol of the library was loaded per flow cell on PRO-114M and run for ~72 h, targeting 80 Gb of data per sample.

Data quality was assessed using NanoStat. Raw Oxford Nanopore signals in pod5 format were basecalled using dorado v0.4.2 in hac mode, passing the parameter "--modified-bases 5mCG_5hmCG" to enable methylation calling. The GRCm39

reference genome was also provided to dorado to enable alignment and produce a bam file containing reference alignments and read wise methylation information. Location-wise consensus methylation levels were derived from the bam files using modbam2bed. Only those cytosines which were covered by at least ten reads in all the four samples were considered for subsequent analysis. Genome-wide methylation analysis and data visualization were performed using custom Python scripts.

## Tail regeneration in *Polypedates maculatus* tadpole

Indian tree frog *Polypedates maculatus* were collected around the campus of Utkal University, Bhubaneshwar (200 21′N 850 53′ E), Odisha, India. Tadpoles were reared and maintained as described previously(Mahapatra et al, 2023; Mahapatra and Mohanty-Hejmadi, 1994). Tadpoles at Gosner stage 26 were used for the study. The tadpoles were anesthetized with MS 222 (Tricaine methanesulfonate) (6 g/L) for 1–2 min before tail amputation. Tail amputation was performed by keeping the specimens laterally on a pre-sterilized porcelain plate and cutting the larva in the middle of the tail using a sterile blade. The morphometric (length) measurements of various stages were taken through graph paper with 10 divisions per centimeter (1 division = 1 mM). The day of amputation was counted as day 0. The amputated tadpoles were divided into groups and treated with LLOMe or inhibitors for 1 h by placing the tadpoles in water.

For the tail regeneration experiment, amputated tadpoles (*n* = 15 tadpoles per group) were treated with different concentrations of LLOMe on days 0, 1, 3, and 5. The length was measured every 2 days until day 9 and then every 3 days until day 15.

For inhibitor experiments, the tadpoles (*n* = 10 tadpoles per group) were initially exposed to inhibitors (Bay-11-7085, pp242, Dynasore Hydrate, and BAPTA-AM) for 1 h by immersing them in water containing specific inhibitors. The tadpoles were then removed and placed in fresh water for 2 h to recover. After the recovery period, tail amputation was performed, followed by treatment with LLOMe on the same day (Day 0). Amputated tadpoles were treated with the inhibitor on days 0, 1, and 3, and with the LLOMe on days 0, 2, and 4.

## CellRox, Lysotracker, and Rhod2AM staining in the cells isolated from tadpole tissue

Posterior cells formed after the amputation were collected (5–7 tadpoles per group) and dissociated into single cells by washing with calcium-magnesium-free 1X PBS. The tissue samples were then incubated with 0.25% trypsin-EDTA for 5 min at room temperature, and the reaction was stopped by diluting with 1X PBS. Physical dispersion was achieved by trituration with a pipette during and after trypsinization, followed by passing through a 70-μm cell strainer. The cells were then centrifuged at 300 × *g* for 5 min, resuspended in PBS, and stained with 5 μM CellRox (30 min) or 50 nM lysotracker (30 min) or 1 μM Rhod2AM (15 min) for flow cytometry analysis.

## Neural regeneration in *Caenorhabditis elegans*

The experiments were conducted using the *Caenorhabditis elegans* muIS32 [Pmec7::GFP] background as described previously (Kumar

et al, 2021). Axotomy was performed on day 3 (A3) animals. One of the PLM neuron axons was axotomized at 50–60 μm from the cell body using a pulsed UV laser (Andor, Oxford Instruments) coupled to a Nikon Ti2 Eclipse microscope under a 100x oil objective (NA = 1.4). Axotomized worms were allowed to recover for 3 h followed by measurement of PTRI on both sides. Then, the worms were transferred to the LLOMe-treated plates and control OP50-seeded plates. After 24 h of axotomy (posterior touch response index), PTRI was measured. The PTRI was calculated based on the number of positive responses from the ten touches. The recovery index was calculated by dividing PTRI at 24 h by PTRI at 3 h (Kumar et al, 2021). Subsequently, the worms were phenotyped using a Nikon Ti2 Eclipse fluorescence microscope to identify the axotomized PLM side.

## Wound healing in mice

C57BL/6 J mice ($n = 14$–15 mice per group) were fed standard chow diets, and all efforts were made to minimize animal suffering. Hair was removed from the mice dorsum using hair removal cream and the surgical site was disinfected with skin disinfectant (70% alcohol). The wound was generated according to previously described methods (Wang et al, 2013). Briefly, mice were anesthetized with a single intraperitoneal injection of ketamine (80 mg/kg body weight)/xylazine (16 mg/kg body weight) diluted in 100 μl of PBS (phosphate-buffered saline). Full-thickness (i.e., epidermis, dermis, and subcutis) wound was induced on the dorsal skin by using a sterile punch biopsy needle (6-mm diameter). After wound creation (day 0), the animal was placed into a warm chamber at 37 °C for its recovery from anesthesia. The wound was treated daily twice (10–12 h intervals) with LLOMe (4 and 8 mM) through topical administration for a minimum of 10–15 min each up to day 9. All mice showed no side effects resulting from LLOMe treatment, such as infection or mortality during the experiments. Every day, pictures of the wound were taken with a digital camera and calculated the area by tracing the wound margins and evaluated as a percent area of the initial wound using ImageJ software.

## Corneal regeneration in mice

The institutional Animal Ethics Care Committee approved the study at the Centre for Cellular and Molecular Biology (CCMB) in Hyderabad, India. The experimental mice (C57BL/6J) of 6–8 weeks were divided into three groups: (A) Control group without injury. (B) Injured Control injured without LLOMe treatment. (C) 4 mM LLOMe-treated group. The eye of each animal, before surgery, or the right eye, served as a negative/normal control group. Briefly, the mice were anesthetized intramuscularly with a combination of Ketamine and Xylazine followed by 1–2 drops of sterile topical 0.5% Proparacaine Hydrochloride ophthalmic solution was applied to the eye in each animal to anesthetize the eye before the surgery. Corneal epithelium damage was achieved in the left eye of the mouse by gentle scraping of the corneal epithelium using a surgical scalpel No.15, and the alkali burn was made by topical, single drop application of 0.1 N sodium hydroxide (NaOH) for 30 s, and the eye was flushed with normal saline. Epithelial damage was identified by staining the eye with fluorescein sodium ophthalmic strips, blotting paper with orange color fluorescein stain (Care

Group, Vadodara, India), and illuminated with cobalt-blue light. The positively stained area of the cornea was measured using ImageJ software (NIH, USA). LLOMe was administered topically two times a day with a 6-h gap for 7 days in the respective groups, and the follow-up continued for days 2, 5, and 7, respectively.

## MEF injection in C57BL/6J mice

Female C57BL/6J mice aged 6 to 8 weeks were shaved and injected subcutaneously on the right and left flank with $1 \times 10^6$ MEF control cells ($n = 3$ mice) and 4 mM LLOMe-treated MEF recovered cells ($n = 3$ mice) resuspended in phosphate-buffered saline (PBS), mixed in a 1:1 volumetric ratio with matrigel (BD Corning) for a final injection volume of 100 μL. The growth was monitored every other day, followed by image acquisition on day 15 post injection with a digital camera.

## IncuCyte analysis

MEF cells were seeded into the six-well plate for overnight incubation. The cells were treated with LLOMe (at varying concentrations), D-erythro-Sphingosine, and GPN, and the live cell images were captured using the IncuCyte S3 live-cell analysis instrument (Sartorius). For the inhibitor assay, MEF cells were pre-treated with Bay-11-7085 and BAPTA/AM, followed by 4 mM LLOMe treatment for 24 h.

For measuring oxidative stress or lysosomes, MEF cells seeded in a six-well plate were incubated with 5 μM of CellRox reagent (Invitrogen #C10448) or 10 nM LysoTracker™ Red DND-99 (Invitrogen #L7528) for 30 min at 37 °C and 5% $CO_2$, respectively. Then, the cells were treated with 4 mM LLOMe in fresh medium for 24 h in the IncuCyte S3 Live-Cell Analysis instrument (Sartorius). Cells were imaged at 20X magnification every 10 or 15 min and the red object count/well was quantified using IncuCyte S3 software.

For revival assays, MEF cells were seeded in a six-well plate with 2 μM ethidium homodimer-1 (Invitrogen #L3224) or 20 nM Sytox Green (Thermo Fisher Scientific, S7020) for 30 min at 37 °C and 5% $CO_2$. The cells were treated with 4 mM LLOMe in fresh medium for 24 h in the IncuCyte S3 Live-Cell analysis instrument (Sartorius). Cells were imaged at 20X magnification every 10 or 15 min, and the red or green object count/well was quantified using IncuCyte S3 software. All the data were plotted using GraphPad Prism software.

## *Drosophila* experiments

All the *Drosophila* stocks and crosses were maintained at 25 °C, in a standard cornmeal diet. The fly stocks used were Canton S, tep4-Gal4, UAS-GFP, and 10XSTAT-GFP. Canton S was used as a control strain, and Canton S flies were crossed to tep4-Gal4, UAS-GFP, or 10XSTAT-GFP, and the first-generation progeny larvae were used for the studies.

The early third instar larvae were collected and transferred to a vial containing distilled water and starved for 2 h. The larvae were then transferred to a vial containing the Whatman filter paper disk on which 150 μL of 8 mM final concentration of the LLOMe was added. The larvae were treated for 14 h with LLOMe. For control, 150 μL nuclease-free water (NFW) was added to the Whatman filter paper disk.

LLOMe or control-treated third instar larvae were used for lymph gland (LG) dissections. The dissections were performed in phosphate buffer saline (PBS), fixed in 4% paraformaldehyde, followed by washes with PBS containing 0.3% Triton X (PBST). The samples were then blocked in 20% normal goat serum for 20 min at room temperature, followed by overnight primary antibody [rabbit anti-H3P (1:200, 06-570 – EMD Millipore)]. incubation at 4 °C. This was followed by PBST washes, blocking, and treatment with appropriate Alexa Fluor-conjugated secondary antibody incubation for two hours at room temperature. The LGs were then mounted in a Vectashield mounting medium containing DAPI (Vector Laboratories, RRID: AB_2336790).

Confocal images were captured using Zeiss LSM 780. The Z projection of the confocal images was used for estimating various lymph gland parameters using ImageJ/Fiji software. The Blood Progenitor Index was estimated by measuring the percentile of GFP-positive area divided by the total area of the primary lobe. A freehand selection tool was used for measuring the area of the Blood Progenitor Index. For the quantification of PH3, the positive signals for the marker were manually counted using the multipoint tool. The lymph gland quantifications were done for individual primary lymph gland lobes.

### MEF replating assay

MEF cells were seeded in six-well plates. The next day, the cells were treated with 4 mM LLOMe and the floating cells were collected by centrifugation (30 min post treatment). The floating cells were plated in complete DMEM media, followed by image acquisition by EVOS 2000 microscope at 10X and 40X magnification at the indicated time points.

### Enzyme-linked immunosorbent assay (ELISA)

ELISA was performed using the ELISA Max™ Deluxe Set Mouse (BioLegend) kit according to the manufacturer's instructions. Briefly, supernatants from MEF cells treated with LLOMe (4 mM) were collected at 0 min, 30 min, 3 h, and 30 h post treatment. Polystyrene 96-well plates were precoated overnight at 4 °C with a specific capture antibody, followed by blocking with blocking buffer for 1 h at room temperature (RT). Standard cytokine dilutions and cell culture supernatants were then added and incubated overnight at 4 °C. Plates were washed with PBS supplemented with 0.05% Tween-20, then incubated with biotinylated detection antibody for 1 h at RT. After washing, the plates were incubated with HRP-Avidin for 30 min at RT, followed by another wash. The signal was developed by adding TMB (3,3′,5,5′-Tetramethylbenzidine) until color appeared. The reaction was stopped by adding 1 M $H_3PO_4$, and absorbance was measured at 450 nm using a microplate reader.

### Comet assay

MEF cells seeded in a six-well plate were treated with repeated cycles (7 cycles) of 4 mM LLOMe every 24 h. Control cells, trypsinized for six cycles, served as the control. The control and LLOMe-treated cells were trypsinized, and $1 \times 10^5$ cells were mixed with molten LM Agarose (at 37 °C) at a ratio of 1:10 (v/v) and applied to slides. The slides were covered by coverslips and kept at 4 °C in the dark for 1 h. The coverslips were removed and the slides

were placed in the alkaline lysis solution (2.5 M NaCl, 100 mM disodium EDTA, 10 mM Tris base, 200 mM NaOH, 1% DMSO, 1% Triton X-100, pH 10) at 4 °C for 1 h. After lysis, the slides were kept in alkaline solution (200 mM NaOH, 1 mM disodium EDTA, pH >13) for 30–60 min, followed by electrophoresis at 21 volts, 0.3 A for 20 min at 4 °C in the dark. Samples were neutralized 1–3X times with 0.4 M Tris at pH 7.5. The slides were then stained with propidium iodide (Invitrogen; P3566). Images were acquired using an EVOS 2000 microscope at 10X magnification.

### β-Galactosidase (β-Gal) assay

MEF cells seeded in a six-well plate were treated with repeated cycles (7 cycles) of 4 mM LLOMe every 24 h. Control cells, trypsinized for six cycles, served as the control. Following treatment, both control and LLOMe-treated cells were fixed and stained using the Senescence β-Galactosidase Staining Kit (Cell Signaling Technology, #9860) according to the manufacturer's instructions. Images were acquired using an EVOS 2000 microscope at 10× and 40× magnifications.

### Mouse bone marrow cells isolation and differentiation

Bone marrow cells were isolated from mice and differentiated into macrophages using a standard procedure. Briefly, 6 to 8-week-old C57BL/6 mice were sacrificed by cervical dislocation. Bone marrow cells were flushed from the tibiae and femurs into RPMI medium. Red blood cells were lysed using RBC lysis buffer containing 155 mM $NH_4Cl$, 12 mM $NaHCO_3$, and 0.1 mM EDTA. The cells were cultured in RPMI medium supplemented with 10% FBS, 1 mM sodium pyruvate, and 0.05 M 2-mercaptoethanol, along with 20 ng/ml Mouse M-CSF (GIBCO PMC2044) for 5 days. Medium was replaced every other day with fresh M-CSF-containing medium.

### Statistical analysis

Statistical analysis was performed using the GraphPad Prism Version 9 software.

## Data availability

The RNA-seq data [MEF cells treated with LLOMe (C1 to C6)] from this publication have been deposited to the Annotare ArrayExpress database at EMBL-EBI (https://www.ebi.ac.uk/biostudies/ArrayExpress/studies/E-MTAB-14641) and assigned the accession number E-MTAB-14641. The RNAseq data [MEF cells treated with LLOMe (C1, C4, and extended-C6)] from this publication have been deposited in the Annotare ArrayExpress database at EMBL-EBI (https://www.ebi.ac.uk/biostudies/ArrayExpress/studies/E-MTAB-15207) and assigned the accession number E-MTAB-15207. The ATAC sequencing data from this publication have been deposited in the Annotare ArrayExpress database at EMBL-EBI (https://www.ebi.ac.uk/biostudies/ArrayExpress/studies/E-MTAB-14658) and assigned the accession number E-MTAB-14658. The Methyl sequencing data from this publication have been deposited to the have been deposited to the GEO database (https://www.ncbi.nlm.nih.gov/geo/query/acc.cgi?acc=GSE285446) and assigned the accession number GSE285446.

The source data of this paper are collected in the following database record: biostudies:S-SCDT-10_1038-S44318-025-00540-y.

## Peer review information

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

## Acknowledgements

SC gratefully acknowledges funding from ICMR (IIRP-2023-1988, IIRPIG-2023-0814, ICMRCAREP-2023-0000176, and DHR-GIA/2020-NCD-1), ANRF (CRG/2022/002777), CSIR (NCP070301), and DBT (BT/PR45223/ MED/29/1613/2022, BT/PR41956/MED/97/515/2021, and BT/PR23942/ BRB/10/1808/2019). AGR gratefully acknowledges the funding from the India Alliance DBT Wellcome Senior fellowship (Grant #IA/S/22/1/506243). RJK gratefully acknowledges DBT Ramalingaswami's Re-entry fellowship. We gratefully acknowledge the support of the CSIR-CCMB and BRIC-ILS Central Facilities (BSL2, animal house, microscopy, and NGS) funded by the Council of Scientific and Industrial Research (CSIR) and Department of Biotechnology (India). We acknowledge the excellent technical assistance of Suman Bandari (CSIR-CCMB) and Bhabani Sankar Sahoo (BRIC-ILS) for the microscopy experiments. We acknowledge the excellent assistance of B Jyothi Lakshmi, N Sai Ram, and S Prashanth for the animal experiments at CSIR-CCMB. A part of this work is filed for an Indian patent (Patent Application: 0135NF2024) and PCT (PCT/IN2025/051017).

## Author contributions

**Kollori Dhar**: Data curation; Validation; Investigation. **Kautilya Kumar Jena**: Conceptualization; Data curation; Validation; Methodology. **Subhash Mehto**: Data curation; Writing—review and editing. **Rinku Sahu**: Data curation. **Krushna C Murmu**: Data curation. **Atharva Anand Mahajan**: Data curation. **Sibaram Behera**: Data curation. **Ravi Kiran Putchala**: Data curation. **Reuben Jacob Mathew**: Formal analysis. **Ramyasingh Bal**: Data curation. **Soumya Kundu**: Data curation. **Santosh Kumar Das**: Data curation. **Swati Chauhan**: Data curation. **Sameekshya Satapathy**: Data curation. **Rina Yadav**: Data curation. **Swatismita Priyadarsini**: Data curation. **Khyathi Ratna Padala**: Data curation. **Prashanth Namdigalla**: Data curation. **Sanchita Mishra**: Data curation. **Prerana Muralidhara**: Data curation. **Kushagra Bansal**: Supervision. **Kesavardhana Sannula**: Supervision. **Punit Prasad**: Supervision. **Kiran Kumar Bokara**: Supervision. **Divya Tej Sowpati**: Supervision. **Anindya Ghosh-Roy**: Supervision. **Pravati Kumari Mahapatra**: Supervision. **Rohan Jayant Khadilkar**: Supervision. **Ramesh Yelagandula**: Supervision. **Santosh Chauhan**: Conceptualization; Resources; Data curation; Formal analysis; Supervision; Funding acquisition; Validation; Investigation; Visualization; Methodology; Writing—original draft; Project administration; Writing—review and editing.

Source data underlying figure panels in this paper may have individual authorship assigned. Where available, figure panel/source data authorship is listed in the following database record: biostudies:S-SCDT-10_1038-S44318-025-00540-y.

## Disclosure and competing interests statement

The authors declare no competing interests.

# Expanded View Figures

**Figure EV1. Cells resuscitate from cell death-like conditions.**

(A) Snapshots of time-lapse live microscopy images of MEF cells treated with 4 mM LLOMe. Magnification 10X. Scale bar, 400 μm. (B) MEF cells were treated with 4 mM LLOMe. The floating cells were collected and washed with PBS before plating in a new dish. Snapshot images of live microscopy of MEF cells after replating. Refer to Movie EV3. Dotted circles represent the revival of the indicated floating cells at the indicated time points. (C) The graph depicts the quantification of the percentage of the well covered by the object (area of the well covered by cells) from time-lapse live microscopy. (D) Representative time-lapse live microscopy images of MEF cells treated with GPN (200 μM), sphingosine (15 μM), LLOMe (4 mM), Siramesine (200 μM), Alum (1 mg/ml), and Silica (1200 μg/ml). Magnification 10X. Scale bar, 400 μm. (E) Representative time-lapse live microscopy images of different cell lines treated with LLOMe. (F) The table depicts the cell lines treated with LLOMe, the concentrations of LLOMe used, and the morphological changes. Magnification 40X. Scale bar, 100 μm. (G) Representative time-lapse live microscopy images of BMDMs cells treated with 0.25 and 0.5 mM LLOMe. Magnification 40X. Scale bar, 75 μm. Source data are available online for this figure.

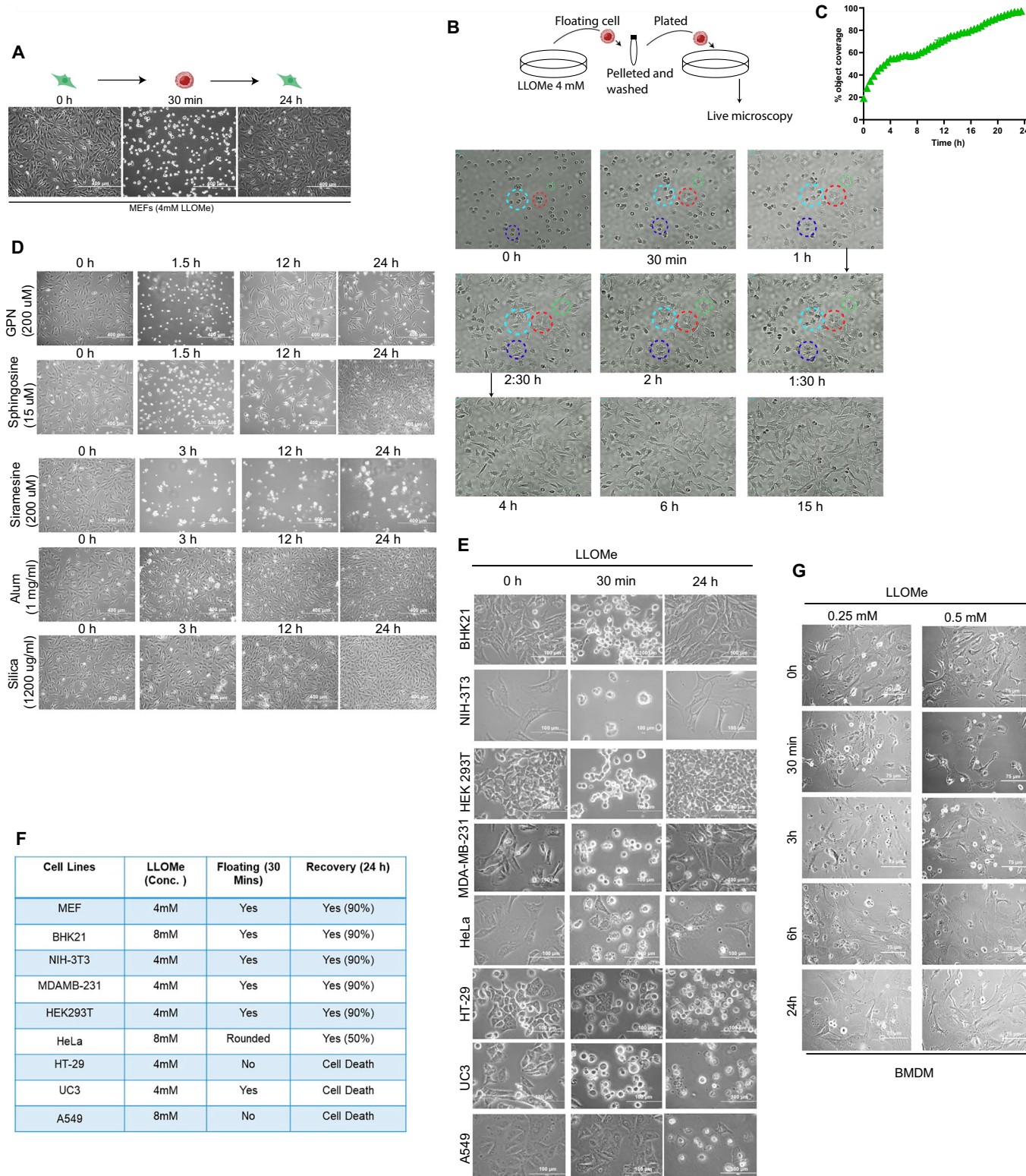

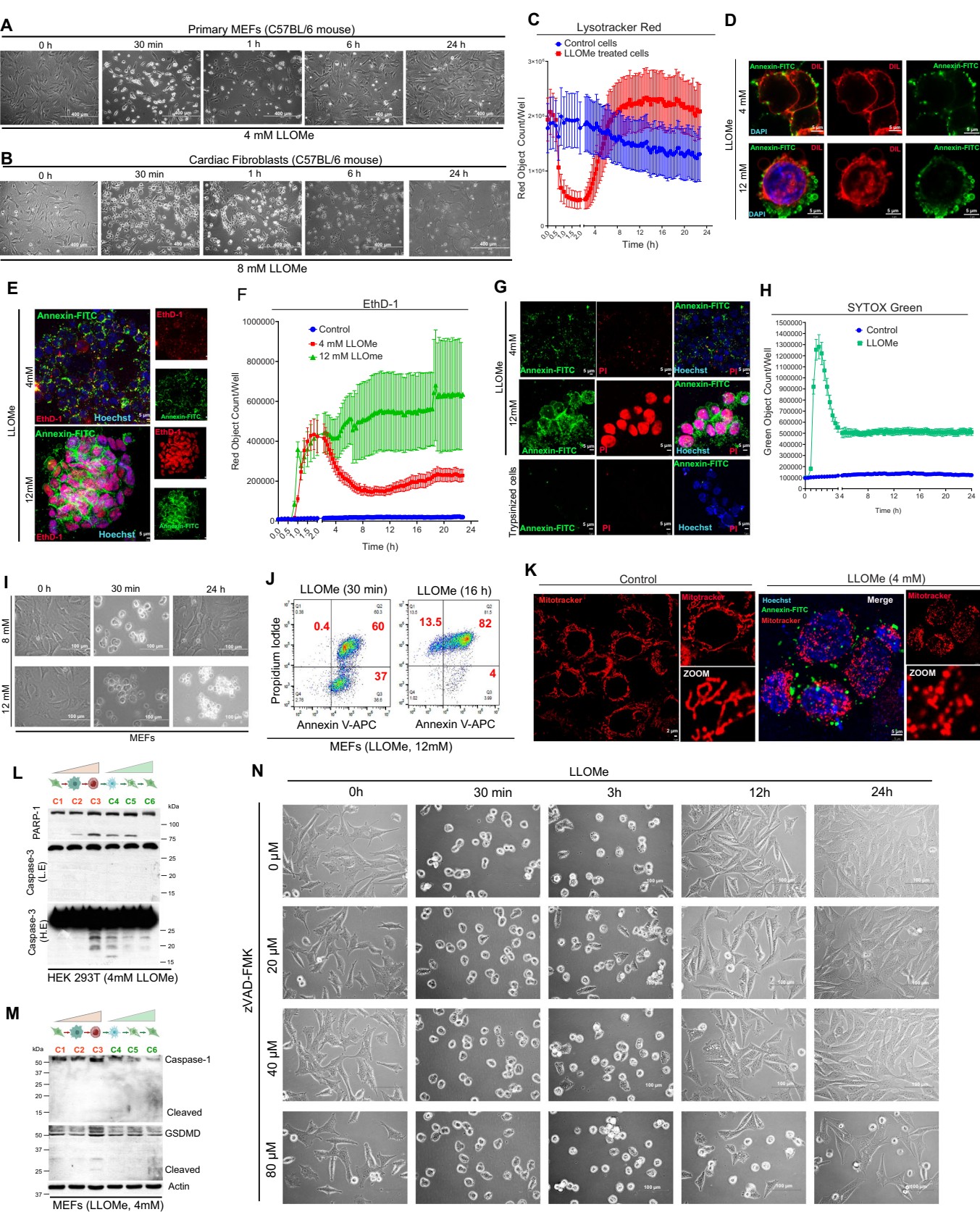

◀ **Figure EV2. Revival from near-cell death.**

(A, B) Representative time-lapse live microscopy images of (A) primary MEF (C57BL/6 mouse) cells treated with 4 mM LLOMe and (B) primary cardiac fibroblast (C57BL/6 mouse) cells treated with 8 mM LLOMe. Magnification 10X. Scale Bar, 400 μm. (C) Time-lapse live microscopy of MEF cells stained with Lysotracker red (10 nM), washed, and then treated with 4 mM LLOMe, performed using IncuCyte S3. Graphs depict quantification of red objects (lysosomes-stained cells) per well, Mean ± SD, nine fields. (D, E) Representative live cell confocal microscopy images of MEF cells treated with LLOMe (4 or 12 mM) for 30 min and stained with (D) Annexin V-FITC (green), DIL (red) and DAPI (blue) (E) Annexin V-FITC (green), ethidium homodimer-1 (EthD-1) (red) and Hoechst (blue). Scale Bar, 5 μm. (F) IncuCyte time-lapse live microscopy of MEF cells stained with 2 μM EthD-1, washed and then treated with 4 mM LLOMe. Graphs depict quantification of red objects (lysosomes-stained cells) per well, mean ± SD, nine fields. Refer to Movie EV8. (G) Representative live cell confocal microscopy of MEF cells treated with LLOMe (4 or 12 mM) for 30 min and stained with Annexin V-FITC (green), and Propidium Iodide, PI (Red). Trypsinized floating cells were used as a control. Scale Bar, 5 μm. (H) Time-lapse live microscopy of MEF cells stained with 20 nM SYTOX Green and then treated with 4 mM LLOMe, performed using IncuCyte S3. Graphs depict quantification of green objects per well, mean ± SD, nine fields. Refer to Movie EV9. (I) Representative time-lapse live microscopy images of MEF cells treated with LLOMe (8 or 12 mM). Magnification 40X. Scale bar, 100 μm. (J) Representative flow cytometry analysis of MEF cells treated with 12 mM LLOMe for 30 min or 16 h and stained with Annexin V-APC and Propidium Iodide (PI). (K) Representative live cell confocal microscopy images of MEF cells, untreated or treated with 4 mM LLOMe for 30 min and stained with Annexin V-FITC (green), Mitotracker (red) and Hoechst (blue) as indicated. Scale Bar, 2 or 5 μm. Zoom panels show digital magnifications. (L, M) Western blot analysis with the lysate of (L) HEK293T cells with Caspase-3 and PARP-1, and (M) MEF cells with Caspase-1 and GSDMD antibodies at different time points. (N) Representative time-lapse live microscopy images of zVAD-FMK-treated MEF cells subjected to LLOMe (4 mM) treatment. Magnification 40X. Scale bar, 100 μm. Source data are available online for this figure.

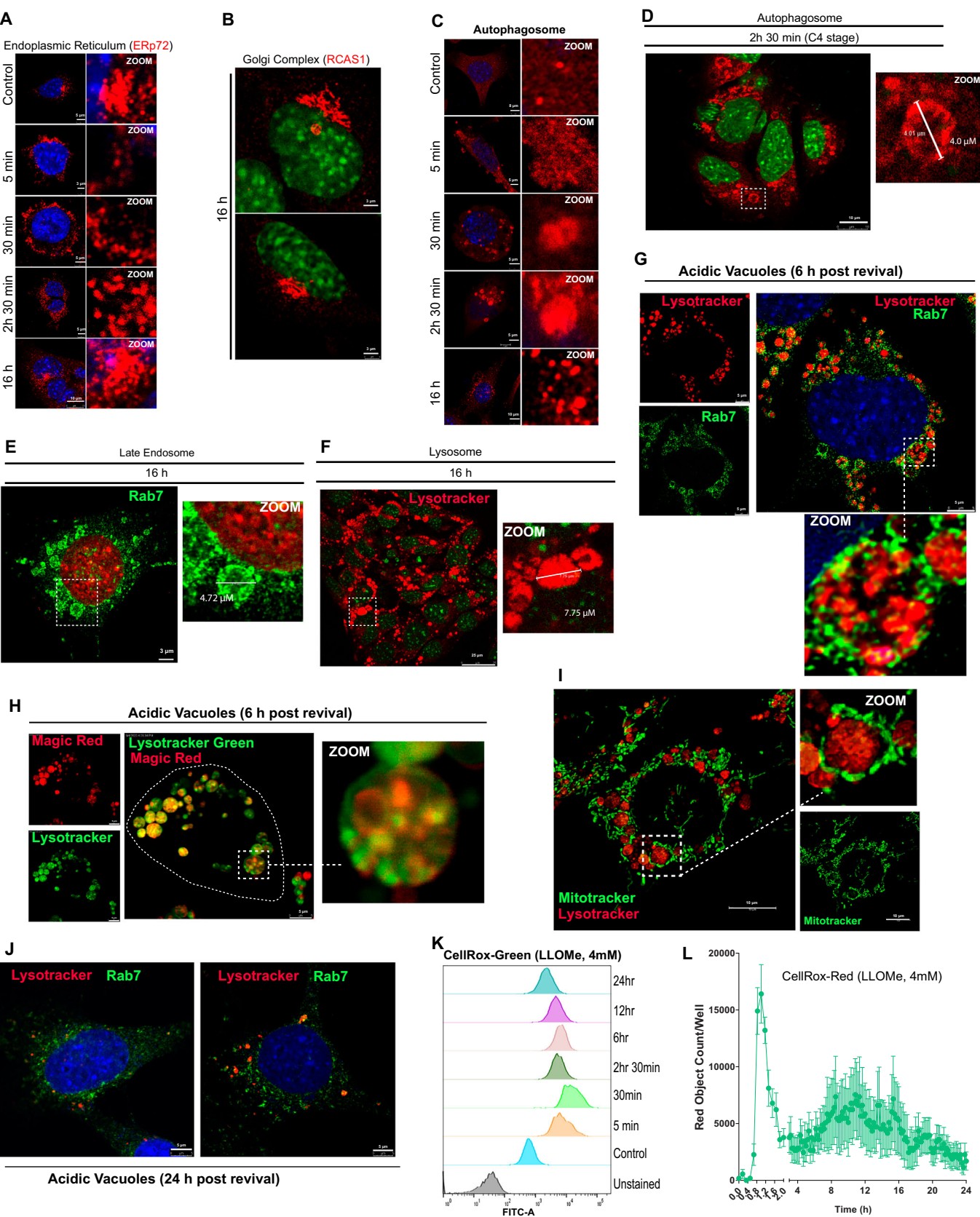

◄ **Figure EV3.** **Organelle dynamics during the cell death phase and revival.**

(A–C) Representative confocal microscopy images of MEF cells treated with 4 mM LLOMe for the indicated time points and immunostained with (A) ERp72 (endoplasmic reticulum), (B) RCAS1 (Golgi complex), and (C) LC3B (autophagosome). The nucleus is stained with DAPI (pseudo-colored to green in (B) for better contrast). Zoom panels show digital magnifications. Scale Bar, 3–10 μm as indicated. (D) Representative confocal microscopy images of MEF cells treated with 4 mM LLOMe for 2 h 30 min (C4 stage) and immunostained with LC3B (autophagosome). The nucleus is stained with DAPI (pseudo-colored to green for better contrast). Zoom panels show digital magnifications. Scale Bar, 10 μm. (E, F) Representative confocal microscopy images of MEF cells treated with 4 mM LLOMe for 16 h and immunostained with (E) Rab7 or stained with (F) lysotracker red. The nucleus is stained with DAPI (pseudo-colored to red/green for better contrast). Zoom panels show digital magnifications. Scale Bar, 3 or 25 μm as indicated. (G–I) Representative confocal microscopy images of MEF cells treated with 4 mM LLOMe for 6 h and immunostained with (G) Rab7 (late endosomes) and stained with lysotracker red or (H) stained with magic red and lysotracker green or (I) stained with mitotracker green and lysotracker red. Zoom panels show digital magnifications. Scale Bar, 5 or 10 μm as indicated. (J) Representative confocal microscopy images of MEF cells treated with 4 mM LLOMe for 24 h post revival and stained with lysotracker red and immunostained with anti-Rab7 (late endosomes). The nucleus is stained with DAPI. Scale Bar, 5 μm. (K) Flow cytometry analysis of ROS production using CellROX Green in MEF cells treated with 4 mM LLOMe at the indicated time points. (L) Time-lapse live microscopy using IncuCyte of MEF cells stained with CellROX-red and then treated with 4 mM LLOMe. Graphs depict quantification of red object count per well, mean ± SD, nine fields. Source data are available online for this figure.

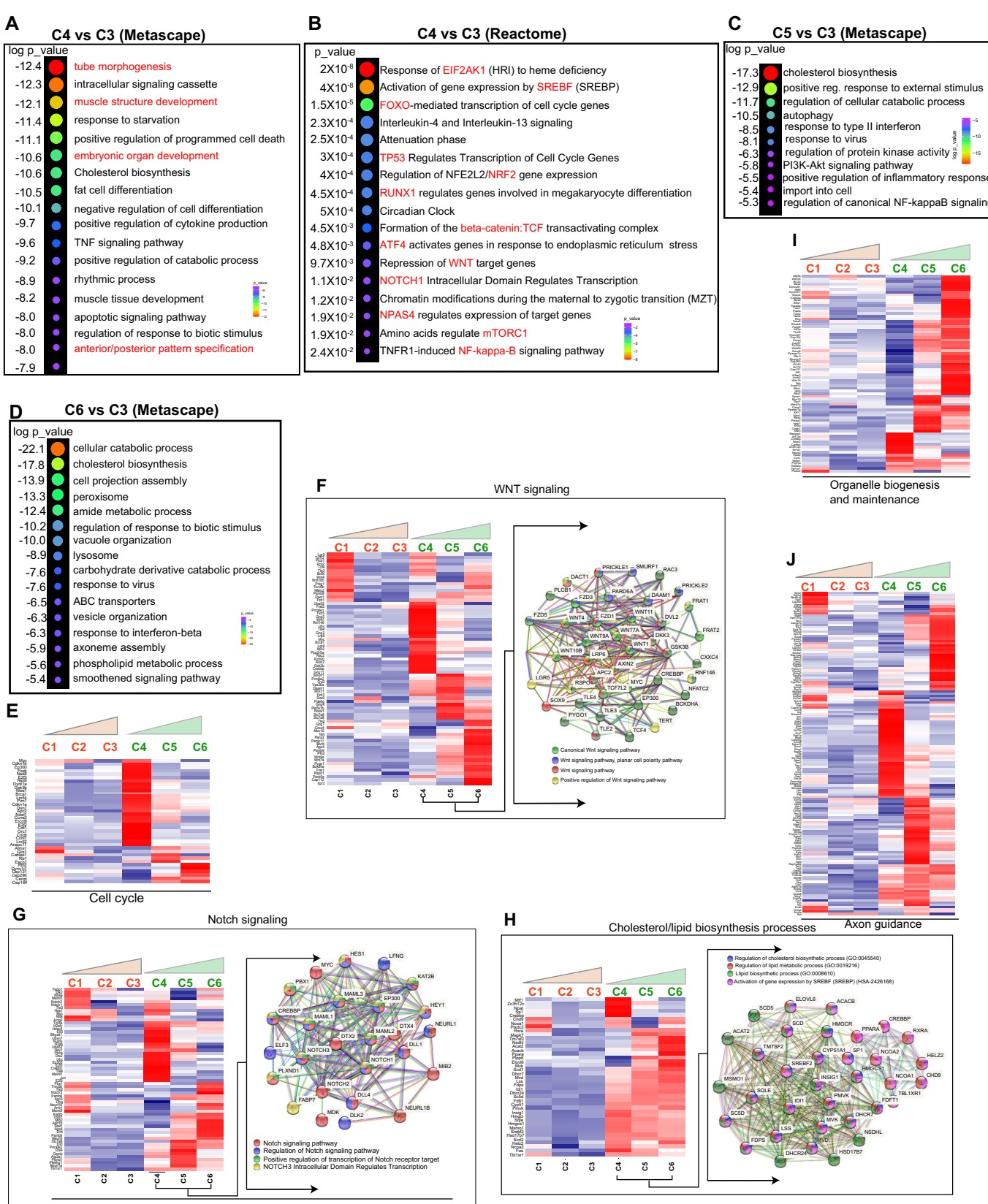

◀ **Figure EV4.  RNA-sequencing analysis of cell death and revival phase.**

(A–D) Bubble plots depict top pathways from metascape (**A**, **C**, **D**) and Reactome (**B**) pathway analysis performed with genes upregulated ($p_{adj} < 0.05$, >1.5 folds, $n = 3$, Wald chi-squared test) in C4 or C5 or C6 stage as compared to C3 stage. (**E–J**) Heatmap generated for genes representing GO terms (**E**) Cell Cycle (**F**) WNT Signaling, (**G**) NOTCH signaling, (**H**) Cholesterol/lipid biosynthesis processes, (**I**) Organelle biogenesis and maintenance, (**J**) Axon guidance induced ($p < 0.05$, >1.5 folds, ≤0.6 folds, base mean >10, $n = 3$, Wald chi-squared test) across C1 to C6 stage. STRING analysis of core genes of the pathway is also depicted.

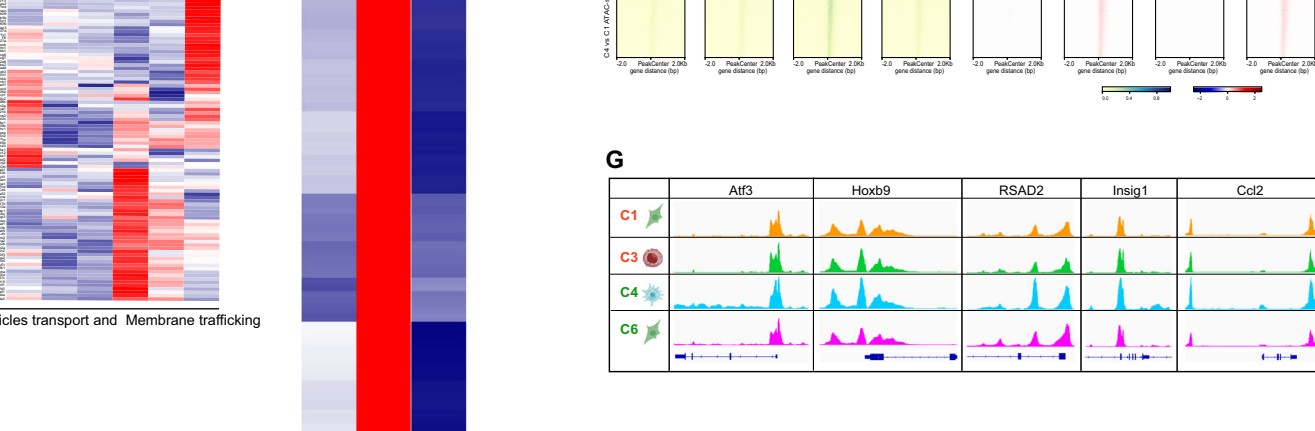

◀ **Figure EV5. Pathways of programmed cell revival.**

(A–C) Heatmap generated for genes ($p < 0.05$, >1.5 folds, base mean >10, $n = 3$, Wald chi-squared test) representing GO terms (A) Nucleotide and fatty acid metabolism. (B) Ion transport and amino acid transport. (C) Vesicles transport and membrane trafficking across different stages. (D) RNA sequencing experiment is performed with MEF cells treated with 4 mM LLOMe at the indicated three time points (C1, C4, and extended-C6) with three biological replicates. (E) The genes induced in the C4 stage ($p < 0.05$, >1.5 fold, see Dataset EV7) compared to the C1 stage was plotted across all three stages (C1, C4, and extended-C6). (F) The heatmaps represent the ATAC-seq signal intensities at loci with increased accessibility at C4 compared to C1 (padj<0.01, log2 foldchange >2) at different stages of 4 mM LLOMe treatment to MEF cells (C1, C3, C4, C6, C3/C1, C4/C1, C4/C3, and C6/C1). (The heatmaps represent a 4 kb window centered on the peak midpoint, and sorted based on the C4 enrichment compared to C1). (G) Genome browser screenshots of five selected genomic regions' signals in C1, C3, C4, and C6. (The blue bar at the bottom indicates the reference sequence (mm10) of the selected genomic regions).

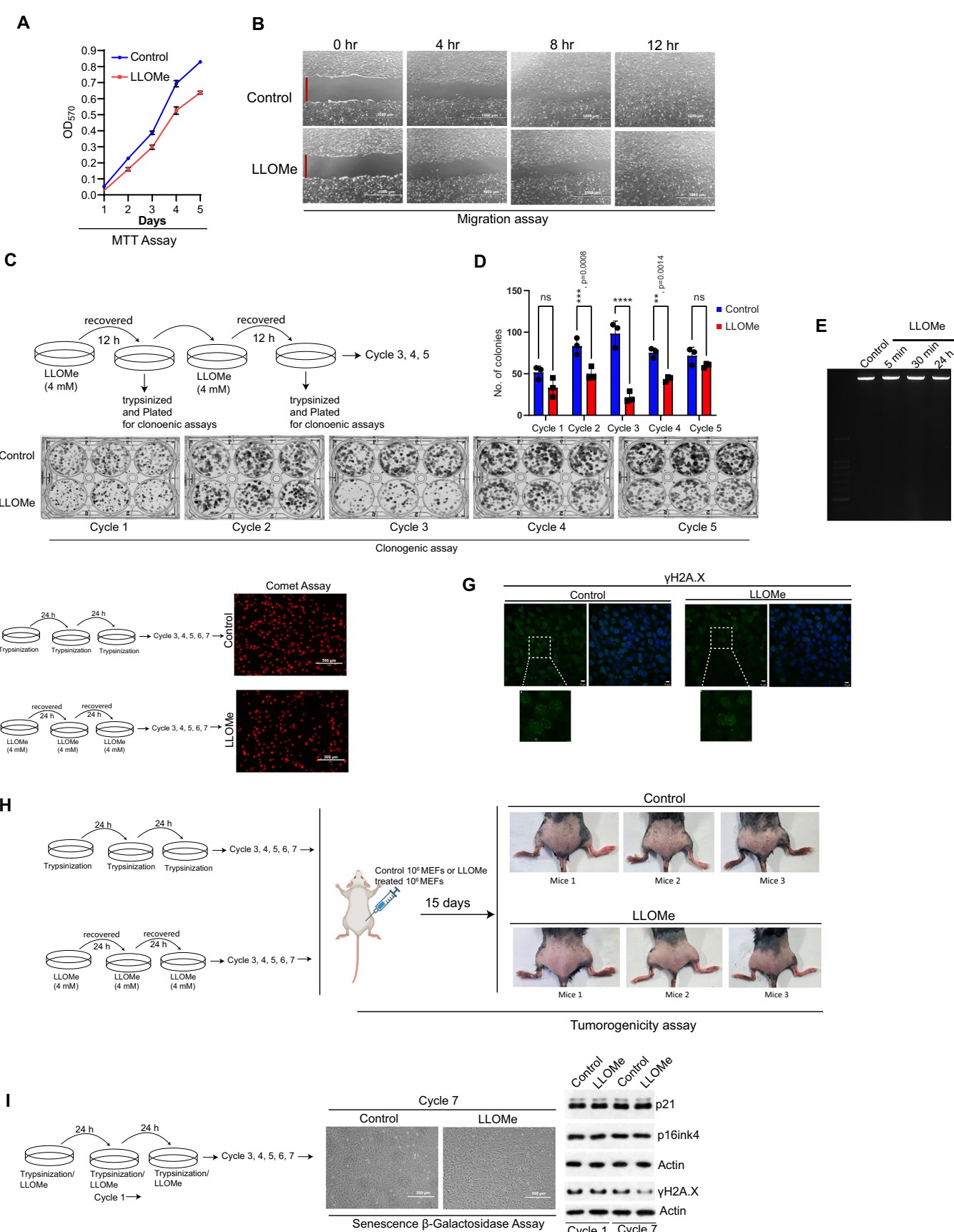

**Figure EV6.  LLOMe treatment does not make cells tumorigenic.**

(A) Analysis of cell proliferation using MTT assays with MEF cells untreated or treated with 4 mM LLOMe for the indicated time points. (three technical replicates, mean ± SD). (B) Representative time-lapse live microscopy images of migration assays performed with control or 4 mM LLOMe-treated MEF cells at the indicated time points. Magnification 4X. Scale bar, 1000 μm. (C, D) Schematic representation of the experimental design where the same cells were given five cycles of 4 mM LLOMe treatment and plated for (C) clonogenic assays in each cycle. (D) Graph depicts the number of colonies from clonogenic assays ($n = 3$, mean ± SD, **$p = 0.0014$, ***$p = 0.0008$, ****$p < 0.0001$, ns non-significant $p > 0.05$, unpaired $t$-test). (E) Agarose gel image for genomic DNA isolated from control and 4 mM LLOMe-treated MEF cells for the indicated time points. (F) Comet assays were performed with MEF cells that were subjected to seven cycles of trypsinization (control) or LLOMe treatment. (G) Immunofluorescence experiments using γH2A.X antibody were performed with MEF cells that were subjected to seven cycles of trypsinization (control) or LLOMe treatment. (H) MEF cells that were subjected to seven cycles of trypsinization (control) or LLOMe treatment were injected into the flanks of mice ($n = 6$). (I) MEF cells that were exposed to seven cycles of trypsinization (control) or LLOMe treatment were subjected to senescence-associated β-galactosidase assays or Western blot with p21, p16ink4, and γH2A.X antibody. Source data are available online for this figure.

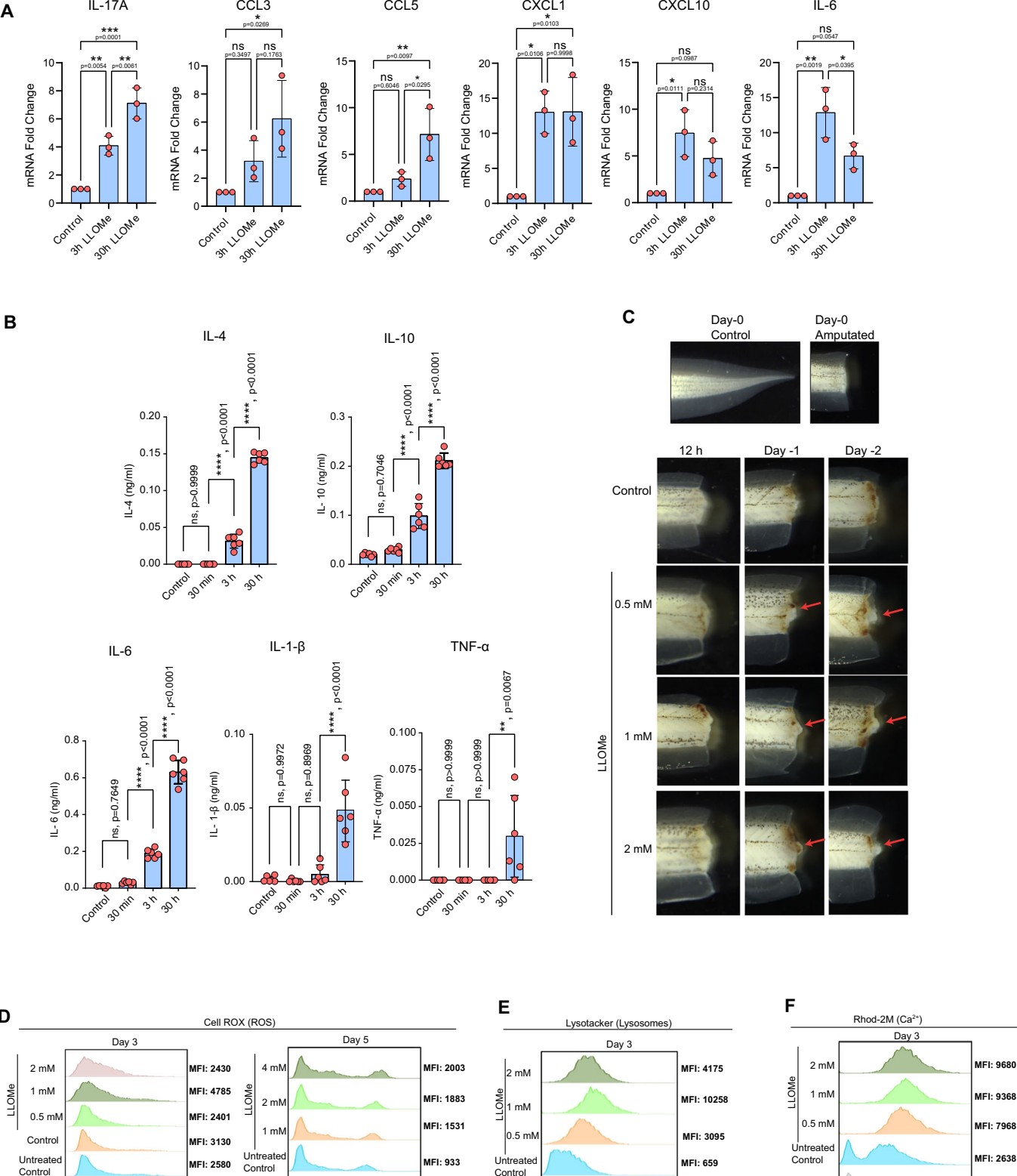

◀ **Figure EV7. LLOMe induces wound healing and tissue regeneration.**

(A) A qRT-PCR analysis of MEF cells treated with LLOMe for different time points as indicated in the figure ($n = 3$, mean ± SD, *$p < 0.05$, **$p < 0.01$, ***$p < 0.001$, ns non-significant $p > 0.05$, ordinary one-way ANOVA, Tukey's multiple comparison test). Exact $p$ values are depicted in the Figure and Appendix Table S2. (B) ELISA with supernatant of the MEF cells treated with LLOMe for different time points as indicated in the figure ($n = 3$, Mean ± SD, **$p = 0.0067$, ****$p < 0.0001$, ns non-significant $p > 0.05$, ordinary one-way ANOVA, Tukey's multiple comparison test). Exact $p$ values are depicted in the Figure and Appendix Table S2. (C) Top panel, representative images of a control tadpole tail and amputated tail on day 0. Bottom panel, representative images of untreated or LLOMe-treated amputated tails at the indicated time points. The red arrow indicates blastema formation. (D–F) Flow cytometry analysis of (D) ROS production using CellROX Green, (E) lysosomes numbers using lysotracker, and (F) mitochondrial calcium using Rhod-2 AM in untreated and 4 mM LLOMe-treated regenerated tissue of tadpole at the indicated time points. Source data are available online for this figure.

