## [Peer Review File · The EMBO Journal]

Programmed cell revival from imminent cell death enhances tissue repair and regeneration

Kollori Dhar, Kautilya Jena, Subhash Mehto, Rinku Sahu, Krushna Murmu, Atharva Mahajan, Sibaram Behera, Putchala Ravi Kiran, Reuben Mathew, Ramyasingh Bal, Soumya Kundu, Santosh Das, Swati Chauhan, Sameekshya Satapathy, Rina Yadav, Swatishmita Priyadarsini, Khyathi Padala, Prashanth Namdigalla, Sanchita Mishra, Prerana Muralidhara, Kushagra Bansal, Sannula Kesavardhana, Punit Prasad, Kiran Bokara, Divya Sowpati, Anindya Ghosh-Roy, Pravati Mahapatra, Rohan Khadilkar, Ramesh Yelagandula, and Santosh Chauhan

Corresponding author(s): Santosh Chauhan (schauhan@ccmb.res.in)

Review Timeline:

Submission Date:	19th Mar 25
Editorial Decision:	7th Apr 25
Revision Received:	9th Jun 25
Editorial Decision:	26th Jun 25
Revision Received:	4th Jul 25
Accepted:	25th Jul 25

Editor: Ioannis Papaioannou

Transaction Report:

Dear Santosh,

Thank you again for submitting your manuscript EMBOJ-2025-120830 for consideration by The EMBO Journal. It has now been seen by three experts in the field, and we have received the full set of their detailed and informative comments, which you can find below.

I am glad to say that, as you will see, all three referees are very supportive of this work. They find it a comprehensive study with novel and intriguing results that will be of broad interest to the community. They also identify a few limitations and errors that should be addressed in a revised version of the manuscript, and they provide some interesting and constructive suggestions for strengthening the work and increasing its impact on the field further.

Given the referees' positive comments and recommendations, I would like to invite you to submit a revised version of your manuscript taking the referees' suggestions on board, along with a detailed point-by-point response addressing all referees' comments. I should add that it is The EMBO Journal policy to allow only a single round of major revision, and acceptance of your manuscript will therefore depend on the completeness of your responses in this revised version. Please let me know if you have any questions or comments that you would like to discuss with me.

We generally allow three months as standard revision time (July 6, 2025). As a matter of policy, competing manuscripts published during this period will not negatively impact our assessment of the conceptual advance presented by your study. However, we request that you contact us as soon as possible upon publication of any related work, to discuss how to proceed. Should you foresee a problem in meeting this three-month deadline, please let us know in advance and we will be able to grant an extension.

Thank you for the opportunity to consider your work for publication in The EMBO Journal. I look forward to your revision.

Best regards,

Ioannis

Instructions for preparing your revised manuscript

1. When you are ready to submit the revision, please upload:

- A Word file of the manuscript text (including legends of main Figures, EV Figures and Tables). Please make sure that changes are highlighted (or "tracked") to be clearly visible.

- Individual production-quality figure files (one file per figure). When assembling your figures, please refer to our figure preparation guidelines in order to ensure proper formatting and readability in print as well as on screen:

If the data shown in a figure are obtained from n {less than or equal to} 2, please use scatter plots showing the individual data points.

- i. the name of the statistical test used to generate error bars and P values
- ii. the number (n) of independent experiments (please specify technical or biological replicates) underlying each data point (discussion of statistical methodology can be reported in the Materials and Methods section, but figure legends should contain a basic description of n , P , and the test applied)
- iii. the nature of the bars and error bars (s.d., s.e.m.).

- A point-by-point response to the referees' comments, with a detailed description of the changes made (as a word file). All referees' concerns must be fully addressed and their suggestions taken on board. When preparing your letter of response to the referees' comments, please bear in mind that this will form part of the Review Process File and will therefore be available online

to the community. Please note that you have the possibility to opt out of the transparent process at any stage prior to publication by letting the editorial office know (contact@embojournal.org); if you do opt out, the Review Process File link will point to the following statement: "No Review Process File is available with this article, as the authors have chosen not to make the review process public in this case.". For more details on our Transparent Editorial Process, please visit our website: <https://www.embopress.org/page/journal/14602075/authorguide#transparentprocess>

- Expanded View (EV) files (replacing Supplementary Information) that are collapsible/expandable online. A maximum of 5 EV Figures can be typeset. EV Figures should be cited as "Figure EV1, Figure EV2" etc. in the text, and their respective legends should be included in the manuscript file after the legends of regular figures. See detailed instructions regarding Expanded View files here:

- For the figures that you do NOT wish to display as Expanded View figures, they should be bundled together with their legends in a single PDF file called "Appendix", which should start with a short Table of Contents (including page numbers). Appendix figures should be referred to in the main text as: "Appendix Figure S1, Appendix Figure S2" etc. Please see detailed instructions here: <https://www.embopress.org/page/journal/14602075/authorguide#expandedview>

- A complete author checklist, which you can download from our author guidelines (<https://www.embopress.org/page/journal/14602075/authorguide>). Please note that the checklist will also be part of the Review Process File.

2. Please note that no statistics should be calculated and shown in Figures if $n=2$. Please also note that each p value should be reported as an exact value.

3. Before submitting your revision, primary datasets (and computer code, where appropriate) produced in this study need to be deposited in appropriate public databases (see <https://www.embopress.org/page/journal/14602075/authorguide#dataavailability>). In particular, we kindly request you to deposit all data from your RNA, ATAC, and Nanopore methylation sequencing experiments. The accession numbers, database, and the specific URLs (links) should be listed in a formal "Data availability" section (placed after Methods), following the example below:

"The RNA-seq datasets produced in this study are available in the following database:
Gene Expression Omnibus GSE46843 (<https://www.ncbi.nlm.nih.gov/geo/query/acc.cgi?acc=GSE46843>)"

*** All links should resolve to a page where the data can be accessed. ***

*** Please remember to provide in the Data availability section of your revised manuscript reviewer passwords if the datasets are not yet public. ***

*** The Data Availability Section is restricted to new primary data that are part of this study. In case you have no data that require deposition in a public database, please state so instead of referring to the database: "Our study includes no data deposited in public repositories." under the heading "Data availability". ***

4. The materials and methods need to be described in the manuscript using our structured methods format, which is now required for all research articles. According to this format, the Methods section includes a single "Reagents and Tools Table" - listing key reagents, experimental models, software and relevant equipment including their sources and relevant identifiers- followed by a "Methods and Protocols" section describing the methods. Please download and fill our Reagents and Tools Table template (.docx), which you can find in our author guide:

<https://www.embopress.org/page/journal/14602075/authorguide#structuredmethods>. When submitting your revised manuscript, please do not include the Reagents and Tools Table in the Methods section of the manuscript but instead upload it as a separate file choosing the file type "Reagent Table".

5. Please check that the title and the abstract of the manuscript are brief, yet explicit, even to non-specialists. The length of the title should not exceed 100 characters, and the abstract should be a single paragraph not exceeding 175 words.

6. Please also note our reference format: <https://www.embopress.org/page/journal/14602075/authorguide#referencesformat>.

8. Please remember: digital image enhancement is acceptable practice, as long as it accurately represents the original data and conforms to community standards. If a figure has been subjected to significant electronic manipulation, this must be noted in the figure legend or in the "Materials and Methods" section. The editors reserve the right to request original versions of figures and the original images that were used to assemble the figure.

9. Our journal encourages inclusion of data citations in the reference list to directly cite datasets that were obtained from public databases. Data citations in the article text are distinct from normal bibliographical citations and should directly link to the database records from which the data can be accessed. In the main text, data citations are formatted as follows: "Data ref: Smith et al, 2001" or "Data ref: NCBI Sequence Read Archive PRJNA342805, 2017". In the Reference list, data citations must be labeled with "[DATASET]". A data reference must provide the database name, accession number/identifiers, and a resolvable link to the landing page from which the data can be accessed at the end of the reference. Further instructions are available at: <https://www.embopress.org/page/journal/14602075/authorguide#referencesformat>.

10. We request authors to consider both actual and perceived competing interests. Please review our policy (<https://www.embopress.org/page/journal/14602075/authorguide#conflictsinterest>) and update your competing interests statement if necessary. Please name this section 'Disclosure and competing interests statement' and place it after the Acknowledgements section.

11. Please note that all corresponding authors are required to provide an ORCID ID upon submission of a revised manuscript (<https://orcid.org/>). Please find instructions on how to link your ORCID ID to your account in our manuscript tracking system in our Author guidelines (<https://www.embopress.org/page/journal/14602075/authorguide#authorshipguidelines>).

12. We use CRediT to specify the contributions of each author in the journal submission system. CRediT replaces the author contribution section, which should be removed from the manuscript. Please use the free text box to provide more detailed descriptions. See also guide to authors: <https://www.embopress.org/page/journal/14602075/authorguide#authorshipguidelines>.

14. We would also welcome the submission of cover suggestions or motifs to be used by our Graphics Illustrator in designing a cover.

15. Please use the link below to submit your revision:
<https://emboj.msubmit.net/cgi-bin/main.plex>

Referee #1:

- You should be trying to help the work get published, not necessarily in this journal but ultimately.
- Don't criticize an experiment unless you can tell the authors how they could do it better:
My mentor would say, "If you just want to throw darts, go to the pub".
- Keep in mind that no one ever built a statue to a critic.
- Try to act as a peer in the process of peer review.

Science Signaling 2009 Michael Yaffe

Title: Revival from Cell Death Phase Unleashes Developmental Cues that Enhance Tissue Repair and Regeneration

Manuscript # EMBOJ-2025-120830

General Remarks

This study looks at cell recovery (or revival) from a sub-lethal apoptotic stimulus by the lysosomotropic detergent LLOMe. The interesting angle is that the cells, to all intent and purposes, seem to have activated cell death beyond the ability to recover. In this sense the manuscript goes against the current dogma, more so than in Gong et al 2017 paper (cited) where cells recover from plasma membrane damage caused by MLKL. This work has similarities with the 2017 Sun et al JCB paper (also cited), nevertheless it goes beyond this work. In some way it is a little disappointing that similar pathways weren't identified in the two works and some discussion should be made of this, in my opinion. It is also quite remarkable in the number of in vivo models used to demonstrate a "healing" effect of low dose LLOMe.

Fig. 1 shows that cells that appear to be undergoing apoptosis induced by LLOMe (blebbing, floating in media, Western blots showing markers of apoptosis such as cleaved PARP and necroptosis e.g. phospho-MLKL) can nevertheless survive. Fig. 2

shows some of the expected disruptions to organelles following LLOMe treatment and recovery using confocal microscopy. Scale bars should be defined in the legends because they are too small to see in the panels. Fig. 3 describes a transcriptomic analysis of the recovery phase, the Western blot confirming some of the implicated pathways looks reasonable with clear upregulation/activation of ATF3 and phospho-p38. Nevertheless, mw markers should be included. Fig. 4 uses ATAC sequencing to implicate NF- κ B signalling as an important element of the recovery. Fig. 5 uses small molecule inhibitors of the signalling pathways implicated in Fig. 4 and demonstrates quite clearly that inhibition of NF- κ B prevents cell recovery. Fig. 6 and 7 uses in vivo models in different model organisms (mouse, rat, tadpole and *Drosophila melanogaster*) to demonstrate that "low dose" LLOMe can increase tissue healing.

I wouldn't personally use the acronym PCR or programmed cell revival because PCR is too strongly associated with a method.

Error bars are defined, but scale bars in micrographs need to be described and Western blots should be correctly labelled with mw markers etc.

Specific Remarks

line 216 - p-RIPK2 should be RIPK3 as a necroptosis marker (as it is in the figure).

Referee #2:

This manuscript by Dhar and colleagues describes a surprising phenomenon whereby dying cells recover from lysosomotropic agents-induced cell death. Although the molecular mechanism controlling this 'resurrection' is not completely elucidated in this study, the authors demonstrated using various mouse, flies and worm models that this pathway promotes wound healing in vivo. Overall, the conclusions are largely supported by the data, and this interesting study that will be of interest to the broad cell death community. I have listed some suggestions can strengthen the conclusion of this study.

1. It is intriguing that the floating cells can reattach and recover from LLOMe-induced cell death. Is it possible that a small fraction of these cells is in fact resistant to LLOMe-induced cell death, and it is this fraction that proliferates over time to give the impression of 'recovery' or 'resurrection'? Can the authors perform more control experiments to further strengthen this conclusion?
2. The authors concluded that cells displaying cell death markers including cleaved caspase-3, PARP, GSDMD and pMLKL can recover from death and revert to healthy viable cells. This is likely an overstatement since western blot measures the bulk population and not at a single cell level. Thus, it is likely that a percentage of cells with death markers progress to cell death, while the others revert. Single cell experiment will be needed to draw this conclusion. For example, the authors can use a fluorogenic caspase-3/7 substrate to demonstrate that a cell with active caspase-3/7 activity is indeed resistant to cell death and reverts to a healthy cell.
3. Since gene expression studies demonstrate that numerous genes linked to wound healing is induced during resuscitation, the authors propose to exploit this pathway for wound healing and first examined that these cells are not oncogenic. However, this requires more robust examination, since it is possible that repeated death-reviving cycles can indeed induce some form of mutation and DNA damage. For example, previous studies showed that minority DNA damage triggers sublethal caspase-3/7 activation and DNA damage that promotes tumorigenesis (PMID: 25866249). I encourage the authors to examine this possibility in more detail if LLOMe-induced death is indeed intended to accelerate wound healing.
4. Do the revived cells display grow as well as untreated cells, and do they enter senescence at a faster rate? This will be important to address should LLOMe-induced death is to be repurposed for in vivo applications.

Minor comments

1. Line 216 Typo error: pRIPK3, not pRIPK2
2. I suggest including molecular weight to all immunoblots, as it is difficult to conclude what are the cleavage bands observed, especially for GSDMD where they can be cleaved into various different sizes.

Referee #3:

In this study, Dhar et al. investigate how activation of cell death pathways can connect to cellular survival and regeneration through a phenomenon they refer to as "programmed cell revival". Using a variety of in vitro and in vivo approaches, including live-cell imaging, flow cytometry, calcium and mitochondrial reporters, immunostaining, western blotting, RNAseq, ATACseq, and mouse, tadpole, and *Drosophila* models, the authors provide a comprehensive phenotypic assessment of the cellular phenomenon and molecular kinetics of programmed cell revival over time, as the cell recovers from a cell death stimulus. This provides key insights into the cellular and organellar changes that are occurring during this process. They found that the revival process activates key pathways in metabolism, organelle biogenesis, membrane trafficking, transport, and cytoskeleton remodeling, ultimately leading to full cellular renewal. Mechanistically, this process enhances tissue regeneration across species, and NF- κ B signaling is the main driver of both revival and regeneration. Overall, this study provides a large body of

robust data and critical new datasets to track the cellular and organellar changes occurring during programmed cell revival. A few refinements would improve the clarity and the overall strength of the manuscript.

Specific Comments

1. The current focus is primarily on non-immune and dividing cells. It would be useful to assess whether the same mechanisms apply to immune and non-dividing cells. The authors could test this by evaluating the revival process in primary murine immune cells, such as macrophages, to determine whether the same molecular pathways are activated. This does not need to be extensively characterized, but a proof-of-concept experiment to show the phenotype would be helpful.
2. If samples are available, can the authors test whether cytokines and growth factors, such as IL-4, IL-10, IL-13, TGF- β , EGF, and TGF- α , are released in response to LLOMe? These molecules play critical roles in wound healing and tissue regeneration, and their release at different timepoints could be important for the regenerative effects observed.
3. The authors observed activation of the cell death molecules PARP-1, caspase-3, p-RIPK3, p-MLKL, and GSDMD at the C3, C4, and/or C5 stages, when cells are in the revival phase. Do these cell death molecules directly contribute to the revival process? This could be tested by using caspase inhibitors (zVAD) to assess how they affect the revival process in response to LLOMe.
4. The discussion would benefit from expansion on key points. Specifically, the authors could discuss how the nature of the cell death induced (e.g., lytic vs non-lytic) impacts the regenerative outcome, and how their data shape our understanding of this phenomenon. Not all cell death triggers induce the survival and regeneration programs seen with LLOMe, and clarifying how different triggers influence regenerative versus pathological outcomes would strengthen the overall understanding of the experimental design and its relevance to tissue repair.
The authors could also discuss how the molecules released from dead and dying cells (e.g., cytokines, DAMPs) contribute to regeneration, using examples from the literature.
5. Supplementary Figures 8 and 9 are mislabeled as 9 and 10 in the results section.

Referee #1:

- You should be trying to help the work get published, not necessarily in this journal but ultimately.

- Don't criticize an experiment unless you can tell the authors how they could do it better:

My mentor would say, "If you just want to throw darts, go to the pub".

- Keep in mind that no one ever built a statue to a critic.

- Try to act as a peer in the process of peer review.

Science Signaling 2009 Michael Yaffe

Title: Revival from Cell Death Phase Unleashes Developmental Cues that Enhance Tissue Repair and Regeneration

Manuscript # EMBOJ-2025-120830

General Remarks

This study looks at cell recovery (or revival) from a sub-lethal apoptotic stimulus by the lysosomotropic detergent LLOMe. The interesting angle is that the cells, to all intent and purposes, seem to have activated cell death beyond the ability to recover. In this sense the manuscript goes against the current dogma, more so than in Gong et al 2017 paper (cited) where cells recover from plasma membrane damage caused by MLKL. This work has similarities with the 2017 Sun et al JCB paper (also cited), nevertheless it goes beyond this work. In some way it is a little disappointing that similar pathways weren't identified in the two works and some discussion should be made of this, in my opinion. It is also quite remarkable in the number of in vivo models used to demonstrate a "healing" effect of low dose LLOMe.

Response: We are deeply grateful to the reviewer for insightful comments and a positive overall assessment of our work. We especially appreciate the reviewer's perspective on the spirit of peer review.

The reviewer rightly pointed out the connections between our findings and those of Sun et al. (2017). We agree that exploring these relationships further strengthens the interpretation of our results. Sun et al., used ethanol as an agent to induce apoptosis. We used a sublethal concentration of LLOMe, a lysosomotropic agent. The sublethal concentration of LLOMe caused lysosomal damage; however, the cell was able to repair this damage. Lysosomes are hubs of signaling pathways, including autophagy, mTOR, AMPK, lipid metabolism, and innate immune signaling (STING). In our study, we believe that disruption of lysosomes played a discernible role in inducing multiple protective and repair pathways.

We analyzed Sun et al., data using the Metascape pathway analysis tool and compared it with our analysis. Interestingly, the activation of developmental pathways and transcription factors that control the early revival genes showed a striking similarity (**New Figure 1**). The red-highlighted pathways and transcription factors were similar to those in our study. Additionally, both datasets reveal early responses involving genes related to chromatin modification (Sun et al., Figure 2B and our manuscript, Figure 3), as well as pathways associated with membrane trafficking, MAPK signaling, WNT signaling, TGF- β signaling, p53 signaling, and organelle biogenesis. One striking difference was the presence of a strong cytokine response (especially an IFN response) in our early response genes compared to those of Sun et al. Similarly, a similarity was also observed in late-response genes and pathways, including some metabolic processes and cell morphogenesis processes. We have added this data to the revised manuscript (**Appendix Figure S3**) and discussed it in the results, lines 420-432

New Figure 1

Early Genes (Sun et al. 2017)

Fig. 1 shows that cells that appear to be undergoing apoptosis induced by LLOMe (blebbing, floating in media, Western blots showing markers of apoptosis such as cleaved PARP and necroptosis e.g. phospho-MLKL) can nevertheless survive. Fig. 2 shows some of the expected disruptions to organelles following LLOMe treatment and recovery using confocal microscopy. Scale bars should be defined in the legends because they are too small to see in the panels.

Response: Thanks for the concern. We have now defined the scale bar in the legends and increased the size of the scale bar throughout the manuscript.

Fig. 3 describes a transcriptomic analysis of the recovery phase, the Western blot confirming some of the implicated pathways looks reasonable with clear upregulation/activation of ATF3 and phospho-p38. Nevertheless, mw markers should be included.

Response: We apologize for our mistake. We have now added molecular weight markers throughout the manuscript.

Fig. 4 uses ATAC sequencing to implicate NF- κ B signalling as an important element of the recovery. Fig. 5 uses small molecule inhibitors of the signalling pathways implicated in Fig. 4 and demonstrates quite clearly that inhibition of NF- κ B prevents cell recovery. Fig. 6 and 7 uses in vivo models in different model organisms (mouse, rat, tadpole and *Drosophila melanogaster*) to demonstrate that "low dose" LLOMe can increase tissue healing. I wouldn't personally use the acronym PCR or programmed cell revival because PCR is too strongly associated with a method.

Response: We agree with the reviewer, and we have removed the acronym "PCR" from the manuscript.

Error bars are defined, but scale bars in micrographs need to be described and Western blots should be correctly labelled with mw markers etc.

Response: We apologize for our mistake. We have now added molecular weight markers and scale bars throughout the figures, and scale bars are not defined in the legends.

Specific Remarks

line 216 - p-RIPK2 should be RIPK3 as a necroptosis marker (as it is in the figure).

Response: We apologize for our mistake. We have corrected this now.

Referee #2:

This manuscript by Dhar and colleagues describes a surprising phenomenon whereby dying cells recover from lysosomotropic agents-induced cell death. Although the molecular mechanism controlling this 'resurrection' is not completely elucidated in this study, the authors demonstrated using various mouse, flies and worm models that this pathway promotes wound healing in vivo. Overall, the conclusions are largely supported by the data, and this interesting study that will be of interest to the broad cell death community. I have listed some suggestions can strengthen the conclusion of this study.

Response: We are very thankful to the reviewer for appreciating our work and providing suggestions for the improvement of the manuscript. We have conducted new experiments and also provided a more accurate description of our previous results.

1. It is intriguing that the floating cells can reattach and recover from LLOMe-induced cell death. Is it possible that a small fraction of these cells is in fact resistant to LLOMe-induced cell death, and it is this fraction that proliferates over time to give the impression of 'recovery' or 'resurrection'? Can the authors perform more control experiments to further strengthen this conclusion?

Response: We appreciate the reviewer's insightful question regarding the possibility of a resistant subpopulation contributing to the observed recovery. This was also our concern when we began our study five years ago, but after conducting hundreds of experiments, we have no doubt that the same cell revives. Over the years, we conducted a series of experiments to dispel this doubt. The results of our multiple live imaging experiments were so apparent that there is not even a remote possibility that LLOMe-resistant cells will proliferate and give the impression of recovery.

1. We have extensive live-cell imaging data (Movie EV 1, 2, 3, 4, 5, 6, 8, 9, 10) that directly demonstrate individual cells undergoing shrinkage, blebbing (characteristic of early apoptosis), and subsequent revival within a relatively short timeframe of approximately 3 hours. In each video, we can easily trace the fate of each cell. The videos show that a large proportion of cells (over 95%) undergo this revival process, which makes it unlikely that the cell population reviving back is caused by the proliferation of resistant cells in such a short period. Considering the doubling time of MEFs is approximately 48 hours, the observed rapid recovery is inconsistent with a scenario of selective proliferation.
2. In addition to LLOMe, we have observed similar revival phenomena with other lysosomotropic agents, such as GPN and sphingosine, when used at sublethal

concentrations. Again, a vast majority (90-95%) of cells undergo the revival process within 3-4 hours (Movie EV 5 and 6). If a small, resistant population were responsible, one would expect to see significant variability in the response to different agents.

3. To directly address the possibility of a resistant subpopulation, we performed

experiments where we specifically harvested ONLY the floating cells (after centrifugation) following LLOMe treatment. These floating cells, which exhibited morphological changes consistent with early apoptosis, were collected, washed, and

replated in a new cell culture dish in fresh media. We observed that 90-95% of these replated cells revived and regained normal morphology within 6 hours, and by 24 hours, they appeared normal, dividing cells. If the revival were due to a small resistant subpopulation, we would not expect such a high percentage of the floating cell population to recover.

To further enhance the visualization of this process, we have performed live microscopy (at 10X magnification) in the revised manuscript. We collected floating cells, washed them with media, and replated them in a new culture plate, then performed live microscopy (New data-**Movie EV4**). This video enables the clear tracing of individual floating cells through revival stages, further supporting our conclusion that almost all individual cells are indeed recovering from LLOMe-induced damage. Here, we provide a snapshot of the same video, tracing multiple floating cells to their revival (**New Figure 2, below**).

We believe that the combined evidence from these experiments, particularly the live-cell imaging and the replating of floating cells, provides strong support for our conclusion that individual cells can revive from a near-death state following LLOMe treatment. We hope that this additional clarification addresses the reviewer's concern and provides a convincing rationale for our interpretation of the data. We have added this data to the revised manuscript (**Movie EV4 and Figure EV1B, C**) and discussed it in the results section, lines 143-147

New Figure 2

2. The authors concluded that cells displaying cell death markers including cleaved caspase-3, PARP, GSDMD and pMLKL can recover from death and revert to healthy viable cells. This is likely an overstatement since western blot measures the bulk population and not at a single cell level. Thus, it is likely that a percentage of cells with death markers progress to cell death, while the others revert. Single cell experiment will be needed to draw this conclusion. For example, the authors can use a fluorogenic caspase-3/7 substrate to demonstrate that a cell with active caspase-3/7 activity is indeed resistant to cell death and reverts to a healthy cell.

Response: We understand the reviewer's concern. Although it has been previously shown that cells could revive even after activation of MLKL and Caspases (using fluorescent markers)(Ding *et al*, 2016; Gong *et al*, 2017; Sun *et al*, 2017), we were also confused. In our set of conditions, we only found very feeble cleavage of Caspase-1/3, GSDMD, and MLKL activation. We have stated this clearly in the original manuscript (Line 225-230). As suggested by the reviewer, fluorogenic caspase-3/7 assays are used to detect and quantify caspase-3/7 activity. These assays typically involve a fluorogenic substrate that is cleaved by caspase-3/7, releasing a fluorescent dye. The release of this dye, which can bind to DNA, indicates caspase activity and apoptosis. Although these assays are suitable for conditions where we want to score cell death, they are not suitable for scoring revival from cell death, as once these dyes **intercalate into DNA**, they may prevent the cell from reviving. Indeed, we found that LLOMe treatment resulted in increased fluorescence in cells, which suggests that sublethal concentrations of LLOMe caused Caspase-3/7 cleavage (**Video file 1-for review purposes only**). However, none of the cells revived, as expected, with the intercalation of the dye into the DNA.

Next, we employed a ZipGFP-based caspase reporter, which constitutively expresses mCherry, but GFP is **only** produced upon caspase activation (To *et al*, 2016). We observed weak caspase activation (GFP) in many transfected HEK293T cells, which revived back upon treatment with sublethal concentrations of LLOMe (Movie EV12). Whereas there was no caspase activity in control cells (Movie EV12). This data is consistent with our Western blot data. Taken together, the data indicate that cells can revive from a low level of caspase activation. This data is discussed in lines 231-237 and presented in Movie EV12.

3. Since gene expression studies demonstrate that numerous genes linked to wound healing is induced during resuscitation, the authors propose to exploit this pathway for wound healing and first examined that these cells are not oncogenic. However, this requires more robust examination, since it is possible that repeated death-reviving cycles can indeed induce some form of mutation and DNA damage. For example, previous studies showed that minority DNA damage triggers sublethal caspase-3/7 activation and DNA damage that promotes tumorigenesis (PMID: 25866249). I encourage the authors

to examine this possibility in more detail if LLOMe-induced death is indeed intended to accelerate wound healing.

Response: We thank the reviewer for raising an important point. Since we also share similar concerns, we have performed multiple experiments in the original manuscript (Supplementary Figure 8, now Fig. EV6). We found that LLOMe-treated cells have no growth advantage in MTT assays (Supplementary Figure 8A, now Fig. EV6) and migration assays (Supplementary Figure 8B, now Fig. EV6). In addition, we treated cells for five cycles of LLOMe and plated cells for clonogenic assays (gold standard) in each cycle (Supplementary Figure 8C-D, now Fig. EV6). We again didn't find much growth difference between the trypsinized control cells and the LLOMe-treated cells; rather, the LLOMe-treated cells were growing slightly slower. In tumorigenicity assays, no tumor growth was observed when LLOMe-treated cells were injected in mice (Supplementary Figure 8F, now Fig. EV6).

We would like to emphasize the key differences between the study reviewer mentioned (PMID: 25866249) and ours. The cited work employed ionizing radiation, a known mutagenic agent that induces significant and widespread DNA damage (in addition to caspase-3 activation), as evidenced by substantial γ H2A.X staining and positive comet assays. In contrast, our data indicate that sublethal concentrations of LLOMe do not elicit a significant DNA damage response in transcriptome analysis (Figure 3D, G, H) or DNA

fragmentation in agarose gel ran from genomic DNA of LLOMe-treated cells (**New Figure 3**).

To further address the reviewer's concerns, we conducted many more experiments to specifically examine DNA damage following LLOMe exposure. We subjected MEFs to **multiple cycles (7 cycles)** of sublethal LLOMe treatment and assessed DNA damage using comet assays and γ H2A.X foci staining (same methods used in the reviewer's cited article) (**New Figure 4, 5**). Our results showed no discernible increase in comet tail length or γ H2A.X foci formation, indicating that LLOMe does not induce significant DNA damage even after multiple cycles of treatment.

Finally, to rigorously evaluate the potential for LLOMe-treated cells to induce tumor formation, we performed in vivo tumorigenicity assays in mice (n=6). Cells treated with one or multiple cycles of LLOMe were injected into mice, and the injection sites were monitored for tumor growth over time. We observed no tumor formation in any of the animals tested (**New Figure 6**).

one or multiple cycles of LLOMe were injected into mice, and the injection sites were monitored for tumor growth over time. We observed no tumor formation in any of the animals tested (**New Figure 6**).

Taken together, our experimental data and preliminary clinical observations strongly suggest that sublethal LLOMe treatment, under the conditions used in our studies, does not induce significant DNA damage or promote oncogenic transformation. We believe this comprehensively addresses the reviewer's concern and provides a strong rationale for exploring LLOMe as a potential therapeutic agent for wound healing. This data is discussed in lines 594-605 and presented in Figure EV6.

4. Do the revived cells display grow as well as untreated cells, and do they enter senescence at a faster rate? This will be important to address should LLOMe-induced death is to be repurposed for in vivo applications.

Response: We didn't find a major difference in the growth rate of control cells and revived MEFs in multiple experiments (original Manuscript Supplementary Figure 8, Fig. EV6). Now, we investigated whether repeated exposure to LLOMe might accelerate senescence, a key consideration for potential in vivo applications. Even after seven cycles of LLOMe treatment, we observed no significant increase in cellular senescence, as determined by senescence-associated β -gal (SASP) staining (New Figure 7). Furthermore, Western blot analysis, using established senescence markers such as p16ink4, p21, and γ H2A.X, revealed no appreciable differences between cells treated with one cycle of LLOMe and those subjected to multiple cycles (New Figure 7). This indicates that LLOMe treatment, at least under these conditions, does not appear to promote premature senescence. This data is discussed in lines 606-613 and presented in Figure EV6.

Minor comments

1. Line 216 Typo error: pRIPK3, not pRIPK2

Response: Thanks, we have corrected this.

2. I suggest including molecular weight to all immunoblots, as it is difficult to conclude what are the cleavage bands observed, especially for GSDMD where they can be cleaved into various different sizes.

Response: We are sorry for this omission. We have added molecular weight markers throughout the manuscript.

Referee #3:

In this study, Dhar et al. investigate how activation of cell death pathways can connect to cellular survival and regeneration through a phenomenon they refer to as "programmed cell revival". Using a variety of in vitro and in vivo approaches, including live-cell imaging, flow cytometry, calcium and mitochondrial reporters, immunostaining, western blotting, RNAseq, ATACseq, and mouse, tadpole, and Drosophila models, the authors provide a comprehensive phenotypic assessment of the cellular phenomenon and molecular kinetics of programmed cell revival over time, as the cell recovers from a cell death stimulus. This provides key insights into the cellular and organellar changes that are occurring during this process. They found that the revival process activates key pathways in metabolism, organelle biogenesis, membrane trafficking, transport, and cytoskeleton remodeling, ultimately leading to full cellular renewal. Mechanistically, this process enhances tissue regeneration across species, and NF- κ B signaling is the main driver of both revival and regeneration. Overall, this study provides a large body of robust data and critical new datasets to track the cellular and organellar changes occurring during programmed cell revival. A few refinements would improve the clarity and the overall strength of the manuscript.

Response: We are very thankful to the reviewer for appreciating the work. We have revised the manuscript in accordance with the reviewer's suggestions.

Specific Comments

1. The current focus is primarily on non-immune and dividing cells. It would be useful to assess whether the same mechanisms apply to immune and non-dividing cells. The authors could test this by evaluating the revival process in primary murine immune cells, such as macrophages, to determine whether the same molecular pathways are activated. This does not need to be extensively characterized, but a proof-of-concept experiment to show the phenotype would be helpful.

Response: Thanks for the suggestion. To investigate the effects of LLOMe on bone marrow-derived macrophages (BMDMs), we performed concentration-response experiments to identify the LLOMe concentration at which cells exhibited recovery following an initial shrinking. Higher (1-4 mM) LLOMe concentrations induced irreversible cell death in BMDMs, suggesting that immune primary cells are much more sensitive to LLOMe. The lower concentrations of 0.25 mM and 0.5 mM resulted in a transient shrinking phenotype followed by cell recovery (**New Figure 8A**). Notably, the morphological changes observed in these BMDMs differed from those typically seen in

primary, non-immune proliferating cells (New Figure 8A). This data is discussed in lines 169-172 and presented in Figure EV1.

New Figure 8

A

New Figure 8B

Next, we performed qRT-PCR using genes that represent major pathways upregulated during revival. We found that, except for one, none of these genes were induced during the revival process in BMDM; rather, their expression was reduced (**New Figure 8B**). The data further suggests that immune cells may not follow the revival processes and pathways as non-immune or dividing cells do. This data is presented in Appendix Figure S2, lines 414-419.

New Figure 9

2. If samples are available, can the authors test whether cytokines and growth factors, such as IL-4, IL-10, IL-13, TGF- β , EGF, and TGF- α , are released in response to LLOMe? These molecules play critical roles in wound healing and tissue regeneration, and their release at different timepoints could be important for the regenerative effects observed.

Response: This is an important suggestion. The wound healing process consists of four phases: homeostasis, inflammation, proliferation, and remodeling. A review of the literature suggests that multiple cytokines/chemokines, both pro-inflammatory and anti-inflammatory, are essential for wound healing (Mahmoud *et al*, 2024; Ridiandries *et al*, 2018). For example, many proinflammatory cytokines, such as TNF- α , IL-6, IL-1 β , IL-17A, and IFN- γ , play a crucial role in the chemotaxis and migration of various immune cells to the wound-healing site. Many chemokines, such as CXCL1, 2, 4, 7, 19, 12, and CCL2, 3, 5, play a role in all four phases of wound healing, including chemotaxis of immune cells (Mahmoud *et al.*, 2024; Ridiandries *et al.*, 2018). We performed expression analysis of many of these cytokines, IL17-A, CCL3, CCL5, CXCL1, CXCL10, and IL-6 by qRT-PCR during various stages of cell revival (**New Figure 9A**). We also tested secretion of IL-4, IL-10, IL-6, IL-1 β , and TNF α using ELISA (**New Figure 9B**). We found that all of these regenerative anti- and pro-inflammatory cytokines were expressed and secreted in significantly higher amounts from reviving cells. It appears that a sublethal concentration of LLOMe induces a balanced immune response, where both pro-inflammatory and anti-inflammatory cytokines/chemokines work together for better tissue repair and regeneration. This could be a standalone study on how these balanced responses are self-limiting and regulated. We have integrated this dataset into Figure EV7 and discussed it in the results section, in lines 639-649

3. The authors observed activation of the cell death molecules PARP-1, caspase-3, p-RIPK3, p-MLKL, and GSDMD at the C3, C4, and/or C5 stages, when cells are in the revival phase. Do these cell death molecules directly contribute to the revival process? This could be tested by using caspase inhibitors (zVAD) to assess how they affect the revival process in response to LLOMe.

Response: This is an interesting question. We tested the effect of zVAD on the revival process. We used a range of zVAD concentrations (20, 40, 80 μ M), which were suggested in the literature to be effective in MEFs. We found that lower concentrations of zVAD (20 and 40 μ M) did not affect the recovery of MEF cells (**New Figure 10A**). However, a higher concentration of zVAD (80 μ M) impaired revival, resulting in a delayed recovery and a final revival rate of only approximately 50-60% (**New Figure 10A**). A similar result was obtained when the LLOMe-treated floating cells were collected and replated in a fresh culture plate (**New Figure 10B**). These findings support the hypothesis that caspase activation may play a role (although not critical) in regulating the revival process. We have

integrated this dataset into Figure EV1 and Appendix Fig. S1 and discussed it in the results section, in lines 241-250.

New Figure 10

4. The discussion would benefit from expansion on key points. Specifically, the authors could discuss how the nature of the cell death induced (e.g., lytic vs non-lytic) impacts the regenerative outcome, and how their data shape our understanding of this phenomenon. Not all cell death triggers induce the survival and regeneration programs seen with LLOMe, and clarifying how different triggers influence regenerative versus pathological outcomes would strengthen the overall understanding of the experimental design and its relevance to tissue repair. The authors could also discuss how the molecules released from dead and dying cells (e.g., cytokines, DAMPs) contribute to regeneration, using examples from the literature.

Response: We agree with the reviewer. We already touched upon this in the original manuscript, but we have now expanded this further. Please see lines 815-861.

5. Supplementary Figures 8 and 9 are mislabeled as 9 and 10 in the results section.

Response: This is now corrected. Thanks!

References

- Ding X, Sun G, Argaw YG, Wong JO, Easwaran S, Montell DJ (2016) CasExpress reveals widespread and diverse patterns of cell survival of caspase-3 activation during development in vivo. *Elife* 5
- Gong YN, Guy C, Olauson H, Becker JU, Yang M, Fitzgerald P, Linkermann A, Green DR (2017) ESCRT-III Acts Downstream of MLKL to Regulate Necroptotic Cell Death and Its Consequences. *Cell* 169: 286-300 e216
- Mahmoud NN, Hamad K, Al Shibitini A, Juma S, Sharifi S, Gould L, Mahmoudi M (2024) Investigating Inflammatory Markers in Wound Healing: Understanding Implications and Identifying Artifacts. *ACS Pharmacol Transl Sci* 7: 18-27
- Ridiandries A, Tan JTM, Bursill CA (2018) The Role of Chemokines in Wound Healing. *Int J Mol Sci* 19
- Sun G, Guzman E, Balasanyan V, Conner CM, Wong K, Zhou HR, Kosik KS, Montell DJ (2017) A molecular signature for anastasis, recovery from the brink of apoptotic cell death. *J Cell Biol* 216: 3355-3368
- To TL, Schepis A, Ruiz-Gonzalez R, Zhang Q, Yu D, Dong Z, Coughlin SR, Shu X (2016) Rational Design of a GFP-Based Fluorogenic Caspase Reporter for Imaging Apoptosis In Vivo. *Cell Chem Biol* 23: 875-882

Dear Santosh,

Thank you for submitting your revised manuscript (EMBOJ-2025-120830R) to The EMBO Journal for our consideration, and for your patience during peer review. As I have already informed you, your manuscript has been sent back for re-review to the original referees #2 and #3, who had previously assessed the initial version of your work, and we have received their comments (included below).

I am very pleased to say that both referees are very satisfied with your revision, find all initially raised criticisms and concerns successfully addressed, and recommend publication of the manuscript without any further comments. In light of this expert input, I am happy to inform you that your manuscript has been in principle accepted for publication in The EMBO Journal. Congratulations on an excellent work!

Before we can move forward with formal acceptance and publication of your article, there are a few changes and corrections we need you to make in a final version of your manuscript:

- Please mark the corresponding author using an asterisk or another symbol on the title page of the manuscript. The e-mail address of the corresponding author should also be provided on this page.
- The names of all co-authors should be listed in first name/last name order on the title page of the manuscript, and their profiles in our manuscript tracking system should be updated accordingly.
- Please provide a list of up to 5 relevant keywords (preferably general/broad terms to enhance online search engine discoverability of your article) after the Abstract of your revised manuscript.
- Thank you for providing access information to the deposited datasets in the Data availability section of your manuscript. The reviewer links and tokens can now be removed from this section. Please make sure that all datasets will be publicly available at the time of publication.
- As per our journal's policy, "data not shown" (on page 18) is not permitted. All data referred to in the paper should be displayed in the main or Expanded View figures, or in the Appendix. Please add these data or change the text accordingly if these data are not central to the study and its conclusions, or properly cite the respective published sources if these data can be found elsewhere.
- We noticed that the following items are called out in the manuscript, but missing: "Supplementary Data 8", "Supplementary Data 7" and "Table EV1"; this should probably be renamed to "Appendix Table S1", as this is missing.
- We also noticed that callouts for Fig. 2I and Dataset EV8 are missing.
- The legend of each EV Dataset should be included as a separate tab/sheet in the respective Excel files.
- The title (first) page of the Appendix PDF file should include the heading "Appendix for" followed by the manuscript's title and a brief Table of Contents including page numbers for the listed items. The Appendix Figures need to be listed sequentially. Please note that the legend for Appendix Table S1 is missing.
- Please note that EMBO press papers are accompanied online by:
 - A) a short (2 sentences) summary of the findings and their significance,
 - B) 2-5 short bullet points highlighting the key results, and
 - C) a synopsis image in .jpg or .png format that is exactly 550 pixels wide and 300-600 pixels high (the height is variable). Please note that all text needs to be legible at the final size.Please upload this information along with your revised manuscript (the text for A and B should be provided in a separate Word file).
- During our standard pre-acceptance Figure checks, we detected cell reuse not detailed in the figure legends between:
 1. Figure 1A and Figure EV1E
 2. Figure 1I and Figure EV2D
 3. Figure EV1A and Figure EV1D.Please check these Figures again and revise them if necessary, or clarify and detail the reuse in the Figure legends if it is intentional and justified.
- During our routine data checks, our data editors have raised the following queries regarding figures, data, and legends. Please make sure that all requests below are completely addressed in the final version of your manuscript (please highlight all changes in the manuscript):

1. Please note that the legend for Figures EV5 F, G is missing in the manuscript. This needs to be rectified.
2. Please note that the exact p values should be provided in the legends of Figures 2I, 5A, B, D, E, F, H; 6B, E, G, I; 7D, G, J, M; EV7 A, B.
3. Please indicate the statistical test used for data analysis in the legends of Figures 4C, 5G; EV4 A-C, F-I; EV5 A, B.
4. Please note that the box plots need to be defined in terms of minima, maxima, centre, bounds of box and whiskers, and percentile in the legends of Figures EV4 F, G, H, I; EV5 A, B.
5. Please note that information related to "n" is missing in the legend of Figure 4C.
6. Please note that the blue and red dotted circles are not defined in the legend of Figure EV1 B. This needs to be rectified.

- We also note that the legend for Movie EV12 is missing; please provide this legend zipped together with the Movie file.

- The order of the manuscript sections must be corrected as follows: Title page - Abstract and Keywords - Introduction - Results - Discussion - Methods - Data Availability - Acknowledgements - Disclosure and Competing Interests Statement - References - Figure Legends - main Tables (if there are any) - Expanded View Figure Legends.

Please also note that as part of the EMBO publications' Transparent Editorial Process, The EMBO Journal publishes online a Peer Review File along with each accepted manuscript. This File will be published in conjunction with your paper and will include the referee reports, your point-by-point response and all pertinent correspondence relating to the manuscript. You can opt out of this by letting the editorial office know (contact@embojournal.org). If you do opt out, the Peer Review File link will point to the following statement: "No Peer Review File is available with this article, as the authors have chosen not to make the review process public in this case."

We look forward to seeing a final version of your manuscript as soon as possible. Please let us know if you have any questions and use this link to submit your revision: <https://emboj.msubmit.net/cgi-bin/main.plex>.

Best regards,

Ioannis

Referee #2:

The authors have nicely addressed all of my suggestions.
Congratulations on a nice study.

Referee #3:

In this revision, the authors have adequately addressed my comments and improved their manuscript to a level suitable for publication. I do not have any additional comments.

All editorial and formatting issues were resolved by the authors.

Dear Santosh,

Congratulations on an excellent manuscript! I am very pleased to inform you that it has been accepted for publication in The EMBO Journal. Thank you for addressing the initially raised referees' concerns, and all editorial requests for changes and corrections.

If you have any questions, please do not hesitate to contact the Editorial Office. Thank you for your contribution to The EMBO Journal. Working with you has been a pleasure!

Best wishes,

Ioannis
